# A Unifying Post-Processing Framework for Multi-Objective Learn-to-Defer Problems

**Mohammad-Amin Charusaie**
Max Planck Institute for Intelligent Systems
Tuebingen, Germany
mcharusaie@tuebingen.mpg.de

**Samira Samadi**
Max Planck Institute for Intelligent Systems
Tuebingen, Germany
samira.samadi@tuebingen.mpg.de

## Abstract

Learn-to-Defer is a paradigm that enables learning algorithms to work not in isolation but as a team with human experts. In this paradigm, we permit the system to defer a subset of its tasks to the expert. Although there are currently systems that follow this paradigm and are designed to optimize the accuracy of the final human-AI team, the general methodology for developing such systems under a set of constraints (e.g., algorithmic fairness, expert intervention budget, defer of anomaly, etc.) remains largely unexplored. In this paper, using a $d$-dimensional generalization to the fundamental lemma of Neyman and Pearson ($d$-GNP), we obtain the Bayes optimal solution for learn-to-defer systems under various constraints. Furthermore, we design a generalizable algorithm to estimate that solution and apply this algorithm to the COMPAS, Hatespeech, and ACSIncome datasets. Our algorithm shows improvements in terms of constraint violation over a set of learn-to-defer baselines and can control multiple constraint violations at once. The use of $d$-GNP is beyond learn-to-defer applications and can potentially obtain a solution to decision-making problems with a set of controlled expected performance measures.

## 1 Introduction

Machine learning algorithms are increasingly used in diverse fields, including critical applications, such as medical diagnostics [72] and predicting optimal prognostics [63]. To address the sensitivity of such tasks, existing approaches suggest keeping the human expert in the loop and using the machine learning prediction as advice [35], or playing a supportive role by taking over the tasks on which machine learning is uncertain [39, 60, 4]. The abstention of the classifier in making decisions, and letting the human expert do so, is where the paradigm of learn-to-defer (L2D) started to exist.

The development of L2D algorithms has mainly revolved around optimizing the accuracy of the final system under such paradigm [60, 50]. Although they achieve better accuracy than either the machine learning algorithm or the human expert in isolation, these works provide inherently single-objective solutions to the L2D problem. In the critical tasks that are mentioned earlier, more often than not, we face a challenging multi-objective problem of ensuring the safety, algorithmic fairness, and practicality of the final solution. In such settings, we seek to limit the cost of incorrect decisions [46], algorithmic biases [13], or human expert intervention [57], while optimizing the accuracy of the system. Although the seminal paper that introduced the first L2D algorithm targeted an instance of such multi-objective problem [44], a general solution to such class of problems, besides specific examples [26, 57, 51, 52], has remained unknown to date. Multi-objective machine learning extends beyond the realm of L2D problems. A prime example that is extensively studied in various settings is ensuring algorithmic fairness [18] while optimizing accuracy. Recent advances in the algorithmic fairness literature have suggested the superiority of *post-processing* methodology for tackling this

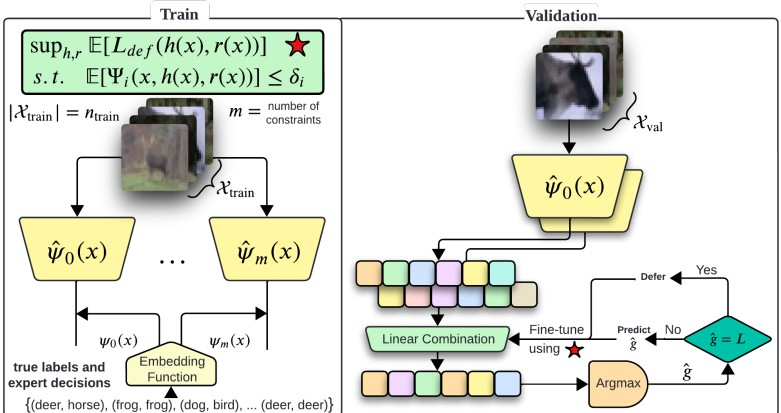

Figure 1: Diagram of applying $d$-GNP to solve multi-objective L2D problem. The role of randomness is neglected due to simplicity of presentation.

multi-objective problem [73, 14, 20, 76]. Post-processing algorithms operate in two steps: first, they find a calibrated estimation of a set of probability scores for each input via learning algorithms, and then they obtain the optimal predictor as a function of these scores. Similarly, in a recent set of works, optimal algorithms to reject the decision-making under a variety of secondary objectives are determined via post-processing algorithms [51, 52], which is in line with classical results such as Chow's rule [16] that is the simplest form of a post-processing method, thresholding the likelihood.

Inspired by the above works, in this paper, we fully characterize the solution to multi-objective L2D problems using a post-processing framework. In particular, we consider a deferral system together with a set of conditional performance measures $\{\Psi_0, \ldots, \Psi_m\}$ that are functions of the system outcome $\hat{Y}$, the target label $Y$, and the input $X$. The goal is to optimize the average value of $\Psi_0$ over data distribution while keeping the average value of the rest of performance measures $\Psi_1, \ldots, \Psi_m$ for all inputs under control. As an example, in binary classification, $\Psi_0$ can be the $0-1$ deferral loss function, while $\Psi_1$ can be the difference between positive prediction rates of $\hat{Y}$ for all instances of $X$ that belong to demographic group $A = 0$ or $A = 1$. The solution for which we aim optimizes the accuracy while assuring that the demographic parity measure between the two groups is bounded by a tolerance value $\delta_1 \in [0, 1]$.

To provide the optimal solution, we move beyond staged learning [12] methodology, in which the classifier $h(x)$ is trained in the absence of human decision-makers, and then the optimal rejection function $r(x)$ is obtained for that classifier to decide when the human expert should intervene ($r(x) = 1$). Instead, we jointly obtain the classifier and rejection function. The reason that we avoid this methodology is that firstly, objectives such as algorithmic fairness are not compositional, i.e., even if the classifier and the human are fair, due to the emergence of Yule's effect [62] the obtained deferral system is not necessarily fair (see Appendix A), and in fact abstention systems can deter the algorithmic biases [36]. Secondly, the feasibility of constraints is not guaranteed under staged learning methodology [74], e.g., there can be cases in which achieving a purely fair solution is impossible, while this occurs neither in vanilla classification [20] nor in our solution.

This paper shows that the joint learning of classifier and rejection function for finding the optimal multi-objective L2D solution boils down to a generalization of the fundamental Neyman-Pearson lemma [55]. This lemma is initially introduced in studying hypothesis testing problems and characterizes the most powerful test (i.e., the test with the highest true positive rate) while keeping the significance level (true negative rate) under control. As a natural extension to this paradigm, we consider a multi-hypothesis setting where for each true positive prediction and false negative prediction, we receive a reward and loss, respectively. Then, we show that the extension of Neyman-Pearson lemma to this setting provides us with a solution for our multi-objective L2D problem.

In summary, the contribution of this paper is as below:

- In Section 3, we show that obtaining the optimal deterministic classifier and rejection function under a constraint is, in general, an NP-Hard problem, then

- by introducing randomness, we rephrase the multi-objective L2D problem into a functional linear programming.
- In Section 4, we show that such linear programming problem is an instance of $d$-dimensional generalized Neyman-Pearson ($d$-GNP) problem, then
- we characterize the solution to $d$-GNP problem, and we particularly derive the corresponding parameters of the solution when the optimization is restricted by a single constraint.
- In Section 5, we show that a post-processing algorithm that is based on $d$-GNP solution generalizes in constraints and objective with the rate $O(\sqrt{\log n/n}, \sqrt{\log(1/\epsilon)/n}, \epsilon')$ and $O((\log n/n)^{1/2\gamma}, (\log(1/\epsilon)/n)^{1/2\gamma}, \epsilon')$, respectively, with probability at least $1 - \epsilon$ where $n$ is the size of the set using which we fine-tune the algorithm, $\epsilon'$ measures the accuracy of learned post-processing scores, and $\gamma$ is a parameter that measures the sensitivity of the constraint to the change of the predictor. Then,
- we show that the use of in-processing methods in L2D problem does not necessarily generalize to the unobserved data, and finally
- we experiment our post-processing algorithm on two tabular datasets and a text dataset, and observe its performance compared to the baselines for ensuring demographic parity and equality of opportunity on final predictions.

Lastly, the $d$-GNP theorem has potential use cases beyond the L2D problem, particularly in vanilla classification problems under constraints. However, such applications are beyond the scope of this paper, and except for a brief explanation of the use of $d$-GNP in algorithmic fairness for multiclass classification, we leave them to future works.

## 2 Related Works

Human and ML's collaboration in decision-making has been demonstrated to enhance the accuracy of final decisions compared to predictions that are made solely by humans or ML [37, 68]. This overperformance is due to the ability to estimate the accuracy and confidence of each agent on different regions of data and subsequently allocate instances between human and ML to optimize the overall accuracy [2]. Since the introduction of the L2D problem, the implementation of its optimal rule has been the focus of interest in this field [8, 50, 12, 51, 9, 43, 48, 45]. The multi-objective classification with abstention problems is studied for specific objectives in [44, 57, 48] via in-processing methods. The application of Neyman-Pearson lemma for learning problems with fairness criteria is recently introduced in [75].
We refer the reader to Appendix B for further discussion on related works.

## 3 Problem Setting

Assume that we are given input features $x_i \in \mathcal{X}$, corresponding labels $y_i \in \mathcal{Y} = \{1, \ldots, L\}$, and the human expert decision $m_i$ for such input, and assume that these are i.i.d. realizations of random variables $X, Y, M \sim \mu = \mu_{XYM}$. Since there exists randomness in the human decision-making process, for the sake of generality, we treat $M$ as a random variable similar to $Y$ and do not assume that $m_i = m(x_i)$ for some function $m$. Further, assume that for the true label $y$ and a certain feature vector $x$, the cost of incorrect predictions is measured by a loss function $\ell_{AI}(y, h(x))$ for the classifier prediction $h(x)$, and a loss function $\ell_H(y, m)$ for human's prediction $m$. The question that we tackle in this paper is the following: *What is an optimal classifier and otherwise an optimal way of deferring the decision to the human when there are constraints that limit the decision-making?* The constraints above can be algorithmic fairness constraints (e.g., demographic parity, equality of opportunity, equalized odds), expert intervention constraints (e.g., when the human expert can classify up to $b$ proportion of the data), or spatial constraints to enforce deferral on certain inputs, or any combination thereof.
Let us put the above question in a formal optimization form. To that end, let $r(x) \in \{0, 1\}$ be the rejection function[1], i.e., when $r(x) = 0$ the classifier makes the decision for input $x$ and otherwise $x$ is deferred to the expert. We obtain the deferral loss on $x$ and given a label $y$ and the expert decision $m$ as

$$\ell_{\text{def}}(y, m, h(x), r(x)) = r(x)\ell_H(y, m) + (1 - r(x))\ell_{AI}(y, h(x)).$$

---

[1]The rejection here differs from hypothesis rejection and indicates that the classifier rejects making a decision and defers the decision to the human expert.

Table 1: A list of embedding functions corresponding to the constraints that are discussed in Section 3. This list is a version of the results in Appendix D when we assume that the input feature contains demographic group identifier $A$. To simplify the notations, we define $t(A, y) := \frac{\mathbb{I}_{A=1}}{Pr(Y=y, A=1)} - \frac{\mathbb{I}_{A=0}}{Pr(Y=y, A=0)}$.

| Name | Embedding Function $\psi_i(x)$ |
|---|---|
| Accuracy | $[\Pr(Y=0\mid x), \ldots, \Pr(Y=n\mid x), \Pr(Y=M\mid x)]$ |
| Expert Intervention Budget [57] | $[0, \ldots, 0, 1]$ |
| OOD Detection [53] | $[0, \ldots, 0, \frac{f_X^{\text{out}}(x)}{f_X^{\text{in}}(x)}]$ |
| Long-Tail Classification [52] | $-\left[\sum_{i=1}^K \frac{\Pr(Y\neq 1, Y\in G_i\mid X=x)}{\alpha_i \Pr(Y\in G_i)}, \ldots, \sum_{i=1}^K \frac{\Pr(Y\neq l, Y\in G_i\mid X=x)}{\alpha_i \Pr(Y\in G_i)}, 0\right]$ and $\frac{\Pr(Y\in G_i\mid X=x)}{\Pr(Y\in G_i)}\left[1, \ldots, 1, 0\right] - \frac{\alpha_i}{K}$ |
| Bound on Type-$K$ Error [69] | $\frac{\Pr(Y=k\mid x)}{\Pr(Y=k)}[1, \ldots, \underbrace{0}_{k\text{-th}}, \ldots, 1, \Pr(M\neq k\mid Y=k, x)]$ |
| Demographic Parity [28] | $(\frac{\mathbb{I}_{A=1}}{Pr(A=1)} - \frac{\mathbb{I}_{A=0}}{Pr(A=0)})[0, 1, \Pr(M=1\mid x)]$ |
| Equality of Opportunity [34] | $t(A, 1)[0, \Pr(Y=1\mid x), \Pr(M=1, Y=1\mid x)]$ |
| Equalized Odds [34] | $t(A, 1)[0, \Pr(Y=1\mid x), \Pr(M=1, Y=1\mid x)]$ and $t(A, 0)[\Pr(Y=0\mid x), 0, \Pr(M=0, Y=0\mid x)]$ |

Therefore, we can find the average deferral loss on distribution $\mu$ as

$$L_{\text{def}}^\mu(h, r) := \mathbb{E}_{X,Y,M\sim\mu}\big[\ell_{\text{def}}(Y, M, h(X), r(X))\big]. \tag{1}$$

We aim to find a randomized algorithm $\mathcal{A}$ that defines a probability distribution $\mu_{\mathcal{A}}$ on $\mathcal{H} \times \mathcal{R}$ that solves the optimization problem

$$\mu_{\mathcal{A}} \in \underset{\mu_{\mathcal{A}}}{\operatorname{argmin}} \mathbb{E}_{(h,r)\sim\mathcal{A}}\big[L_{\text{def}}^\mu(h, r)\big],$$

$$s.t. \quad \mathbb{E}_{X,Y,M\sim\mu}\mathbb{E}_{(h,r)\sim\mu_{\mathcal{A}}}\big[\Psi_i\big(X, Y, M, h(X), r(X)\big)\big] \leq \delta_i \tag{2}$$

where $\Psi_i$ is a performance measure that induces the desired constraint in our optimization problem. We assume that $\Psi_i$, similar to $\ell_{\text{def}}$, is an *outcome-dependent* function, i.e., if the deferral occurs, the outcome of the classifier does not change $\Psi_i$, and otherwise, if deferral does not occur, the human decision does not change $\Psi_i$. In other words, the value of the constraints can only be a function of input feature $x$ and of the deferral system prediction $\hat{Y} = r(x)M + \big(1 - r(x)\big)h(x)$. Here, $\hat{Y}$ is the expert decision when deferral occurs, and is the classifier decision otherwise.

**Types of constraints.** Before we discuss our methodology to solve (2), it is beneficial to review the types of constraints with which we are concerned: **(1) expert intervention budget** that can be written in form of $\Pr\big(r(X)=1\big) \leq \delta$, limits the rejection function to defer up to $\delta$ proportion of the instance, **(2) demographic parity** that is formulated as $\big|P(\hat{Y}=1\mid A=0) - P(\hat{Y}=1\mid A=0)\big| \leq \delta$, ensures that the proportion of positive predictions for the first demographic group ($A=0$) is comparable to that for the second demographic group ($A=1$). **(3) equality of opportunity** that is defined as $\big|Pr(\hat{Y}=1\mid A=1, Y=1) - Pr(\hat{Y}\mid A=0, Y=1)\big| \leq \delta$ limits the differences between correct positive predictions among two demographic groups, **(4) equalized odds** that is similar to equality of opportunity but targets the differences of correct positive and negative predictions among two groups, i.e., $\max_{y=0,1}\big|Pr(\hat{Y}=1\mid A=1, Y=y) - Pr(\hat{Y}=1\mid A=0, Y=y)\big| \leq \delta$, **(5) out-of-distribution (OOD) detection** that is written as $\Pr_{\text{out}}(r(X)=0) \leq \delta$ limits the prediction of the classifier on points that are outside its training distribution and incentivizes deferral in such cases, **(6) long-tail classification** deals with high class imbalances. This method aims to minimize a balanced error of classifier prediction on instances where deferral does not occur. Achieving this objective as

mentioned in [53] is equivalent to minimizing $\sum_{i=1}^{K} \frac{1}{\alpha_i} \Pr(Y \neq h(X), r(X) = 0 | Y \in G_i)$ when the feasible set is $\Pr(r(X) = 0, Y \in G_i) = \frac{\alpha_i}{K}$, and where $\{G_i\}_{i=1}^{K}$ is a partition of classes, and finally **(7) type-$k$ error bounds** that is a generalization of Type-I and Type-II errors, limits errors of a specific class $k$ using $\Pr(\hat{Y} \neq k | Y = k) \leq \delta$.

All above constraints are expected values of outcome-dependent functions (see Appendix D for proof). To put it informally, if we change the classifier outcome after the rejection, such constraints do not vary.

**Linear Programming Equivalent to** (2)**.** The outcome-dependence property helps us to show that (see Appendix C) obtaining the optimal classifier and rejection function is equivalent to obtaining the solution of

$$f^* = [f_1^*, \ldots, f_d^*] \in \underset{f \in \Delta_d^{\mathcal{X}}}{\arg\max} \, \mathbb{E}\big[\langle f(X), \psi_0(X)\rangle\big], \quad \text{s.t. } \mathbb{E}\big[\langle f(x), \psi_i(x)\rangle\big] \leq \delta_i, i \in [1:m] \quad (3)$$

where $\Delta_d$ is a simplex of $d$ dimensions, $d = L + 1$, and $\psi_i : \mathcal{X} \to \mathbb{R}^d$ is defined as

$$\psi_i(x) := \mathbb{E}_{Y,M|X=x}\Big[\big[\Psi_i(x, Y, M, 1, 0), \ldots, \Psi_i(x, Y, M, l, 0), \Psi_i(x, Y, M, 0, 1)\big]\Big] \quad (4)$$

that we name the *embedding function*[2] corresponding to the performance measure $\Psi_i$ for $i \in [0:m]$, where for simplifying the notation we define $\Psi_0 \equiv -\ell_{\text{def}}$. Furthermore, the optimal algorithm is obtained by predicting $h(x) = i$ with normalized probability of $f_i^*(x) / \sum_{j=1}^{d-1} f_j^*(x)$, where $\sum_{j=1}^{d-1} f_j^*(x) \neq 0$, and rejecting $r(x) = 1$ with probability $f_d^*(x)$. In case of $\sum_{j=1}^{d-1} f_j^*(x) = 0$ the classifier is defined arbitrarily. A list of embedding functions for the mentioned constraints and objectives is provided in Table 1 (See Appendix D for derivations).

**Hardness.** We first derive the following negative result for the optimal deterministic predictor in (3). We use the similarity between (3) and $0-1$ Knapsack problem (see [58, pp. 374]) to show that there are cases in which solving the former is equivalent to solving an NP-Hard problem. More particularly, if we assume that the distribution of $X$ contains finite atoms $x_1, \ldots, x_n$, each of which have probability of $\Pr(X = x_i) = p_i$, and if we set $\psi_1(x_i) = [0, \frac{w_i}{p_i}]$ and $\psi_0(x_i) = [0, \frac{v_i}{p_i}]$ for $v_i, w_i \in \mathbb{R}^+$, then (3) reduces in $\arg\max \sum_i f^1(x_i) v_i$ subjected to $f^1 : \mathcal{X} \to \{0, 1\}$ and $\sum_i f^1(x_i) w_i \leq \delta_1$, which is the main form of the Knapsack problem. In the following theorem, we show that a similar result can be obtained if we choose $\psi_0$ and $\psi_1$ to be embedding functions corresponding to accuracy and expert intervention budget. All proofs of theorems can be found in the appendix.

**Theorem 3.1** (NP-Hardness of (2))**.** *Let the human expert and the classifier induce $0 - 1$ losses and assume $\mathcal{X}$ to be finite. Finding an optimal deterministic classifier and rejection function for a bounded expert intervention budget is an NP-Hard problem.*

Note that the above finding is different from the complexity results for deferral problems in [49, Theorem 1] and [23, Theorem 1]. NP-hardness results in these settings are consequences of restricting the search to a specific space of models, i.e., the intersection of half-spaces and linear models on a subset of the data. However, in our theorem, the hardness arises due to a possibly complex data distribution and not because of the complex model space.

The above hardness theorem for deterministic predictors justifies our choice of using randomized algorithms to solve multi-objective L2D. In the next section, by finding a closed-form solution for the randomized algorithm, we show that such relaxation indeed simplifies the problem.

## 4 $d$-dimensional Generalization of Neyman-Pearson Lemma

The idea behind minimizing an expected error while keeping another expected error bounded is naturally related to the problem that is designed by Neyman and Pearson [55]. They consider two hypotheses $H_0, H_1$ as two distributions with density functions $g_0(x)$ and $g_1(x)$ for which a given point $x$ can be drawn. Then, they maximize the probability of correctly rejecting $H_0$, while bounding the probability of incorrectly rejecting $H_0$, i.e., for a test $T(x) \in [0, 1]$ that rejects the null hypothesis when $T(x) = 1$, they solved the problem

$$\max_{T \in [0,1]^{\mathcal{X}}} \mathbb{E}_{X \sim g_1}\big[T(X)\big], \quad s.t. \, \mathbb{E}_{X \sim g_0}\big[T(X)\big] \leq \alpha. \quad (5)$$

---

[2]We named this an embedding function because it embeds the constraint or loss of the optimization problem into a vector function.

They concluded that thresholding the likelihood ratio is a solution to the above problem. Formally, they show that all optimal hypothesis tests take the value $T(x) = 1$ when $g_1(x)/g_0(x) > k$ and take the value $T(x) = 0$ when $g_1(x)/g_0(x) < k$, where $k$ is a scalar and dependent on $\alpha$.

**Multi-hypothesis testing with rewards.** In this section, we aim to solve (3) as a generalization of Neyman-Pearson lemma for binary testing to the case of multi-hypothesis testing, in which correctly and incorrectly rejecting each hypothesis has a certain reward and loss. To clarify how the extension of this setting and the problem (3) are equivalent, assume the general case of $d$ hypotheses $H_0, \ldots, H_{d-1}$, each of which corresponding to $X$ being drawn from the density function $g_i(x)$ for $i \in \{0, \ldots, d-1\}$. Further, assume that for each hypothesis $H_i$, in case of true positive, we receive the reward $r_i(x)$, and in case of false negative, we receive the loss $\ell_i(x)$. Assume that we aim to find a test $f : \mathcal{X} \to \Delta_d$ that for each input $x \in \mathcal{X}$ rejects $d-1$ hypotheses, each hypothesis $H_i$ with probability $1 - f^i(x)$ and maximizes a sum of true positive rewards, and that keeps the sum of false negative losses under control. Then, this is equivalent to $\operatorname*{argmax}_{f \in \Delta_d^{\mathcal{X}}} \sum_{i=0}^{d-1} \mathbb{E}_{X \sim g_i}\left[f^i(x)r_i(x)\right]$ subjected to $\sum_{i=0}^{d-1} \mathbb{E}_{X \sim g_i}\left[(1 - f^i(x))\ell_i(x)\right] \leq \delta_1$ which in turns is equivalent to

$$\operatorname*{argmax}_{f \in \Delta_d^{\mathcal{X}}} \mathbb{E}_{X \sim g_0}\left[\sum_{i=0}^{d-1} f^i(x)r_i(x)\frac{g_i(x)}{g_0(x)}\right] \quad \text{s.t.} \quad \mathbb{E}_{X \sim g_0}\left[\sum_{i=0}^{d-1} f^i(x)\sum_{j \neq i}\ell_j(x)\frac{g_j(x)}{g_0(x)}\right] \leq \delta_1. \quad (6)$$

This problem can be seen as instance of (3), when we set $\psi_0(x) = [r_0(x), \ldots, r_{d-1}(x)\frac{g_{d-1}(x)}{g_0(x)}]$ and $\psi_1(x) = \left[\sum_{j \neq 0}\ell_j(x)\frac{g_j(x)}{g_0(x)}, \ldots, \sum_{j \neq d-1}\ell_j(x)\frac{g_j(x)}{g_0(x)}\right]$. Similarly, we can show that for all $\psi_0(x), \psi_1(x)$ in (3) there exists a set of densities $g_1(x), \ldots, g_{d-1}(x)$ and rewards and losses such that (6) and (3) are equivalent. This can be done by setting $g_i \equiv g_0$ and noting that the mapping from $\ell_i$s and $r_i$s into $\psi_0$ and $\psi_2$ is invertible.

The formulation of (3) can be seen as an extension of the setting in [69] when we move beyond type-$k$ error bounds to a general set of constraints. That work achieves the optimal test by applying strong duality on the Lagrangian form of the constrained optimization problem. However, we avoided using this approach in proving our solution, since finding $f^*$, and not the optimal objective, is possible via strong duality only when we know apriori that the Lagrangian has a single saddle point (for more details and fallacy of such approach, see Section E). As another improvement to the duality method, we not only find a solution to (3), but also show that there is no other solution that works as well as ours.

Before we express our solution in the following theorem, we define an import notation as an extension of the argmax function that helps us articulate the optimal predictor. In fact, we define

$$\mathcal{T}_d = \left\{\tau : \mathbb{R}^d \times \mathbb{R}^d \to \Delta_d \mid \sum_{i:x_i = \max\{x_1,\ldots,x_d\}} \left(\tau(\mathbf{x}_1^d, \cdot)\right)(i) = 1\right\} \quad (7)$$

that is a set of functions that result in one-hot encoded argmax when there is a clear maximum, and otherwise, based on its second argument, results in a probability distribution on all components that achieved the maximum value.

**Theorem 4.1** ($d$-GNP). *For a set of functions $\psi_i$ where $i \in [0, m]$, assume that $(\delta_1, \ldots, \delta_m)$ is an interior point[3] of the set $\mathcal{F} = \left\{\left(\mathbb{E}[\langle r(x), \psi_1(x)\rangle], \ldots, \mathbb{E}[\langle r(x), \psi_m(x)\rangle]\right) : f \in \Delta_d^{\mathcal{X}}\right\}$. Then, there is a set of fixed values $k_1, \ldots, k_m$ and $\tau \in \mathcal{T}_d$ such that the predictor*

$$f^*(x) = \tau\left(\psi_0(x) - \sum_{i=1}^{m} k_i\psi_i(x), x\right), \quad (8)$$

*obtains the optimal solution of $\sup_{f \in \Delta_d^{\mathcal{X}}} \mathbb{E}[\langle f(x), \psi_0(x)\rangle]$, subjected to the constraints being achieved tightly, i.e., when for $i \in [1 : m]$ we have $\mathbb{E}[\langle f(x), \psi_i(x)\rangle] = \delta_i$. If $k_1, \ldots, k_m$ are further non-negative, then $f^*(x)$ is the optimal solution to (3). Moreover, all optimal solutions of (3) that tightly achieve the constraints are in form of (8) almost everywhere on $\mathcal{X}$.*

**Example 1** (L2D with Demographic Parity). In the setting that we have a deferral system and we aim for controlling demographic disparity under the tolerance $\delta$, we can set $\psi_0(x) = \left[\Pr(Y = \right.$

---

[3]A point is an interior point of a set, if the set contains an open neighborhood of that point.

$0|x), \Pr(Y = 1|x), \Pr(Y = M|x)]$ and $\psi_1(x) = s(A)\big[0, 1, \Pr(M = 1|x)\big]$, using Table 1, where $s(A) := \big(\frac{\mathbb{I}_{A=1}}{Pr(A=1)} - \frac{\mathbb{I}_{A=0}}{Pr(A=0)}\big)$. Therefore, $d$-GNP, together with the discussion after (4) shows that the optimal classifier and rejection function are obtained as

$$h(x) = \begin{cases} 1 & \Pr(Y = 1|x) > \frac{1+ks(A)}{2} \\ 0 & \Pr(Y = 1|x) < \frac{1+ks(A)}{2} \end{cases},$$

and

$$r(x) = \begin{cases} 1 & \Pr(Y = M|x) - ks(A)\Pr(M = 1|x) > \lambda(A, x) \\ 0 & \Pr(Y = M|x) - ks(A)\Pr(M = 1|x) < \lambda(A, x) \end{cases},$$

for a fixed value $k \in \mathbb{R}$, and where $\lambda(A, x) := \max\{\Pr(Y = 0|x), \Pr(Y = 1|x) - ks(A)\}$. The above identities imply that the optimal fair classifier for the deferral system thresholds the scores for different demographic groups using two thresholds $ks(0)$ and $ks(1)$. This is similar in form to the optimal fair classifier in vanilla classification problem [14, 20]. However, the rejection function does not merely threshold the scores for different groups, but adds an input-dependent threshold $ks(A)\Pr(M = 1|x)$ to the unconstrained deferral system scores.

It is important to note that although we have a thresholding rule for the classifier, the thresholds are not necessarily the same as of isolated classifier under fairness criteria. Furthermore, the deferral rule is dependent on the thresholds that we use for the classifier. Therefore, we cannot train the classifier for a certain demographic parity and a rejection function in two independent stages. This further affirms the lack of compositionality of algorithmic fairness that we discussed earlier in the introduction of this paper.

**Example 2** (L2D with Equality of Opportunity). Here, similar to the previous example, we can obtain the embedding function for accuracy and equality of opportunity constraint as $\psi_0(x) = \big[p_x^0, p_x^1, p_x^M\big]$ and $\psi_1(x) = t(A, 1)\big[0, p_x^1, \Pr(M = 1, Y = 1|x)\big]$, respectively, where $p_x^i := \Pr(Y = i|x)$ for $i \in \{1, 2\}$ and similarly $p_x^M = \Pr(Y = M|x)$. Therefore, the characterization of optimal classifier and rejection function using $d$-GNP results in

$$h(x) = \begin{cases} 1 & \big(2 - kt(A, 1)\big)p_x^1 > 1 \\ 0 & \big(2 - kt(A, 1)\big)p_x^1 < 1 \end{cases},$$

and

$$r(x) = \begin{cases} 1 & p_x^M\big(1 - kt(A, 1)\Pr(M = 1|Y = M, x)\big) > \nu(A, x) \\ 0 & p_x^M\big(1 - kt(A, 1)\Pr(M = 1|Y = M, x)\big) < \nu(A, x) \end{cases},$$

for $k \in \mathbb{R}$ and where $\nu(A, x) := \max\{p_x^0, \big(1 - kt(A, 1)\big)p_x^1\}$. Assuming $2 - kt(A, 1)$ takes positive values for all choices of $A$, we conclude that the optimal classifier is to threshold positive scores differently for different demographic groups. However, the optimal deferral is a function of probability of positive prediction by human expert.

**Example 3** (Algorithmic Fairness for Multiclass Classification). In addition to addressing the L2D problem, the formulation of $d$-GNP in Theorem 4.1 allows for finding the optimal solution in vanilla classification. In fact, for an $L$-class classifier, if we aim to set constraints on demographic parity $\big|\Pr(\hat{Y} = 0|A = 0) - \Pr(\hat{Y} = 0|A = 1)\big| \leq \delta$ or equality of opportunity $\big|\Pr(\hat{Y} = 0|Y = 0, A = 0) - \Pr(\hat{Y} = 0|Y = 0, A = 1)\big| \leq \delta$ on Class 0, then we can follow similar steps as in Appendix D to find the embedding functions as $\psi_{\text{DP}} = s(A)\big[1, 0, \ldots, 0\big]$ and $\psi_{\text{EO}} = t(A, 0)\big[p_x^0, 0, \ldots, 0\big]$, where $p_x^i := \Pr(Y = i|x)$ for $i \in [L]$.

As a result, since the accuracy embedding function is $\psi_0(x) = \big[p_x^0, \ldots, p_x^L\big]$, then, by neglecting the effect of randomness, the optimal classifier under such constraints are as

$$h_{\text{DP}}(x) = \text{argmax}\,\big\{p_x^0 - ks(A), p_x^1, \ldots, p_x^L\big\},$$

and

$$h_{\text{EO}}(x) = \text{argmax}\,\big\{p_x^0\big(1 - kt(A, 0)\big), p_x^1, \ldots, p_x^L\big\}.$$

Equivalently, for demographic parity, the optimal classifier includes a shift on the score of Class 0 as a function of demographic group, and for equality of opportunity, the optimal classifier includes a multiplication of the score of Class 0 with a value that is a function of demographic group. It is easy to show that under condition of positivity of the multiplied value, these classifiers both reduce to thresholding rules in binary setting.

Note that although Theorem 4.1 characterizes the optimal solution of (3), it leaves us uninformed regarding parameters $k_1, \ldots, k_m$, and further does not give us the form of the optimal solution when $\psi_0(x) - \sum_{i=1}^m k_i \psi_i(x)$ has more than one maximizer. In the following theorem, we address these issues for the case that we have a single constraint.

**Theorem 4.2** ($d$-GNP with a single constraint). *The optimal solution* (8) *of the optimization problem* (3) *with one constraint is equal to* $f_{k,p}^*(x) = \tau\big(\psi_0(x) - k\psi_1(x), x\big)$ *where* $\tau$ *is a member of* $\mathcal{T}_d$ *such that if there is a non-singleton set* $\mathcal{I}$ *of maximizers of a vector* $\mathbf{y} \in \mathbb{R}^d$, *then we have* $\big(\tau(\mathbf{y}, x)\big)(i) = p$ *and* $\big(\tau(\mathbf{y}, x)\big)(j) = 1 - p$, *where* $i$ *and* $j$ *are the first indices in* $\mathcal{I}$ *that minimizes* $\psi_1(x)$, *and maximizes* $\psi_0(x)$, *respectively. In this case,* $k$ *is a member of the set* $\mathcal{K} = \Big\{ t : \delta \in \big[ \lim_{\tau \uparrow t} C(\tau), C(t) \big] \Big\}$ *where* $C(t) = \mathbb{E}\big[\langle f_{t,0}^*(x), \psi_0(x)\rangle\big]$ *is the expected constraint of the predictor* $f_{t,0}^*$. *Moreover,* $p = \frac{C(k) - c}{C(k) - \lim_{\tau \uparrow t^-} C(\tau)}$, *if* $C(\cdot)$ *is lower-discontinuous at* $k$, *and otherwise* $p = 0$.

This theorem reduces the complexity of finding $k_i$s from the complexity of an exhaustive search to the complexity of finding the root of the monotone function $C(t) - \delta$ (see Lemma J.2 for the proof of monotonicity), and further finds the randomized response for the cases that Theorem 4.1 leaves undetermined.

Before we proceed to the designed algorithm based on $d$-GNP, we should address two issues. Firstly, during the course of optimization, it can occur that the solution of Theorem 4.1 does not compute non-negative values $k_i$ for an $i \in [1 : m]$. This means that the constraints are not achieved tightly in the final solution of (3). Therefore, we are able to achieve the optimal solution with the constraint $\delta_i' < \delta_i$. Now, if we can assure that the constraint tuples are still inner points of $\mathcal{F}$ when we substitute $\delta_i$ by $\delta_i'$, then Theorem 4.1 shows that (8) is still an optimal solution to (3).

Secondly, for tackling various objectives that are defined in Section 3, we usually need to upper- and lower-bound a performance measure by $\delta$ and $-\delta$. However, since both bounds cannot hold tightly and simultaneously unless the tolerance is $\delta = 0$, then we can use only one of the constraints in turn and apply the result of Theorem 4.2 and check whether the constraint is active in the final solution. In the next section, we design an algorithm based on these results and show its generalization to the unseen data.

## 5   Empirical $d$-GNP and its Statistical Generalization

In previous sections, we obtained the optimal solution to the constrained optimization problem (3) using $d$-GNP. Based on this optimal solution, we can design a plug-in method (see Algorithm 1 in Appendix F) to solve the constrained learning problem using empirical data. This algorithm varies from many Lagrangian-based algorithms for solving constrained learning problem (e.g., Primal-Dual method [10]) in which the optimal predictor parameter and constraint penalties are dependent to each other, and therefore we should learn them iterativaly. However, as we saw in Theorem 5.1 (respectively in Algorithm 1), the solution of $d$-GNP is a mere thresholding on the corresponding embedding functions, where the threshold is obtained in a post-hoc manner and from validation dataset. Therefore, although Lagrangian-based algorithms can lead to oscillations or converge with a large computational cost, the $d$-GNP can potentially reduce such complexity costs and improve convergence conditions. To show such convergence, we bound the generalization error of the objective and constraints based on this solution. These results are extensions of the generalization results for Neyman-Pearson [1, 71] and further hold when multiple constraints should be controlled at once. The first result is the following theorem that shows if the solution to our plug-in method meets constraints of the optimization problem on training data, this generalizes to the unseen data.

**Theorem 5.1** (Generalization of the Constraints). *For the approximation of the Neyman-Pearson solution* $\hat{f}_{\hat{k},\hat{p}}(x)$ *in Algorithm 1 such that* $\mathbb{E}_{S^n}\big[\langle \hat{f}_{\hat{k},\hat{p}}(x), \hat{\psi}_i(x)\rangle\big] \leq \delta_i$ *for* $i \in [1 : m]$, *if we assume that embedding functions are bounded, then for* $d_n(\epsilon) \simeq O(\frac{\sqrt{\log n} + \sqrt{\log 1/\epsilon}}{\sqrt{n}})$ *and* $S^n \sim \mu$ *we have* $\mathbb{E}_\mu\big[\langle \hat{f}_{\hat{k},\hat{p}}(x), \psi_i(x)\rangle\big] \leq \delta_i + d_n(\frac{\epsilon}{m})$ *for all* $i \in [1 : m]$ *and with probability at least* $1 - \epsilon$.

In the above theorem, we show that the optimal empirical solution for the constraint, probably and approximately satisfies the constraint on true distribution. Therefore, if we assume that we have an approximation $\hat{\psi}_i(x)$ in hand where $\|\hat{\psi}_i(x) - \psi_i(x)\|_\infty \leq \epsilon'$ with high probability, this theorem together with Hölder's inequality shows that we need to assure $\mathbb{E}_{S^n}\big[\langle \hat{f}_{\hat{k},\hat{p}}(x), \hat{\psi}_i(x)\rangle\big] \leq \delta - d_n(\frac{\epsilon}{m}) - \epsilon'$ to achieve the corresponding generalization with high probability.

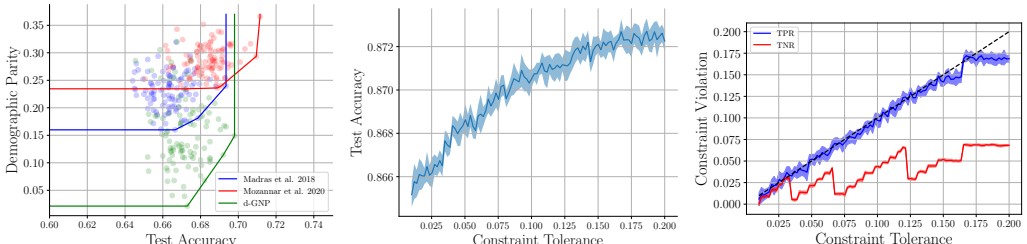

Figure 2: Performance of $d$-GNP on COMPAS dataset (left), and ACSIncome (center and right)

Next, we ask whether the objectives of the empirical optimal solution and the true optimal solution are close. We answer to this question positively in the following theorem. First, however, let us define the notions of $(\gamma, \Delta)$-sensitivity condition as the following. This is an extension to detection condition in [71] and assumes that changing the parameter in predictor leads to a detectable change in constraints.

**Definition 5.2.** For an embedding function $\psi_1$, and a distribution $\mu_X$ on $\mathcal{X}$, we refer to a function $r_k(x)$ as a prediction with $(\gamma, \Delta)$-sensitivity around $k$, if there exists $C \in \mathbb{R}^+$ such that for all $\delta \in (0, \Delta]$ we have

$$\left| \mathbb{E}_{\mu_X} \left[ \langle r_k(x) - r_{k+\delta}(x), \psi_1(x) \rangle \right] \right| \geq C\delta^\gamma. \tag{9}$$

Now, we express the following generalization theorem for predictors that address the above conditions:

**Theorem 5.3** (Generalization of Objective). *Assume that $(\delta - \epsilon_l, \delta + \epsilon_u)$ is a subset of of all achievable constraints $\mathbb{E}\left[ \langle f(x), \psi_1(x) \rangle \right]$, and that $\|\psi_i(x)\|_\infty \leq 1$ for $i = 1, 2$. Further, let the size $n$ of validation data be large enough such that $d_n(\delta/3) \leq \frac{\epsilon_l}{2}$. Now, if the optimal predictor $f^*_{k,0}(x)$ is $(\gamma, \Delta)$-sensitive around optimal $k^*$ for $\Delta \simeq \Omega\left(d_n^{1/\gamma}(\delta/3), \delta_0^{1/2\gamma}, \delta_1^{1/2\gamma}\right)$ and $\gamma \leq 1$, then for $n \geq \frac{16}{\epsilon_l^2} \log \frac{3}{\delta}$, and with probability at least $1 - \delta$, the optimal empirical classifier, as of Algorithm 1 has an objective that is at most $O\left(d_n^{1/\gamma}(\delta/3), \delta_0^{1/\gamma}, \delta_0^{1/2}, \delta_1^{1/2}, C^{-1/\gamma}, C^{-1/2}\right)$-far from the true optimal objective.*

Now that we have proven generalization of our post-processing method, we should briefly compare this to other possible algorithms to learn an approximation of the optimal classifier and rejection function pair. A possible method is to find the appropriate 'defer' or 'no defer' value for each instance in the training dataset, and for a given set of constraints. Although these types of in-processing algorithms can perform computationally efficient (e.g., $O(n \log n)$ complexity for $\frac{1}{n}$-suboptimal solution for human intervention budget as shown in Theorem G.1), they do not necessarily generalize to unseen data. In particular, we can show that for all algorithms that estimate *deferral labels* from empirical data, there exist two underlying distributions on the data on which the algorithm results in similar deferral labels, while the optimal rejection functions for these two distributions are not interchangeable. This argument is further formalized in the following proposition:

**Proposition 5.4** (Impossibility of generalization of deferral labels). *For every deterministic deferral rule $\hat{r}$ for empirical distributions and based on the two losses $\mathbb{1}_{m \neq y}$ and $\mathbb{1}_{h(x) \neq y}$, there exist two probability measures $\mu_1$ and $\mu_2$ on $\mathcal{X} \times \mathcal{Y} \times \mathcal{M}$ such that the corresponding $(\hat{r}, X)$ for both measures is distributed equally. However, the optimal deferral $r^*_{\mu_1}$ and $r^*_{\mu_2}$ for these measures are not interchangeable, that is $L^{\mu_i}_{\text{def}}(h, r^*_{\mu_i}) \leq \frac{1}{3}$ while $L^{\mu_i}_{\text{def}}(h, r^*_{\mu_j}) = \frac{2}{3}$ for $i = 1, 2$ and $j \neq i$.*

In a nutshell, this proposition implies that, every algorithm that reduces the two-bit data of human accuracy and AI accuracy for an input into a single-bit data of 'defer' or 'no defer' looses the information that is important for obtaining the optimal rejection function that generalizes to the unseen data. This is a drawback of in-processing algorithms that are used in multi-objective L2D problems. We refer the reader to Appendix M for more details and proof of aforementioned proposition.

## 6 Experiments

**COMPAS dataset.** We implemented [4] Algorithm 1, first for COMPAS dataset [27] in which the recidivism rate of 7214 criminal defendants is predicted. The human assessment is done in this

---

[4]The code is available in `https://github.com/AminChrs/PostProcess/`.

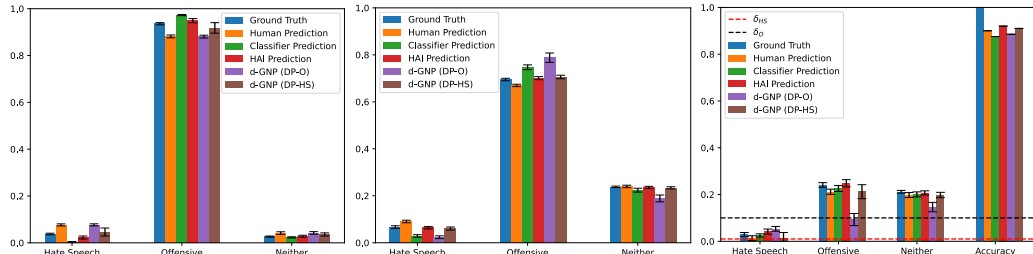

Figure 3: Prediction of $d$-GNP on Hatespeech dataset [22] and for tweets with predicted African-American (left), and Non-African-American (center) dialect and the disparity between groups (right).

dataset on 1000 cases by giving humans a description of the case and asking them whether the defendant would recidivate within two years of their most recent crime.[5] The demographic parity is assessed for two racial groups of white and non-white defendants. Figure 2 shows the average performance of $d$-GNP over 10 random seeds compared to two baselines: (1) Madras et al. [44] in which a demographic parity regularizer is added to the surrogate loss, and over a variation of 100 regularizer coefficient, and (2) Mozannar et al. [50] in which after training the classifier and rejector pair, we shift the corresponding scores to find a new thresholding rule. All scores, classifiers, and rejection functions are trained on a 1-layer feed-forward neural network. The figure shows that achieving better fairness criteria is possible using $d$-GNP, while this might not lead to better accuracy when the constraint violation is not of interest.

**Hatespeech dataset.** The next experiment is on flagging offensive tweets in Hatespeech dataset [22]. This dataset contains 24,802 tweets that are labeled by at least three crowd workers as hate speech, offensive but not hate speech, or neither hate speech nor offensive. We used a pre-trained model [5] to detect whether the tweet contains an African-American dialect. Next, we used $d$-GNP method to control the demographic disparity of predicting a tweet hate speech or offensive bounded by $\delta_{HS} = 0.1$ and $\delta_O = 0.01$. In the result of this experiment that is displayed in Figure 3 we can observe the following points: (i) in test-time the resulting demographic disparity for both classes are bounded as expected, (ii) the accuracy of $d$-GNP method is bounded by the vanilla deferral method, while stricter constraint control (in here offensive prediction parity) keeps the accuracy lower, and (iii) interestingly, the performance of $d$-GNP for controlled offensive prediction parity copies that of human. Therefore, a good strategy for obtaining such constrained learn-to-defer system seems to be to defer the offensive tweet prediction to human, when the tweet contains African-American dialect, and otherwise either bias the classifier scores or use a mixture of human and classifier involvement to achieve the final controlled disparity.

**ACS dataset.** We further tested our method on `folktables` dataset [25] that contains an income prediction task based on 1.6M rows of American Community Survey data. Since we had no access to human expert data, we simulated a human expert that has different accuracy on two racial groups of white and non-white individuals (85% and 60%, respectively). We considered the L2D problem with bounded equalized odds violation. Figure 2 shows our method's accuracy and constraint violation, coupled with a confidence bound that is obtained using ten iterations of bootstrapping. This figure shows that violation bounds are accurately met for the test data, and the performance increases when these bounds are loosened.

## 7   Conclusion

The $d$-GNP is a general framework that obtains the optimal solution to various constrained learning problems, including but not limited to multi-objective L2D problems. Using this post-processing framework, we can first estimate the scores related to our problem and then find a linear rule of these scores by fine-tuning for specific violation tolerances. This method reduces the computational complexity of in-processing methods while guaranteeing achieving a near-optimal solution in a large data regime.

---

[5]This is as opposed to the experiment in [44] where the human decision is simulated.

## 8 Acknowledgment

M.A. Charusaie thanks the International Max Planck Research School for Intelligent Systems (IMPRS-IS) and Tübingen AI Center for the support and funding of this project. He is further grateful to Matthäus Kleindessner for his significant intellectual contributions to the first draft of this paper. The idea of obtaining an extension to the Neyman-Pearson lemma emerged from discussions with André Cruz and Florian Dorner. The very initial draft of this paper was written during an hours-long train delay in Germany, and thus, M.A. Charusaie is thankful to Deutsche Bahn in that regard.

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

# Content of Appendices

## A   Lack of Compositionality of Fairness Criteria

Here, we show an example of lack of compositionality of fairness criteria for learn-to-defer problems. This falls in line with [29], where the authors studied the effect of the operators such as 'OR' or 'AND'. Here, we show that a similar non-compositionality holds for the operator 'DEFER'. The following example is found based on the insight that a fair predictor is fair over all the space $\mathcal{X}$, and if it could take a decision over only a subset of $\mathcal{X}$ it will not necessarily be a fair predictor. This can be seen as a particular application of Yule's effect [62] which explains that vanishing correlation in a mixture of distributions does not necessarily concludes vanishing correlation on each of such distributions.

Let us assume that the space $\mathcal{X}$ contains only four points $x_1$, $x_2$, $x_3$, and $x_4$, and that the input takes these values with probability $\Pr(X = x_1) = \Pr(X = x_2) = \Pr(X = x_3) = \Pr(X = x_4) = \frac{1}{4}$. The first two points $x_1, x_2$ are corresponded to the demographic group $A = 0$ and the last two points are corresponded to the demographic group $A = 1$. Further, assume that the conditional target probability is $\Pr(Y = 1|x_1) = \Pr(Y = 1|x_2) = \Pr(Y = 1|x_3) = \Pr(Y = 1|x_4) = 1$. Moreover, we consider the equality of opportunity as the measure of fairness. Now, assume that the classifier $h(\cdot) : \mathcal{X} \to \{0,1\}$ is taking values $h(x_1) = 1$, $h(x_2) = 0$, $h(x_3) = 1$, and $h(x_4) = 0$ and the human decision maker predicts $M = 0$ conditioned on $x_1$, $M = 1$ conditioned on $x_2$, and $M = 1$ conditioned on $x_3$, and $M = 0$ conditioned on $x_4$. Therefore, both classifier and human expert have accuracy of $\frac{1}{2}$ on the data.

Following the above assumptions, we can find the fairness measure for classifier as

$$
\begin{aligned}
&\Pr(h(X) = 1|Y = 1, A = 0) - \Pr(h(X) = 1|Y = 1, A = 1) \\
&= \Pr(h(X) = 1|Y = 1, A = 0, X = x_1) \Pr(X = x_1|Y = 1, A = 0) \\
&\quad + \Pr(h(X) = 1|Y = 1, A = 0, X = x_2) \Pr(X = x_2|Y = 1, A = 0) \\
&\quad - \Pr(h(X) = 1|Y = 1, A = 1, X = x_3) \Pr(X = x_3|Y = 1, A = 1) \\
&\quad - \Pr(h(X) = 1|Y = 1, A = 1, X = x_4) \Pr(X = x_4|Y = 1, A = 1) = \frac{1}{2} + 0 - \frac{1}{2} - 0 = 0,
\end{aligned}
\tag{10}
$$

which means that the classifier is fully fair. We can derive a similar result for the human expert, i.e.,

$$
\Pr(M = 1|Y = 1, A = 0) - \Pr(M = 1|Y = 1, A = 1) = 0.
\tag{11}
$$

Now that we established a fair classifier and a fair expert, we take the step to find an optimal deferral solution, i.e., a deferral system that minimizes the overall loss. We can observe that for $x_1$ the classifier is accurate, while for $x_2$ the human expert is accurate. Furthermore, for $x_3$ and $x_4$ they both are equally inaccurate. Therefore, an optimal solution is not to defer for $x_1$, and defer for $x_2$, and take an arbitrary decision for $x_3$ and $x_4$. Now, if we find the fairness measure of the resulting deferral predictor, we have

$$
\begin{aligned}
&\Pr(\hat{Y} = 1|Y = 1, A = 0) - \Pr(\hat{Y} = 1|Y = 1, A = 1) \\
&= \Pr(h(X) = 1|Y = 1, A = 0, X = x_1) \Pr(X = x_1|Y = 1, A = 0) \\
&\quad + \Pr(M = 1|Y = 1, A = 0, X = x_2) \Pr(X = x_2|Y = 1, A = 0) \\
&\quad - \Pr(h(X) = 1|Y = 1, A = 1, X = x_3) \Pr(X = x_3|Y = 1, A = 1) \\
&\quad - \Pr(h(X) = 1|Y = 1, A = 1, X = x_4) \Pr(X = x_4|Y = 1, A = 1) = \frac{1}{2} + \frac{1}{2} - \frac{1}{2} - 0 = \frac{1}{2},
\end{aligned}
\tag{12}
$$

or equivalently the resulting predictor is unfair for the demographic group $A = 1$. This means the 'DEFER' composition of the predictors does not preserve fairness. One can further easily show that no deferral system from the above classifier and human expert that has the accuracy better than $\frac{1}{2}$ is fair.

## B   Extended Related Works

The deferral problem has been studied under a variety of conditions. Rejection learning [19, 3, 11, 15] or selective classification [30, 33, 32], assumes that a fixed cost is incurred to the overall loss, when ML decides not to make a prediction on an input. The first Bayes optimal rule for rejection learning was derived in [16]. Assuming that the accuracy of human, and consequently the cost of deferring to

the human, can vary for different inputs, [50] obtained the Bayes optimal deferral rule. The deferral problem is further studied assuming that the number of available instances for deferral are bounded and a near-optimal classifier and deferral rule is required as a solution of empirical risk minimization [23, 24]. Most recently, the implementation of deferral rules using neural networks and surrogate losses is studied for binary and multi-class classification [8, 50, 12, 51, 9, 43, 48, 45]. A possible shift in human expert for L2D methods recently studied in [67]. The problem multi-objective L2D and rejection learning is mainly studied in an in-processing approach. A few instances of tackling such problems can be found in [57, 52, 53] and [74, 41] for L2D and rejection learning, respectively. Neyman-Pearson's fundamental lemma is introduced in [55] originally for binary hypothesis testing and later was generalized in [56] to give a close-form formulation for a variety of binary constrained optimization problems. Later, [21] found conditions for which Neyman and Pearson solution exists and is unique. The generalization error of the empirical solution to Neyman-Pearson problem is studied in two lines of works: (i) the generalization of direct (in-processing) solutions to the optimization problem [65, 64, 61], and (ii) the generalization of plug-in methods [71] that first approximate the score functions and then use Neyman-Pearson lemma to approximate the predictor. The generalization of Neyman-Pearson lemma to multiclass setting is first empirically studied in [40] and under strong duality assumption is proved in [69]. Our lemma $d$-GNP extends these works in order to (i) be able to control a general set of constraints instead of Type-$K$ errors, and (ii) be valid in absence of strong duality assumption. Further, the idea of using Neyman-Pearson lemma for controlling fairness criteria originally dates back to [76] (later as [75]). More recently, a similar post-processing method is introduced in [14] using cost-sensitive learning and strong duality technique. Although these works cover binary classification problem, in this paper we focus on solving multi-class classification problem, and particularly in a deferral system.

Moreover, his work differs from multi-class classification with complex performance metrics [54] in the sense that they consider constraints that are non-linear functions of confusion matrix, while ignoring the dependence on input $x$. In our setting, the constraints are linear in terms of confusion matrix when conditioned on the input, but the linear coefficients vary with the input.

Finally, the work [70] has recently studied an extension of post-processing method to other constrained learning problems. The difference of that work with our method is threefold: (i) while we prove that the optimal post-processing method is a linear combination of scores, they have no such claim, (ii) we have no assumption on the format of the loss function, while they assume a particular set of strictly convex loss functions, (iii) we have no bound on our hypothesis class while they assume the representation of the predictor with a multidimensional vector and a fixed dimension.

## C  Rephrasing (2) into Linear Functional Programming

Here, we first characterize functions that are outcome-dependent. To that end, we define $\imath(x)$ as

$$\imath = \big[\mathbb{I}_{r(x)=0}\mathbb{I}_{h(x)=1}, \ldots, \mathbb{I}_{r(x)=0}\mathbb{I}_{h(x)=L}, \mathbb{I}_{r(x)=1}\big]. \tag{13}$$

This function can retrieve the value of $r(x)$ and can retrieve the value of $h(x)$ only if $r(x) = 0$. In fact, we can obtain $r(x) = \big(\imath(x)\big)(L+1)$ and $h(x) = i$ if $r(x) = 0$ and $\big(\imath(x)\big)(i) = 1$. Therefore, for a function $\overline{\Psi}(x, h(x), r(x)) = \mathbb{E}_{Y,M|X=x}\big[\Psi(x, Y, M, h(x), r(x))\big]$ and $\overline{\ell}_{\text{def}}(x, h(x), r(x)) = \mathbb{E}_{Y,M|X=x}\big[\ell_{\text{def}}(x, Y, M, h(x), r(x))\big]$ to be outcome dependent, it must only be a function of $x$ and $\imath(x)$. In fact, we must have

$$\overline{\Psi}_i(x, h(x), r(x)) = \Psi'_i(x, \imath(x)), \tag{14}$$

and

$$\overline{\ell}_{\text{def}}(x, h(x), r(x)) = \ell'_{\text{def}}(x, \imath(x)), \tag{15}$$

for a choice of $\Psi'$ and $\ell'_{\text{def}}$, where $\overline{\ell}_{\text{def}}(x, h(x), r(x)) = \mathbb{E}_{Y,M|X=x}\big[\ell_{\text{def}}(x, Y, M, h(x), r(x))\big]$. Now, we can check that $\imath(x)$ can take $L+1$ different values, in each of which one of its components takes the value 1 and others take the value 0. Therefore, by conditioning on each of these $L+1$ values we have

$$\Psi'_i(x, \imath(x)) = \sum_{i=1}^{L+1} \Psi'(x, [0, \ldots, \underbrace{1}_{i}, \ldots, 0])\Big(\big(\imath(x)\big)(i)\Big) = \langle \imath(x), \psi_i(x)\rangle, \tag{16}$$

where $\psi_i(x)$ is defined as

$$\psi_i(x) = \Big[\Psi'_i(x, [1, 0, \ldots, 0]), \ldots, \Psi'(x, [0, 0, \ldots, 1])\Big]$$
$$= \big[\overline{\Psi}_i(x, 1, 0), \ldots, \overline{\Psi}_i(x, L, 0), \overline{\Psi}_i(x, 0, 1)\big]. \tag{17}$$

Similarly, we can show that

$$\ell'_{\text{def}}(x, \imath(x)) = \langle \imath(x), \vec{\ell}_{\text{def}}(x)\rangle, \tag{18}$$

where $\vec{\ell}_{\text{def}}(x)$ is defined as

$$\vec{\ell}_{\text{def}}(x) = \big[\overline{\ell}_{\text{def}}(x, 1, 0), \ldots, \overline{\ell}_{\text{def}}(x, L, 0), \overline{\ell}_{\text{def}}(x, 0, 1)\big]. \tag{19}$$

Next, we know that due to the randomization of $\mathcal{A}$, the vector $\imath(x)$ can take various values on each instance $x$. This, however, is not the case for $\psi_i(x)$ and $\vec{\ell}_{\text{def}}(x)$, since they are defined independent of $r(x)$ and $h(x)$. Therefore, the average of constraints and loss can be rewritten as

$$\mathbb{E}_{(r,h)\sim\mathcal{A}}\big[\overline{\Psi}_i(x, h(x), r(x))\big] = \mathbb{E}_{(r,h)\sim\mathcal{A}}\big[\langle\psi_i(x), \imath(x)\rangle\big] = \langle f(x), \psi_i(x)\rangle, \tag{20}$$

and

$$\mathbb{E}_{(r,h)\sim\mathcal{A}}\big[\overline{\ell}_{\text{def}}(x, h(x), r(x))\big] = \mathbb{E}_{(r,h)\sim\mathcal{A}}\big[\langle\vec{\ell}_{\text{def}}(x), \imath(x)\rangle\big] = \langle f(x), \vec{\ell}_{\text{def}}(x)\rangle, \tag{21}$$

where $f(x)$ is defined as

$$f(x) = \mathbb{E}[\imath(x)] = \big[\Pr(r(x) = 0, h(x) = 1), \ldots, \Pr(r(x) = 0, h(x) = L), \Pr(r(x) = 1)\big]. \tag{22}$$

Therefore, the optimization problem in (2) is effectively reduced to the linear programming problem in (3). Moreover, if $f^*(x)$ is the solution to that linear program, then the corresponding $r(x)$ should be distributed as $\Pr(r(x) = 1) = \big(f^*(x)\big)(L + 1)$, where $h(x)$ should be distributed as $\Pr(h(x) = i) = \Pr(h(x) = i | r(x) = 0) = \dfrac{\big(f(x)\big)(i)}{\sum_{i=1}^{L}\big(f(x)\big)(j)}$. Note that the assumption of independence of $h(x)$ and $r(x)$ comes with no loss of generality, since the value of $h(x)$ does not variate the loss or constraints in the system when we have $r(x) = 1$.

## D    Derivation of Embedding Functions

In this appendix we derive the embedding functions in Table 1 that are corresponded to the constraints of choice, as named in Section 3. The trick that we use for all these constraints is that we first rewrite the constraint in terms of the expected value of a function over the randomness of the algorithm $\mathcal{A}$ and the input variable $X$, and then we use (17) to transform that function into the embedding function.

- **Overall Loss**: To find the embedding function that is corresponded to the overall loss of the system, we should first note that by loss we mean the probability of incorrectness of $\hat{Y}$. Therefore, the corresponding $\ell_{def}(x, h(x), r(x))$ in this case, as defined in (1) is obtained as

$$\overline{\ell}_{def}(x, h(x), r(x)) = \mathbb{E}_{Y,M|X=x}\big[\mathbb{I}_{r(x)=1}\mathbb{I}_{M\neq Y} + \mathbb{I}_{r(x)=0}\mathbb{I}_{h(x)\neq Y}\big]$$
$$= \mathbb{I}_{r(x)=1}\Pr(M = Y | X = x) + \mathbb{I}_{r(x)=0}\Pr(Y \neq h(x) | X = x).$$

Therefore, using (19) we find $\vec{\ell}_{\text{def}}$ as

$$\vec{\ell}_{\text{def}} = \big[\Pr(Y \neq 1 | X = x), \ldots, \Pr(Y \neq n | X = x), \Pr(Y \neq M | X = x)\big].$$

- **Expert intervention budget**: In this case, similar to the case before, we first derive $\overline{\Psi}(x, h(x), r(x))$. To that end, we first note that the expert intervention constraint in Section 3 is equivalent with

$$\Pr(r(X) = 1) = \mathbb{E}_{x\sim\mu_X,(r,h)\sim\mathcal{A}}\big[\mathbb{I}_{r(x)=1}\big] \leq \delta,$$

which in turn suggests that

$$\overline{\Psi}(x, h(x), r(x)) = \mathbb{I}_{r(x)=1}.$$

Next, we find $\psi(x)$ using (17), as

$$\psi(x) = [0, \ldots, 0, 1].$$

- **OOD Detection**: To obtain the corresponding embedding function to the OOD detection constraint in Section 3, we can rewrite $\Pr_{\text{out}}\big(r(X)=1\big)$ as

$$\Pr_{\text{out}}\big(r(X)=1\big) = \mathbb{E}_{X\sim f_X^{\text{out}},(r,h)\sim\mathcal{A}}\big[\mathbb{I}_{r(X)=1}\big] = \mathbb{E}_{X\sim\mu_{X^{\text{in}}},(r,h)\sim\mathcal{A}}\Big[\tfrac{\mathbb{I}_{r(X)=1}f_X^{\text{out}}(X)}{f_X^{\text{in}}(X)}\Big],$$

where the last equation holds when $X$ and $X_{\text{out}}$ are absolutely continuous distributions, and therefore have probability density functions. A similar assumption is made by [53]. This results in $\overline{\Psi}(x,h(x),r(x))$ being obtained as

$$\overline{\Psi}(x,h(x),r(x)) = \frac{\mathbb{I}_{r(x)=1}f_X^{\text{out}}(X)}{f_X^{\text{in}}(X)}.$$

Therefore, we conclude that the embedding function can be calculated using (17) as

$$\psi(x) = \big[0,\ldots,0,\tfrac{f_X^{\text{out}}(X)}{f_X^{\text{in}}(X)}\big].$$

In the simple case that $f_X^{\text{out}}(x) = \frac{f_X^{\text{in}}(x)\mathbb{I}_{f_X^{\text{in}}(x)\leq\epsilon}}{\int f_X^{\text{in}}(x)\mathbb{I}_{f_X^{\text{in}}(x)\leq\epsilon}dx}$, the embedding function is equal to

$$\psi(x) = \big[0,\ldots,0,\tfrac{\mathbb{I}_{f_X^{\text{in}}(x)\leq\epsilon}}{\Pr_{\text{in}}(f_X^{\text{in}}(X)\leq\epsilon)}\big].$$

- **Long-Tail Classification**: This methodology aims to minimize the balanced loss

$$\frac{1}{K}\sum_{i=1}^{K}\Pr(Y\neq h(X)|r(X)=0,Y\in G_i).$$

However, as mentioned in [52], this optimization problem can be rewritten as

$$\sum_{i=1}^{K}\frac{\Pr(Y\neq h(X),r(X)=0|Y\in G_i)}{\alpha_i}, \quad\text{s.t.}\quad \Pr(r(X)=0|Y\in G_i)=\frac{\alpha_i}{K}.$$

Therefore, the objective can rewritten as

$$\sum_{i=1}^{K}\frac{\mathbb{E}_{(r,h)\sim\mathcal{A},X'\sim\mu_X}\big[\Pr(Y\neq h(X),r(X)=0,Y\in G_i|X=X')\big]}{\alpha_i\Pr(Y\in G_i)},$$

which together with (17) shows that

$$\psi_0(x) = -\Big[\sum_{i=1}^{K}\frac{\Pr(Y\neq 1,Y\in G_i|X=x)}{\alpha_i\Pr(Y\in G_i)},\ldots,\sum_{i=1}^{K}\frac{\Pr(Y\neq L,Y\in G_i|X=x)}{\alpha_i\Pr(Y\in G_i)},0\Big].$$

The reason that we use negative sign is because in the definition of (3) we aim to maximize the objective.

Similarly, we can rewrite the objectives as

$$\frac{\mathbb{E}_{(r,h)\sim\mathcal{A},X'\sim\mu_X}\big[\Pr(r(X)=0,Y\in G_i|X=X')-\frac{\alpha_i}{K}\Pr(Y\in G_i)\big]}{\Pr(Y\in G_i)}.$$

Therefore, using (17) we can obtain $\psi_i(x)$ as

$$\psi_i(x) = \frac{\Pr(Y\in G_i|X=x)}{\Pr(Y\in G_i)}\big[1,\ldots,1,0\big]-\frac{\alpha_i}{K}. \tag{23}$$

- **Type-$k$ Error Bound**: We first rewrite Type-$k$ constraint in 3 as

$$\Pr(\hat{Y}\neq k|Y=k) = \frac{\Pr(\hat{Y}\neq k,Y=k)}{\Pr(Y=k)}$$

$$\stackrel{(a)}{=} \frac{\mathbb{E}_{X\sim\mu_X}\big[\Pr(\hat{Y}\neq k,Y=k|X=x)\big]}{\Pr(Y=k)}$$

$$= \frac{\mathbb{E}_{X\sim\mu_X}\big[\Pr(\hat{Y}\neq k|Y=k,X=x)\Pr(Y=k|X=x)\big]}{\Pr(Y=k)}, \tag{24}$$

where $(a)$ is followed by chain rule.

Next, we condition $\Pr(\hat{Y} \neq k|Y = k, X = x)$ on $r(X)$ being 1 and 0, which concludes that

$$
\begin{aligned}
\Pr(\hat{Y} \neq k|Y = k, X = x) &= \Pr(\hat{Y} \neq k, r(x) = 1|Y = k, X = x) \\
&\quad + \Pr(\hat{Y} \neq k, r(x) = 0|Y = k, X = x) \\
&= \Pr(M \neq k, r(x) = 1|Y = k, X = x) \\
&\quad + \Pr(h(x) \neq k, r(x) = 0|Y = k, X = x) \\
&= \mathbb{E}_{(r,h) \sim \mathcal{A}, M|X=x, Y=k}\left[\mathbb{I}_{M \neq k}\mathbb{I}_{r(x)=1} + \mathbb{I}_{h(x) \neq k}\mathbb{I}_{r(x)=0}\right] \\
&= \mathbb{E}_{(r,h) \sim \mathcal{A}|X=x, Y=k}\left[\Pr(M \neq k|X = x, Y = k)\mathbb{I}_{r(x)=1} \right. \\
&\qquad\qquad\qquad\qquad \left. + \mathbb{I}_{h(x) \neq k}\mathbb{I}_{r(x)=0}\right].
\end{aligned}
$$

Therefore, using (24) we conclude that

$$
\begin{aligned}
\Pr(\hat{Y} \neq k|Y = k) ={}& \frac{\mathbb{E}_{X' \sim \mu_X, (r,h) \sim \mathcal{A}}\left[\mathbb{I}_{r(X)=1}\Pr(M \neq k, Y = k|X = X')\right]}{\Pr(Y = k)} \\
&+ \frac{\mathbb{E}_{X' \sim \mu_X, (r,h) \sim \mathcal{A}}\left[\mathbb{I}_{h(X') \neq k}\mathbb{I}_{r(X')=0}\Pr(Y = k|X = X')\right]}{\Pr(Y = k)},
\end{aligned}
$$

which together with (17) shows that the embedding function is obtained as

$$
\psi(x) = \frac{\Pr(Y = k|X = x)}{\Pr(Y = k)}\left[1, \dots, 1, \underbrace{0}_{k}, 1, \dots, 1, \Pr(M \neq k|X = x, Y = k)\right].
$$

Note that here we used the assumption that $(Y, M)$ and $\mathcal{A}$ are independent for each choice of $X$, i.e., the value noise that is introduced in $\mathcal{A}$ for each $X = x$ is generated independent of the value of $Y$ and $M$, which is the true assumption, since the algorithm only has access to $X$ and not true label or the human label.

- **Demographic Parity**: We know that the demographic parity constraint in Section 3 can be written as
$$
-\delta \leq \Pr(\hat{Y} = 1|A = 0) - \Pr(\hat{Y} = 1|A = 1) \leq \delta. \tag{25}
$$
Here, we find the corresponding embedding function $\psi(x)$ for the upper-bound in the above inequality. For the lower-bound, we can use $-\psi(x)$ and follow the steps that are proposed in the main text of the manuscript.

To find the embedding function that corresponds to the upper-bound of (25), we first rewrite $\Pr(\hat{Y} = 1|A = 0) - \Pr(\hat{Y} = 1|A = 1)$ as

$$
\Pr(\hat{Y} = 1|A = 0) - \Pr(\hat{Y} = 1|A = 1) = \frac{\Pr(\hat{Y} = 1, A = 0)}{\Pr(A = 0)} - \frac{\Pr(\hat{Y} = 1, A = 1)}{\Pr(A = 1)}. \tag{26}
$$

Now, similar to what we did in previous section, we condition $\Pr(\hat{Y} = 1, A = a)$ for $a \in \{0, 1\}$ on the value of $h(x)$ and $r(x)$, and we conclude

$$
\begin{aligned}
\Pr(\hat{Y} = 1, A = a) &= \Pr(\hat{Y} = 1, A = a, r(X) = 1) + \Pr(\hat{Y} = 1, A = a, r(X) = 0) \\
&= \Pr(M = 1, A = a, r(X) = 1) + \Pr(h(X) = 1, A = a, r(X) = 0) \\
&= \mathbb{E}_{X, A, M, \mathcal{A}}\left[\mathbb{I}_{M=1}\mathbb{I}_{A=a}\mathbb{I}_{r(X)=1} + \mathbb{I}_{h(X)=1}\mathbb{I}_{A=a}\mathbb{I}_{r(X)=0}\right] \\
&= \mathbb{E}_{X, \mathcal{A}}\left[\Pr(M = 1, A = a|X = x)\mathbb{I}_{r(X)=1} \right. \\
&\qquad\qquad \left. + \Pr(A = a|X = x)\mathbb{I}_{h(X)=1}\mathbb{I}_{r(X)=0}\right]. \tag{27}
\end{aligned}
$$

Here, we used the assumption of independence of $X$ and $(M, Y)$ given a choice of $X$. As a result of (26), (27), and (17) we can find the embedding function as

$$
\begin{aligned}
\psi(x) = \Big[& 0, \frac{\Pr(A = 1|X = x)}{\Pr(A = 1)} - \frac{\Pr(A = 0|X = x)}{\Pr(A = 0)}, \\
& \frac{\Pr(M = 1, A = 1|X = x)}{\Pr(A = 1)} - \frac{\Pr(M = 1, A = 0|X = x)}{\Pr(A = 0)}\Big].
\end{aligned}
$$

- **(In-)Equality of Opportunity**: Similar to the previous items, we rewrite equality of opportunity constraint in Section 3 as

$$-\delta \leq \Pr(\hat{Y} = 1 | Y = 1, A = 1) - \Pr(\hat{Y} = 1 | Y = 1, A = 0) \leq \delta.$$

Again, we only consider the upper-bound and rewrite $\Pr(\hat{Y} = 1 | Y = 1, A = 1) - \Pr(\hat{Y} = 1 | Y = 1, A = 0)$ as

$$\Pr(\hat{Y} = 1 | Y = 1, A = 1) - \Pr(\hat{Y} = 1 | Y = 1, A = 0)$$
$$= \frac{\Pr(\hat{Y} = 1, Y = 1, A = 1)}{\Pr(Y = 1, A = 1)} - \frac{\Pr(\hat{Y} = 1, Y = 1, A = 0)}{\Pr(Y = 1, A = 0)}. \quad (28)$$

Next, by conditioning on $r(X) = 1$ and $r(X) = 0$, we rewrite $\Pr(\hat{Y} = 1, Y = 1, A = a)$ for $a \in \{0, 1\}$ as

$$\Pr(\hat{Y} = 1, Y = 1, A = a) = \Pr(\hat{Y} = 1, Y = 1, A = a, r(X) = 1)$$
$$+ \Pr(\hat{Y} = 1, Y = 1, A = a, r(X) = 0)$$
$$= \Pr(M = 1, Y = 1, A = a, r(X) = 1)$$
$$+ \Pr(h(X) = 1, Y = 1, A = a, r(X) = 0)$$
$$= \mathbb{E}_{X,Y,M,A,\mathcal{A}} \big[ \mathbb{I}_{M=1} \mathbb{I}_{Y=1} \mathbb{I}_{A=a} \mathbb{I}_{r(X)=1}$$
$$+ \mathbb{I}_{h(X)=1} \mathbb{I}_{Y=1} \mathbb{I}_{A=a} \mathbb{I}_{r(X)=0} \big]$$
$$= \mathbb{E}_{X,\mathcal{A}} \big[ \mathbb{I}_{r(X)=1} \Pr(M = 1, Y = 1, A = a | X = x)$$
$$+ \mathbb{I}_{h(X)=1} \mathbb{I}_{r(X)=0} \Pr(Y = 1, A = a | X = x) \big], \quad (29)$$

where the last identity is followed by the assumption of independence of $\mathcal{A}$ and $(Y, M, A)$ given an instance $X = x$.

As a result of (28), (29), and (17) we can obtain the embedding function as

$$\psi(x) = \Big[ 0, \frac{\Pr(Y = 1, A = 1 | X = x)}{\Pr(Y = 1, A = 1)} - \frac{\Pr(Y = 1, A = 0 | X = x)}{\Pr(Y = 1, A = 0)}$$
$$\frac{\Pr(M = 1, Y = 1, A = 1 | X = x)}{\Pr(Y = 1, A = 1)} - \frac{\Pr(M = 1, Y = 1, A = 0 | X = x)}{\Pr(Y = 1, A = 0)} \Big].$$

- **(In-)Equality of Odds**: This induces the same constraint as that of equality of opportunity, and further induces an extra constraint that is in nature similar to equality of opportunity with the difference that it uses $Y = 0$ instead of $Y = 1$. Therefore, we have two embedding functions, one is similar to that of equality of opportunity as

$$\psi_1(x) = \Big[ 0, \frac{\Pr(Y = 1, A = 1 | X = x)}{\Pr(Y = 1, A = 1)} - \frac{\Pr(Y = 1, A = 0 | X = x)}{\Pr(Y = 1, A = 0)}$$
$$\frac{\Pr(M = 1, Y = 1, A = 1 | X = x)}{\Pr(Y = 1, A = 1)} - \frac{\Pr(M = 1, Y = 1, A = 0 | X = x)}{\Pr(Y = 1, A = 0)} \Big],$$

and another similar to that with changing $Y = 1$ into $Y = 0$, and therefore as

$$\psi_2(x) = \Big[ \frac{\Pr(Y = 0, A = 1 | X = x)}{\Pr(Y = 0, A = 1)} - \frac{\Pr(Y = 0, A = 0 | X = x)}{\Pr(Y = 0, A = 0)}, 0$$
$$\frac{\Pr(M = 1, Y = 0, A = 1 | X = x)}{\Pr(Y = 0, A = 1)} - \frac{\Pr(M = 1, Y = 0, A = 0 | X = x)}{\Pr(Y = 0, A = 0)} \Big].$$

## E   Limitations of Cost-Sentitive Methods

A variety of works have tackled constrained classification problems using cost-sensitive modeling [42, 17, 57]. In other words, they use the expected loss that is penalized with the constraints and solve that for certain coefficients for those constraints (a.k.a., they form Lagrangian from that problem). In the next step, they optimize the coefficients and obtain the optimal predictor. The issue that we discuss further in the following we concern is that during this process, the optimal predictor is achieved only

when the corresponding cost-sensitive Lagrangian has a single saddle point in terms of coefficients and predictors. Such assumption, unless by analyzing the Lagrangian closely, is hard to be validated. However, our results in this paper have no such assumption, and instead use statistical hypothesis testing methods to show their optimality.

To further clarify the issue with such methodology, we bring an example of L2D problem when human intervention budget is controlled. Suppose that the features in $\mathcal{X}$ are distributed with an atomless probability measure $\mu_X$ (e.g., normal or uniform distribution).[6] Further, assume that the human has perfect information of the label, i.e. $Y = M$, while the input features have no information of the label, i.e., $\Pr(Y = 1|X = x) = 1/2$ for all $x \in \mathcal{X}$. Moreover, let the classifier and the human induce the $0 - 1$ loss function. In this case, we can see that the optimal classifier is the maximizer of the scores (see the early discussion of Section H), which in this case, since there is no clear maximizer, without loss of generality can be set to $h(x) \equiv 1$.

For such assumptions, if we write the Lagrangian in form of

$$L(\lambda, r) = L_{\text{def}}^\mu(h, r) + \lambda(\mathbb{E}[r(X)] - b) = \frac{1}{2} - \frac{1}{2}\mathbb{E}\big[r(X)\big] + \lambda(\mathbb{E}[r(X)] - b),$$

then strong duality shows that

$$\min_{r \in [0,1]^{\mathcal{X}}} \max_{\lambda \geq 0} L(\lambda, r) = \max_{\lambda \geq 0} \min_{r \in [0,1]^{\mathcal{X}}} L(\lambda, r), \tag{30}$$

or to put it informally, the objective is invariant under the interchange of minimum and maximum over Lagrange multipliers and the variable of interest. However, this does not prove the interchangeability of the saddle points in these settings, i.e., we cannot guarantee $\operatorname{argmin}_{r \in [0,1]} L(\lambda^*, r) = f^*$, where $\lambda^* \in \operatorname{argmax}_\lambda \min_{r \in [0,1]} L(\lambda, r)$, and $f^* \in \operatorname{argmin}_{r \in [0,1]} \max_\lambda L(\lambda, r)$. In fact, this guarantee holds only in particular examples, e.g., when $L(\lambda_r^*, r)$ is strictly convex [7, page 8].

In fact, if we optimize $r$ for all $\lambda$ as in RHS of (30), we can show that $r_\lambda(x) = \begin{cases} 1 & \lambda < \frac{1}{2} \\ 0 & \text{otherwise} \end{cases}$.

Therefore, $\lambda^*$ can be obtained as $\lambda^* = \operatorname*{argmax}_{\lambda \geq 0}(\lambda - 1/2)^- - \lambda b$ where $(x)^- := \min\{x, 0\}$. This can be rewritten as

$$\lambda^* = \operatorname*{argmax}_{\lambda \geq 0} \begin{cases} -\frac{1}{2} - \lambda(b-1) & 0 \leq \lambda \leq \frac{1}{2} \\ -\lambda b & \lambda > \frac{1}{2} \end{cases} = \frac{1}{2}.$$

Hence, the condition $\lambda < 1/2$ is never satisfied and we have $r_{\lambda^*}(x) = 0$, i.e., we should never defer. For the deferral rule $r_{\lambda^*}$, the deferral loss (1) is

$$L_{\text{def}}^\mu(h, \hat{f}) = \mathbb{E}_{X,Y,M}\big[\ell_{AI}(Y, h(X), X)\big] = \frac{1}{2}.$$

To show that $r_{\lambda^*}$ is not optimal, we provide random and deterministic deferral rules $f^*$ and $r^{**}$ that satisfy the constraint in (2), while having a smaller deferral loss:

◇ Let $f^*(x) = b$, that is a random deferral rule that defers with probability $b$ everywhere on $\mathcal{X}$. Therefore, on average $b$ proportion of inquiries are deferred and thus it satisfies the constraint in (2). The deferral loss for $f^*(x)$ is equal to

$$L_{\text{def}}^\mu(h, f^*) = \underbrace{\mathbb{E}[r(X)]}_{b} \cdot \underbrace{\mathbb{E}[\ell_H(Y, M)]}_{0}$$
$$+ \underbrace{\mathbb{E}[1 - r(X)]}_{1-b} \cdot \underbrace{\mathbb{E}[\ell_{AI}(Y, h(X))]}_{\frac{1}{2}}$$
$$= \frac{1-b}{2} < \frac{1}{2}.$$

◇ The second example is a deterministic deferral rule. Since the probability measure on $\mathcal{X}$ is atomless, for all $b \in [0, 1]$ there exists a set $\mathcal{A}$ such that $\Pr(X \in \mathcal{A}) = b$ [31, Proposition 215D]. Hence, defining $r^{**}(x) = \mathbb{1}_{x \in \mathcal{A}}$ the constraint in (2) is met. Similar to the last example $L_{\text{def}}^\mu(h, r^{**}) = \frac{1-b}{2} < \frac{1}{2}$.

The above two examples show that the deferral rule $r_{\lambda^*}$ is sub-optimal. The reason is that, for optimality of $r_{\lambda^*}$ we should make sure that $L(\lambda_r^*, r)$ has a single minimizer of $r$. However, in our example we had $L(\frac{1}{2}, r) = -\lambda b$ has infinite number of minimizers in terms of $r(x)$. Therefore, the equality of the solutions to minimax problem and maximin problem is not guaranteed.

---

[6]If we have a probability measure that contains atoms, one can follow the same steps for the first counterexample.

## F   $d$-GNP Learning Algorithm

---

**Algorithm 1** Finding Optimal Classifier and Rejection Function

---

**Require:** The formulation of $\ell_{\text{def}}(\cdot,\cdot,\cdot)$ and $\{\Psi_i(\cdot,\cdot,\cdot)\}_{i=1}^m$, and the datasets $\mathcal{D}_{\text{train}} = \{(x^i, a^i, m^i, y^i)\}_{i=1}^{n_{\text{train}}}$, $\mathcal{D}_{\text{val}} = \{(x^i, a^i, m^i, y^i)\}_{i=n_{\text{train}}+1}^{n_{\text{train}}+n_{\text{val}}}$, and tolerances $\{\delta_i\}_{i=1}^m$

**Ensure:** Optimal deferral rule $r^*(x)$ and classifier $h^*(x)$

1: **Parameters:** $\epsilon = 1e - 8$
2: **procedure** CONSTRAINEDDEFER($\ell_{\text{def}}, \{\Psi_i\}_{i=1}^m, \mathcal{D}_{\text{train}}, \mathcal{D}_{\text{val}}$)
3:     Obtain closed-form of $\{\psi_i(x)\}_{i=0}^m$ using $\ell_{\text{def}}$ and $\Psi_i$s via (4) and in terms of the scores as in Table 1
4:     Estimate the scores in Table 1 using classification/regression methods on $\mathcal{D}_{\text{train}}$
5:     Find estimate $\{\hat{\psi}_i\}_{i=0}^m$ using estimated scores in previous step and closed-form of Step 3
6:     **if** $m = 2$ **then**
7:         Define routine $\hat{f}_{k,p}(x) := \tau\big(\hat{\psi}_0(x) - k\hat{\psi}_1(x), x\big)$ for $\tau$ in Theorem 4.2
8:         Define routine $\hat{C}(t) := \widehat{\mathbb{E}}_{\mathcal{D}_{\text{val}}}\Big[\langle \hat{f}_{k,0}(x_i), \hat{\psi}_1(x_i)\rangle\Big]$
9:         Find $\hat{k} = \min k$ over the feasibility set $\hat{C}(t) \leq \delta_1$
10:         **if** $\hat{k} = \emptyset$ **then**
11:             **Return** 'Not Feasible'
12:         **else**
13:             **if** $\hat{C}(\hat{k} - \epsilon) - \hat{C}(k^*) \leq 1e - 3$ **then**
14:                 $\hat{p} \leftarrow 0$
15:             **else**
16:                 $\hat{p} \leftarrow \frac{\delta - \hat{C}(\hat{k})}{\hat{C}(\hat{k}-\epsilon) - \hat{C}(\hat{k})}$
17:             **end if**
18:         **end if**
19:         $s(x) := \hat{f}_{\hat{k},\hat{p}}(x)$
20:     **else**
21:         Optimize (3) for $\mathcal{D}_{\text{val}}$ and for $f(x) = \tau(\hat{\psi}_0(x) - \sum_{i=1}^m \hat{\psi}_i(x), x)$ for $\tau$ as in Theorem 4.1 and via exhaustive search over $\{k_1, \ldots, k_m\}$ and randomizations of $\tau$ and find $s(x) := \hat{f}(x)$
22:     **end if**
23:     $h^*(x) := \underset{i\in[0:L-1]}{\operatorname{argmax}} s_i(x)$
24:     $r^*(x) := \underset{i\in\{0,1\}}{\operatorname{argmax}} \big[s_{h^*(x)}(x), s_L(x)\big]$
25:     **Return** $h^*(x), r^*(x)$
26: **end procedure**

---

## G   On Failure of In-Processing Methods

One might think that the need of using post-processing methods does not necessarily appear in some examples of multi-objective L2D problem. As an instance, for the expert intervention budget we can rank samples based on the difference between machine and human loss and defer the top $b$-proportion of samples for which the machine loss is higher than the human one. This method is illustrated in Algorithm 2. Indeed, in the following we show that the sub-optimality of such deterministic deferral rule, compared to the optimal deferral rule diminishes as the size of training set increases.

**Theorem G.1** (Optimal Deferral for Empirical Distribution). *For a classifier $h(x)$ and dataset $\mathcal{D} = \{(x_i, y_i, m_i)\}_{i=1}^n$, where we assume $x_i \neq x_j$, $i \neq j$, the deterministic deferral rule as in Algorithm 2 is (i) the optimal deterministic deferral rule for the empirical distribution on $\mathcal{D}$ and bounded expert intervention budget, and (ii) at most $\frac{1}{n}$-suboptimal (in terms of deferral loss) compared to the optimal random deferral rule for the empirical distribution on $\mathcal{D}$.*

Next in the following, we show that such policy does not provide sufficient information to determine the optimal deferral rule for the true distribution. To that end, we first recall that in classification tasks, the optimal classifier typically thresholds an estimation of conditional probability of the label $Y$ given $X$ that is obtained using the available dataset. As a result, if we observe enough pairs of $(x_i, y_i)$, then

**Algorithm 2** Deterministic Algorithm for Deferring Tasks to Human or AI for the Empirical Distribution and Expert Intervention Budget

---

**Input**: The dataset $\mathcal{D}$, he human and classifier loss $\ell_H$ and $\ell_{AI}$ and available proportion $b$ of instances to defer

**Output**: Labels of "defer" or "no defer" for each instance in $\mathcal{D}$

1: **procedure** DEFERTASKS($\mathcal{D}, \ell_H, \ell_{AI}, b$)
2:     Make the set $A = \{(x, y, m) \in \mathcal{D} : \ell_H(y, m) - \ell_{AI}(y, h(x)) \leq 0\}$
3:     **if** $|A| \geq b|\mathcal{D}|$ **then**
4:         Defer all tasks in $A$ to human
5:     **else**
6:         Defer the $b|\mathcal{D}|$ tasks with the lowest $\ell_H(x, y, m) - \ell_{AI}(x, y)$ to human
7:     **end if**
8: **end procedure**

---

we improve upon such estimation of conditional probability and increase the accuracy of the obtained classifier. However, we argue that this paradigm is inapplicable in the case of deferral rule as follows. Although the output $\hat{r}$ of Algorithm 2 for each feature $x$ is a deterministic 0 or 1 label, it varies with the choice of the dataset $\mathcal{D}$. Hence, if we draw datasets from a distribution $\mu$, the outcome of $\hat{r}$ becomes probabilistic. In the following, we introduce two probability distributions $\mu_1$ and $\mu_2$ over $(X, Y, M)$ such that for random draws of the dataset from $\mu_i$, the conditional probability of such optimal deferral label $\hat{r}$ given $X$ is equal, yet the optimal deferral rule for the true distribution is different.

Although the following discussion bears some resemblance with the No-Free-Lunch theorem [e.g. 66], there exists the following difference between the two. The No-Free-Lunch theorem states that for each learning algorithm, there exists a data distribution on which the algorithm does not generalize well. However, in the following discussion, we assume that we can observe infinite number of datasets and indeed, we can find the underlying probability of the deferral labels. In fact, we show that the limiting factor for finding the optimal deferral for the true distribution is that we only use deferral labels and if we use values of both losses, we can accordingly find the optimal deferral rule as suggested in Theorem 4.1.

Assume that we have a dataset $\mathcal{D} = \{(x_i, y_i, m_i)\}_{i=1}^n$ that contains i.i.d. samples from the distribution $\mu_{XYM}$. Further, assume that we have no budget constraint, that is $b = 1$ in Algorithm 2. Therefore, the optimal randomized deferral rule over the empirical distribution is the solution of the problem

$$\min_{\hat{r}_i \in [0,1]} \sum_{i=1}^n \mathbb{1}_{m_i \neq y_i} \hat{r}_i + \mathbb{1}_{h(x_i) \neq y_i} (1 - \hat{r}_i).$$

It is easy to see that the solution to this problem is given by $\hat{r}_i = 0$ if $\mathbb{1}_{h(x_i) \neq y_i} < \mathbb{1}_{m_i \neq y_i}$ and $\hat{r}_i = 1$ if $\mathbb{1}_{h(x_i) \neq y_i} > \mathbb{1}_{m_i \neq y_i}$. As a result, the optimal deferral is obtained as

$$\hat{r}_i = \begin{cases} 1 & m_i = y_i,\ h(x_i) \neq y_i \\ 0 & m_i \neq y_i,\ h(x_i) = y_i \\ \text{any value in } [0, 1] & o.w. \end{cases} \tag{31}$$

Among all the possible policies, we can choose

$$\hat{r}_i = \begin{cases} 1 & m_i = y_i\ \&\ h(x_i) \neq y_i \\ 0 & o.w. \end{cases}.$$

Next, we assume that we have a classifier $h$ and two probability distributions $\mu_1$ and $\mu_2$ over $(X, Y, M)$. For both distributions $X$ is uniformly distributed over $[0, 1]$, and we have $\mu_1(Y = M, h(X) = Y) = \frac{2}{3}, \mu_1(Y = M, h(X) \neq Y) = \frac{1}{3}$ and $\mu_2(Y \neq M, h(X) = Y) = \frac{2}{3}, \mu_2(Y = M, h(X) \neq Y) = \frac{1}{3}$. We can see that although the observed $\hat{r}$s are fixed for a given choice of $\mathcal{D}$, since $\mathcal{D}$ is randomly drawn, $\hat{r}$ values are randomly distributed. Furthermore, the distribution of $\Pr(\hat{r}|X)$ is according to $Bern(\frac{1}{3})$, since in both cases we have $\mu_i(Y = M, h(X) \neq Y) = \frac{1}{3}$. However, the optimal deferral rule for the first distribution is $r_1^*(x) = 1$ for all $x \in \mathcal{X}$, since we have $L_{\text{def}}^{\mu_1}(h, r_1^*) = 0$, while for the second case the optimal deferral rule is $r_2^*(x) = 0$ for all $x \in \mathcal{X}$ because we have $L_{\text{def}}^{\mu_2}(h, r_2^*) = \frac{1}{3}$. Furthermore, such deferral

rules are not interchangeable, because we have $L_{\text{def}}^{\mu_1}(h, r_2^*) = L_{\text{def}}^{\mu_2}(h, r_1^*) = \frac{2}{3}$. As a result, $\Pr(\hat{r}|X)$ does not provide sufficient information for obtaining optimal deferral rule on true distribution.

For an arbitrary choice of deterministic deferral rule for empirical distribution, we state the following proposition as a proof of insufficiency of deferral labels for obtaining optimal deferral rule over the true distribution.

**Proposition G.2** (Impossibility of generalization of deferral labels). *For every deterministic deferral rule $\hat{r}$ for empirical distributions and based on the two losses $\mathbb{1}_{m \neq y}$ and $\mathbb{1}_{h(x) \neq y}$, there exist two probability measures $\mu_1$ and $\mu_2$ on $\mathcal{X} \times \mathcal{Y} \times \mathcal{M}$ such that the corresponding $(\hat{r}, X)$ for both measures is distributed equally. However, the optimal deferral $r_{\mu_1}^*$ and $r_{\mu_2}^*$ for these measures are not interchangeable, that is $L_{\text{def}}^{\mu_i}(h, r_{\mu_i}^*) \leq \frac{1}{3}$ while $L_{\text{def}}^{\mu_i}(h, r_{\mu_j}^*) = \frac{2}{3}$ for $i = 1, 2$ and $j \neq i$.*

*Proof.* As mentioned in (31), there are four possibilities of a deterministic deferral rule for empirical distribution based on the events $h(X) \neq Y$ and $M \neq Y$. The reason is that

$$\hat{r} = \begin{cases} 1 & h(x) \neq y , \, m = y \\ 0 & h(x) = y , \, m \neq y \\ a & h(x) \neq y , \, m \neq y \\ b & h(x) = y , \, m = y \end{cases},$$

the parameters $a$ and $b$ can take binary values. One of the cases in which $a = b = 0$ is analyzed previously in this section. We study the other cases as follows:

1. $\mathbf{a = 1, b = 0}$: In this case we have

$$\hat{r} = \begin{cases} 1 & h(x) \neq y \\ 0 & o.w. \end{cases}.$$

   If we define a measure $\mu_1$ such that

   $$\mu_1\big(h(X) \neq Y, M = Y\big) = \frac{1}{3}, \quad \mu_1\big(h(X) = Y, M \neq Y\big) = \frac{2}{3},$$

   and a measure $\mu_2$ such that

   $$\mu_2\big(h(X) \neq Y, M = Y\big) = \frac{1}{3}, \quad \mu_2\big(h(X) = Y, M = Y\big) = \frac{2}{3},$$

   then on one hand one can see that $\hat{r}$ is according to $Bern(\frac{1}{3})$ in both cases. On the other hand, because the probability of classifier accuracy is larger than human accuracy in $\mu_1$ and is smaller than human accuracy in $\mu_2$, we have $r_{\mu_1}^*(x) = 0$ while $r_{\mu_2}^*(x) = 1$. Therefore, we conclude that

   $$L_{\text{def}}^{\mu_1}(r_{\mu_1}^*, h) = \frac{1}{3},$$

   and

   $$L_{\text{def}}^{\mu_2}(r_{\mu_2}^*, h) = 0,$$

   while the losses with interchanging deferral policies are equal to

   $$L_{\text{def}}^{\mu_1}(r_{\mu_2}^*, h) = L_{\text{def}}^{\mu_2}(r_{\mu_1}^*, h) = \frac{2}{3}.$$

2. $\mathbf{a = 0, b = 1}$: In this case, the deferral rule is as

$$\hat{r} = \begin{cases} 0 & m \neq y \\ 1 & o.w. \end{cases}.$$

   Next, if we set two probability measures $\mu_1$ and $\mu_2$ such that

   $$\mu_1\big(M \neq Y, h(X) = Y\big) = \frac{1}{3}, \quad \mu_1\big(M = Y, h(X) \neq Y\big) = \frac{2}{3},$$

   and

   $$\mu_2\big(M \neq Y, h(X) = Y\big) = \frac{1}{3}, \quad \mu_2\big(M = Y, h(X) = Y\big) = \frac{2}{3},$$

then $\hat{r}$ is according to $Bern(\frac{2}{3})$ in both cases. However, $r^*_{\mu_1} = 1$ while $r^*_{\mu_2} = 0$. Furthermore, the expected deferral losses are equal to

$$L^{\mu_1}_{\text{def}}(r^*_{\mu_1}, h) = \frac{1}{3}, \;\; L^{\mu_2}_{\text{def}}(r^*_{\mu_2}, h) = 0,$$

while after interchanging the deferral policies we have

$$L^{\mu_1}_{\text{def}}(r^*_{\mu_2}, h) = L^{\mu_2}_{\text{def}}(r^*_{\mu_1}, h) = \frac{2}{3}.$$

3. $\mathbf{a = 1, b = 1}$: In this case, the deferral rule is as

$$\hat{r} = \begin{cases} 0 & h(x) = y, m \neq y \\ 1 & o.w. \end{cases}.$$

Next, if we set two probability measures $\mu_1$ and $\mu_2$ such that

$$\mu_1(M \neq Y, h(X) = Y) = \frac{1}{3}, \;\; \mu_1(M = Y, h(X) \neq Y) = \frac{2}{3},$$

and

$$\mu_2(M \neq Y, h(X) = Y) = \mu_2(M \neq Y, h(X) \neq Y) = \mu_2(M = Y, h(X) = Y) = \frac{1}{3},$$

then we can see that $\hat{r}$ has the distribution $Bern(\frac{2}{3})$. However, one can find the optimal deferral policies for the true distributions are $r^*_{\mu_1} = 1$ and $r^*_{\mu_2} = 0$. Furthermore, we have

$$L^{\mu_1}_{\text{def}}(r^*_{\mu_1}, h) = \frac{1}{3},$$

and

$$L^{\mu_2}_{\text{def}}(r^*_{\mu_2}, h) = \frac{2}{3},$$

while

$$L^{\mu_1}_{\text{def}}(r^*_{\mu_1}, h) = \frac{1}{3}, \;\; L^{\mu_2}_{\text{def}}(r^*_{\mu_2}, h) = \frac{1}{3}.$$

$\square$

## H  Proof of Theorem 3.1

Let $\mathcal{X} = \{x_1, \ldots, x_n\}$ and $\mathcal{Y} = \{1, \ldots, n\}$. We first show that obtaining the optimal classifier is of $O(n)$ complexity, since in this case is equivalent to obtaining the Bayes optimal classifier in isolation. The reason is that, the unconstrained Bayes optimal classifier is a deterministic classifier that minimizes

$$h^*(x) \in \operatorname*{argmin}_{\hat{y}} \mathbb{E}_{\mu_{Y|X}}\left[\ell_{AI}(Y, \hat{y}, X)|X = x\right],$$

for all $x \in \mathcal{X}$. This is regardless of whether the deferral occurs or not. Therefore, this solution is further the solution to

$$h^*(x) \in \operatorname*{argmin}_{\hat{y}} \mathbb{E}_{\mu_{Y|X}}\left[(1 - r(X))\ell_{AI}(Y, \hat{y}, X)|X = x\right]$$
$$= \operatorname*{argmin}_{\hat{y}} \mathbb{E}_{\mu_{Y,M|X}}\left[(1 - r(X))\ell_{AI}(Y, \hat{y}, X) + r(X)\ell_H(Y, M, X)|X = x\right],$$

for every rejection function $r$, including the optimal rejection function of the constrained optimization problem. In the particular case of expert intervention budget, the constraint is further independent of $h$ and is only a function of $r$. Therefore, the unconstrained Bayes classifier is an optimal classifier for the constrained L2D problem with human intervention budget.

Next, we consider a specific case in which $\mathbb{E}_{\mu_{Y|X}}\left[\ell_{AI}(Y, 1, X)|X = x\right] > \mathbb{E}_{\mu_{Y|X}}\left[\ell_{AI}(Y, 0, X)|X = x\right]$ for all $x \in \mathcal{X}$, and therefore $h(x) = 1$ over all input space.

Further, we assume the data distribution has the property $\mu_{XYM} = \mu_{XY}\delta(M = Y)$, i.e. $M = Y$ on all the data. In this case, we know that

$$\mathbb{E}\big[\ell_H(Y, M, X)|X = x_i\big] = \mathbb{E}\big[\mathbb{1}_{M \neq Y}|X = x_i\big] = 0,$$

and we define

$$\mathbb{E}\big[\ell_{AI}(Y, h(X), X)|X = x_i\big] = \mathbb{E}\big[\mathbb{1}_{Y \neq 1}|X = x_i\big] = \ell_i.$$

Now, if we set $\Pr(X = x_i) = p_i$, and $r(x_i) = r_i$, then the optimization problem

$$f^* = \underset{r(\cdot) \in \{0,1\}^{\mathcal{X}}}{\operatorname{argmin}} L_{\text{def}}^{\mu}(h, r),$$

is equivalent to

$$\underset{r_i \in \{0,1\}}{\operatorname{argmin}} \sum_{i=1}^{n} p_i \times 0 \times r_i + p_i \times (1 - r_i) \times \ell_i, \quad s.t. \quad \sum_{i=1}^{n} p_i r_i \leq b,$$

that is equivalent to

$$\underset{r_i \in \{0,1\}}{\operatorname{argmax}} \sum_{i=1}^{n} p_i r_i \ell_i, \quad s.t. \quad \sum_{i=1}^{n} p_i r_i \leq b. \tag{32}$$

Next, we show that the above problem spans all instances of the $0 - 1$ knapsack problem, which is known to be NP-hard (Theorem 15.8 of [58]). Let

$$\underset{r_i \in \{0,1\}}{\operatorname{argmax}} \sum_{i=1}^{n} r_i c_i, \quad s.t. \quad \sum_{i=1}^{n} w_i r_i \leq K, \tag{33}$$

be an instance of the $0 - 1$ knapsack problem [7] with $w_i, c_i > 0$, $i \in [n]$, and $K > 0$. With $\ell_i = \frac{c_i/w_i}{\sum_{i=1}^{n} c_i/w_i}$, $p_i = \frac{w_i}{\sum_{i=1}^{n} w_i}$ and $b = \frac{K}{\sum_{i=1}^{n} w_i}$, problem (33) can be written in the form of (32). Because of $\sum_{i=1}^{n} l_i = \sum_{i=1}^{n} p_i = 1$ this yields indeed a valid problem.

# I   Proof of Theorem 4.1

We start this proof by introducing a few useful lemmas:

**Lemma I.1.** *The set $\mathcal{F} = \Delta_n^{\mathcal{X}}$ of all functions that map $\mathcal{X}$ to an $n-$dimensional probability is weakly compact, i.e., for each sequence $\{f_n\}_{n=1}^{\infty}$, there is a sub-sequence $\{f_{n_i}\}$ and a function $f^* \in \mathcal{F}$ such that for all measurable embedding functions $\psi$, we have*

$$\lim_{k \to \infty} \mathbb{E}\big[\langle f_{n_k}, \psi \rangle\big] = \mathbb{E}\big[\langle f^*, \psi \rangle\big].$$

*Proof.* We know that all components of each element of the function sequence is bounded by 1. We define $\{f_m^i\}_{m=1}^{\infty}$ as the sequence of the $i$th component of the function sequence. Therefore, using [42, Theorem A.5.1] we know that there is a sub-sequence $\{f_{m_k^1}^1\}_{k=1}^{\infty}$ and a non-negative 1-bounded function $f_1^*$, such that for each $\mu$-integrable function $\psi_1(x)$ we have

$$\lim_{k \to \infty} \mathbb{E}_{\mu}\big[f_{m_k}^1(x)\psi_1(x)\big] = \mathbb{E}_{\mu}\big[f_1^*(x)\psi_1(x)\big].$$

Next, we can repeat the same process for $\{f_{m_k^1}^i\}_{k=1}^{\infty}$ where $i \in [2 : n]$, and we can find a sub-sequence $m_k^{i+1}$ of $m_k^i$ and a non-negative 1$-$bounded function $f_{i+1}^*$ for which

$$\lim_{k \to \infty} \mathbb{E}_{\mu}\big[f_{m_k^{i+1}}^{i+1}(x)\psi_{i+1}(x)\big] = \mathbb{E}_{\mu}\big[f_{i+1}^*(x)\psi_{i+1}(x)\big].$$

Now, since all sub-sequences of a converging sequence converges to the same limit, we can use $m_k^n$ that is the intersection of all sequences and show that

$$\lim_{k \to \infty} \mathbb{E}_{\mu}\big[f_{m_k^n}^i(x)\psi_i(x)\big] = \mathbb{E}_{\mu}\big[f_i^*(x)\psi_i(x)\big],$$

---

[7]Note that in case that $w_i = 0$ the Knapsack problem has a degenerate solution of $r_i = 1$. Hence, we could drop that point and without loss of generality assume that $w_i$ is non-zero.

for all $i \in [1 : n]$ and integrable functions $\psi_i$. As a result, due to interchangeability of limit and summation, when the sum is over a finite set of elements, it is easy to show that

$$\lim_{k \to \infty} \mathbb{E}\big[\langle f_{m_k^n}, \psi \rangle\big] = \lim_{k \to \infty} \mathbb{E}\Big[\sum_{i=1}^n f_{m_k^n}^n(x)\psi_i(x)\Big] = \sum_{i=1}^n \lim_{k \to \infty} \mathbb{E}\big[f_{m_k^n}^n(x)\psi_i(x)\big]$$

$$= \sum_{i=1}^n \mathbb{E}_\mu\big[f_i^*(x)\psi_i(x)\big] = \mathbb{E}_\mu\big[\langle f^*(x), \psi(x)\rangle\big].$$

Next, we need to show that $f^* \in \mathcal{F}$. We already know that all components of $f^*$ is 1-bounded and non-negative. Therefore, we only need to prove that all elements of $f^*$ sum up to 1 almost everywhere. If not, then assume that there is a non-zero set $A$ where $\mu(A) > 0$ and there exists $l > 0$ such that $|\sum_i f_i^* - 1| \geq l$ for all $x \in A$. We know that there is either a subset $B \subseteq A$ with $\mu(B) > 0$ such that for all $x \in B$ we have $\sum_i f_i^*(x) \geq 1 + l$, or similarly a subset for which $\sum_i f_i^*(x) \leq 1 - l$. The reason is that otherwise a non-zero measure set $A$ is a union of two zero-measure set, which is a contradiction. Without loss of generality we assume the first, which means $\sum_i f_i^*(x) \geq 1 + l$ for $x \in B$. Now, if we define $\hat{\psi}(x) = [1, \dots, 1]$ for $x \in B$ and otherwise $\hat{\psi}(x) = [0, \dots, 0]$, then we have

$$\mathbb{E}_\mu\big[\langle f^*(x), \hat{\psi}(x)\rangle\big] \geq (1 + l)\mu(B),$$

while

$$\mathbb{E}_\mu\big[\langle f_{m_k^n}(x), \psi\rangle\big] = 1,$$

for all $k \in \mathbb{N}$. This is a contradction, because the limit of a constant sequence is not different from that constant value. Hence, $f^*$ sums up to 1 almost everywhere, and that completes the proof.

**Proof of Theorem 4.1**: We prove the theorem using the following steps: (i) for the class $\mathcal{C}$ of prediction functions for which $\mathbb{E}\big[\langle f(x), \psi_i(x)\rangle\big] = \delta_i$ for $i \in [1 : m]$, we show that the supremum of the objective function $\mathbb{E}\big[\langle f(x), \psi_0(x)\rangle\big]$ is a maximum, (ii) we show that it is sufficient for a predictor $f \in \mathcal{C}$ to be in form of (8) to achieve the maximum objective $\mathbb{E}\big[\langle f(x), \psi_0(x)\rangle\big]$ in $\mathcal{C}$ and also for all predictors with $\mathbb{E}\big[\langle f(x), \psi_i(x)\rangle\big] \leq \delta_i$, (iii) we show that the space of all possible constraints for any prediction function in $\Delta_d^{\mathcal{X}}$ is convex and compact, and (iv) we show that if the tuple of constraints is an interior point of all possible tuples of constraints, then a point in $\mathcal{C}$ achieves its maximum if and only if it follows the thresholding rule (8) almost everywhere.

- **Step (i)**: Due to the definition of supremum, we know that for each $\epsilon > 0$, there exists a function $f_\epsilon$ in $\mathcal{C}$ such that $\mathbb{E}\big[\langle f_\epsilon, \psi_0(x)\rangle\big] \geq \sup_{f \in \mathcal{C}} \mathbb{E}\big[\langle f, \psi_0(x)\rangle\big] - \epsilon$. Equivalently, there is a sequence of functions $f_n$ for which $\lim_{n \to \infty} \mathbb{E}\big[\langle f_n, \psi_0(x)\rangle\big] = \sup_{f \in \mathcal{C}} \mathbb{E}\big[\langle f, \psi_0(x)\rangle\big]$. Using weakly-compactness of the function class $\Delta_{n+1}^{\mathcal{X}}$ as in Lemma I.1, we know that for the sequence $f_n$, there exists a subsequence $f_{n_k}$ and a function $f^* \in \Delta_{n+1}^{\mathcal{X}}$ such that

$$\lim_{k \to \infty} \mathbb{E}\big[\langle f_{n_k}, \psi_{m+1}(x)\rangle\big] = \mathbb{E}\big[\langle f^*(x), \psi_{m+1}(x)\rangle\big].$$

Furthermore, we know that each subsequence $a_{n_k}$ of a converging sequence $a_n$ has the same limit as the limit of the mother sequence $a_n$ [59, Chapter 2, Theorem 1]. Therefore, we have

$$\mathbb{E}\big[\langle f^*(x), \psi_{m+1}(x)\rangle\big] = \sup_{f \in \mathcal{C}} \mathbb{E}\big[\langle f, \psi_{m+1}(x)\rangle\big],$$

which means that the supremum of the objective is achievable by $f^*$.

Moreover, for $\psi_i(x)$ where $i \in [1 : m]$, we have $\mathbb{E}\big[\langle f_n, \psi_i(x)\rangle\big] = \delta_i$ for all $n$, which concludes

$$\delta_i = \lim_{k \to \infty} \mathbb{E}\big[\langle f_{n_k}, \psi_i(x)\rangle\big] = \mathbb{E}\big[\langle f^*(x), \psi_i(x)\rangle\big].$$

This means that the equality constraints holds for $f^*$, i.e., $f^* \in \mathcal{C}$, if it holds for each predictor $f_n$.

- **Step (ii)**: Assume that there is a predictor $\hat{f}$ such that $\mathbb{E}\big[\langle \hat{f}, \psi_i \rangle\big] \leq \delta_i$. In this step, we show that if exists a predictor $f$ in form of (8) and in $\mathcal{C}$, then $\hat{f}$ always has smaller objective than $\hat{f}$. To that end, consider the following expression:

$$A = \mathbb{E}\Big[\langle f(x) - \hat{f}(x), \psi_0(x) - \sum_{i=1}^m k_i\psi_i(x)\rangle\Big].$$

Now, we know that

$$\mathbb{E}\big[\langle f(x) - \hat{f}(x), \sum_{i=1}^{m} k_i \psi_i(x)\rangle\big] = \sum_{i=1}^{m} k_i \Big(\mathbb{E}\big[\langle f(x), \psi_i(x)\rangle\big] - \mathbb{E}\big[\langle \hat{f}(x), \psi_i(x)\rangle\big]\Big)$$

$$\stackrel{(a)}{=} \sum_{i=1}^{m} k_i \Big(\delta_i - \mathbb{E}\big[\langle \hat{f}(x), \psi_i(x)\rangle\big]\Big) \geq 0,$$

where $(a)$ holds because $f \in \mathcal{C}$. As a result, if $A \geq 0$, then we could show that

$$\mathbb{E}\big[\langle f(x) - \hat{f}(x), \psi_0(x)\rangle\big] \geq 0, \tag{34}$$

and complete the proof.

To that end, first note that both $f$ and $\hat{f}$ are in $\Delta_d^{\mathcal{X}}$, and therefore

$$\langle f(x), [1, \ldots, 1]\rangle = \langle \hat{f}(x), [1, \ldots, 1]\rangle = 1.$$

As a result, for any fixed scalar $c$, we have

$$\langle f(x) - \hat{f}(x), \psi_0(x) - \sum_{i=1}^{m} k_i \psi_i(x)\rangle = \langle f(x) - \hat{f}(x), \psi_0(x) - \sum_{i=1}^{m} k_i \psi_i(x) - c\rangle. \tag{35}$$

Next, we fix $c$ to be the maximum component of the vector $\psi_0(x) - \sum_{i=1}^{m} k_i \psi_i(x)$, i.e.,

$$c := \max_{i \in [1:d]} \{\psi_0^i(x) - \sum_{j=1}^{m} k_j \psi_j^i(x)\}.$$

Now, we rewrite $A$ using (35) as

$$A = \mathbb{E}\Big[\langle f(x) - \hat{f}(x), \psi_0(x) - \sum_{i=1}^{m} k_i \psi_i(x) - c\rangle\Big]$$

$$= \sum_{i=1}^{d} \mathbb{E}\Big[(f_i(x) - \hat{f}_i(x))(\psi_0^i(x) - \sum_{j=1}^{m} k_j \psi_j^i(x) - c)\rangle\Big]$$

Now, we consider two cases for which $E_1^i(x) : f_i(x) > \hat{f}_i(x)$, and $E_2^i(x) : f_i(x) \leq \hat{f}_i(x)$. If $f_i(x) > \hat{f}_i(x)$, then we have $f_i(x) > 0$, because $1 \geq \hat{f}_i(x) \geq 0$ for all $i \in [1 : d]$. Therefore, using the definition of $\mathcal{S}_d$ and because $f \in \mathcal{S}_d$ we have

$$\psi_0^i(x) - \sum_{j=1}^{m} k_j \psi_j^i(x) = \max_{i \in [1:d]} \{\psi_0^i(x) - \sum_{j=1}^{m} k_j \psi_j^i(x)\} = c. \tag{36}$$

Consequently, we have

$$A = \sum_{i=1}^{d} \mathbb{E}\Big[(f_i(x) - \hat{f}_i(x))(\psi_0^i(x) - \sum_{j=1}^{m} k_j \psi_j^i(x) - c)\rangle\Big]$$

$$= \sum_{i=1}^{d} \mathbb{E}\Big[(f_i(x) - \hat{f}_i(x))(\psi_0^i(x) - \sum_{j=1}^{m} k_j \psi_j^i(x) - c)\rangle | E_1^i(x)\Big] \Pr\big(E_1^i(x)\big)$$

$$+ \sum_{i=1}^{d} \mathbb{E}\Big[(f_i(x) - \hat{f}_i(x))(\psi_0^i(x) - \sum_{j=1}^{m} k_j \psi_j^i(x) - c)\rangle | E_2^i(x)\Big] \Pr\big(E_2^i(x)\big)$$

$$\stackrel{(a)}{=} \sum_{i=1}^{d} \mathbb{E}\Big[(f_i(x) - \hat{f}_i(x))(\psi_0^i(x) - \sum_{j=1}^{m} k_j \psi_j^i(x) - c)\rangle | E_2^i(x)\Big] \Pr\big(E_2^i(x)\big)$$

$$\stackrel{(b)}{\geq} 0,$$

where $(a)$ holds due to (36) and $(b)$ holds because $f_i(x) \leq \hat{f}_i(x)$ and $\psi_{m+1}^i(x) - \sum_{j=1}^{m} k_j \psi_j^i(x) \leq c = \max_{i \in [1:n+1]} \{\psi_{m+1}^i(x) - \sum_{j=1}^{m} k_j \psi_j^i(x)\}$. As a result, we have $A \geq 0$ that concludes (34) and completes the proof of this step.

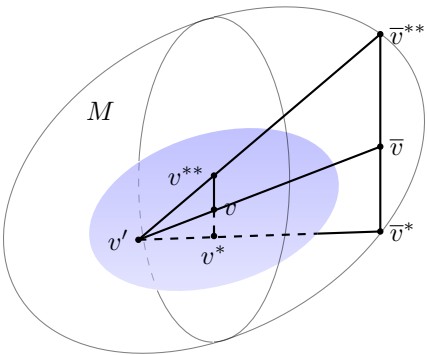

Figure 4: If an interior point of $\mathcal{N}$ has one corresponding point at $M$, then so are all interior points of $N$

- **Step (iii)**: In this step, we show that the space of joint set of expected inner-products

$$\mathcal{G} = \left\{ \big(\mathbb{E}[\langle f(x), \psi_1(x)\rangle], \ldots, \mathbb{E}[f(x), \psi_m(x)]\big) : f \in \Delta_d^{\mathcal{X}} \right\},$$

is compact under Euclidean metric, and convex.

To show the compactness of that space, assume that there is a sequence $\{g_n\}_{n=1}^\infty$ such that $\lim_{n \to \infty} g_n = g$, or accordingly there is a sequence of $f_n \in \Delta_d^{\mathcal{X}}$ for which $\lim_{n \to \infty} \big(\mathbb{E}[\langle f_n(x), \psi_1(x)\rangle], \ldots, \mathbb{E}[f_n(x), \psi_m(x)]\big) = (g_1, \ldots, g_m)$. Since the metric is Euclidean, this is equivalent to $\lim_{n \to \infty} \mathbb{E}[\langle f_n(x), \psi_i(x)\rangle] = g_i$ for all $i \in [1 : m]$. The weak compactness of $\Delta_d^{\mathcal{X}}$, as proved in Lemma I.1, shows that there exists $f^*$ and a subsequence $f_{n_k}$ such that $\lim_{k \to \infty} \mathbb{E}[\langle f_{n_k}(x), \psi_i(x)\rangle] = \mathbb{E}[\langle f^*, \psi_i(x)\rangle]$ for all $i \in [1 : d]$. Therefore, because of the choice of Euclidean metric, we have

$$\lim_{k \to \infty} \Big(\mathbb{E}[\langle f_{n_k}(x), \psi_1(x)\rangle], \ldots, \mathbb{E}[\langle f_{n_k}(x), \psi_m(x)\rangle]\Big)$$
$$= \Big(\mathbb{E}[\langle f^*, \psi_1(x)\rangle], \ldots, \mathbb{E}[\langle f^*, \psi_m(x)\rangle]\Big),$$

which is equivalent to compactness of $\mathcal{G}$.

To show the convexity of $\mathcal{G}$, it is enough to prove the convexity of $\Delta_d^{\mathcal{X}}$. The reason is that $g(f) = \big(\mathbb{E}[\langle f(x), \psi_1(x)\rangle], \ldots, \mathbb{E}[f(x), \psi_m(x)]\big)$ is a linear functional of $f$, and a linear functional images a convex set to another convex set.

To prove the convexity of $\Delta_d^{\mathcal{X}}$, let $f, f' \in \Delta_d^{\mathcal{X}}$. This means that $f_i(x), f_i'(x) \in [0, 1]$ for all $i \in [1 : d]$ and $\sum_{i=1}^d f_i(x) = \sum_{i=1}^d f_i'(x) = 1$. Consequently, $a f_i(x) + (1 - a) f_i'(x) \geq 0$, since $a, 1 - a \geq 0$. Moreover, $\sum_{i=1}^d a f_i(x) + (1 - a) f_i'(x) = a \sum_{i=1}^d f_i(x) + (1 - a) \sum_{i=1}^d f_i'(x) = a + 1 - a = 1$. As a result of these two facts, $a f + (1 - a) f' \in \Delta_d^{\mathcal{X}}$, and the proof of this step is completed.

- **Step (iv)**: In this step we show that if the tuple of constraints is an interior points of all possible achievable tuples of constraints using different prediction functions, then a point in $\mathcal{C}$ achieves its supremum in terms of objective $\mathbb{E}\big[\langle f(x), \psi_0(x)\big]$ if and only if it is in form of (8) almost everywhere. This is an extension to [21, Theorem 3.1] and its proof resembles to the proof that is provided there. The sufficiency is already shown in Step (ii). Therefore, we only need to show that if a prediction function in $\mathcal{C}$ maximizes the objective, then it is in form of (8).

Firstly, using Step (iii), we know that the space $\mathcal{N}$ of all points $\big(\mathbb{E}[\langle f(x), \psi_1(x)\rangle], \ldots, \mathbb{E}[f(x), \psi_m(x)]\big)$ and the space $M$ of all points $\big(\mathbb{E}[\langle f(x), \psi_1(x)\rangle], \ldots, \mathbb{E}[f(x), \psi_0(x)]\big)$ are compact and convex. Now, assume that $v = (\delta_1, \ldots, \delta_m)$ is an interior point of $\mathcal{N}$. Then, the corresponding set in $M$, i.e., $B_v = \{(\delta_0, \ldots, \delta_m) \in M : \delta_0 \in \mathbb{R}\}$ has a supremum and an infimum of the first component that we name $\delta^{**}$ and $\delta^*$. Now, since $M$ is compact, then $v^{**} = (\delta^{**}, \delta_1, \ldots, \delta_m)$ and $v^* = (\delta^*, \delta_1, \ldots, \delta_m)$ are contained in $M$. Next, assume the following two cases:

1. $\delta^{**} = \delta^*$: In this case for all other points $\overline{v} = (\overline{\delta}_1, \ldots, \overline{\delta}_m)$ of $\mathcal{N}$, the corresponding set $B_{v'}$ in $\mathcal{M}$ is a single point. The reason is that, if it is not so, then we have two points $\overline{v}^{**} = (\overline{\delta}^{**}, \overline{\delta}_1, \ldots, \overline{\delta}_m)$ and $\overline{v}^* = (\overline{\delta}^*, \overline{\delta}_1, \ldots, \overline{\delta}_m)$ where $\overline{\delta}^{**} > \overline{\delta}^*$. Now, since $v$ is an interior point of $\mathcal{N}$, then on any direction in a small neighborhood around $v$ there exists a point $v'$ within $\mathcal{N}$. Let that direction be opposite the connecting line of $v$ and $\overline{v}$, i.e., let $v$ be on a connecting line of $v'$ and $\overline{v}^*$. Now, make a convex hull using the three points $v'$, $\overline{v}^{**}$, and $\overline{v}^*$, which are all in $\mathcal{M}$. Because of the convexity of $\mathcal{M}$, the convex hull is also a subset of $\mathcal{M}$. Since $v$ is an interior point of the convex hull, this means that a neighborhood of $v$ along any direction is inside $\mathcal{M}$. Now, if we set $(m+1)$th axis to be that direction, we contradict with the fact that $\delta^* = \delta^{**}$. (See Figure 4)

   Now, we know that in such case all points within $\mathcal{N}$ have one corresponding point in $\mathcal{M}$. Because of the convexity of $\mathcal{M}$ this is equivalent to $\mathcal{M}$ being a subset of a hyperplane with the generating formula $x_0 = \sum_{i=1}^{m} k_i x_i + k_0$. Therefore, we have $\mathbb{E}[\langle f, \psi_0 \rangle] = \mathbb{E}[\langle f, \sum_{i=1}^{m} k_i \psi_i \rangle] + k_0$ for all $f \in \Delta_d^{\mathcal{X}}$. Therefore, for $d \geq 2$, if we set $f_1 = (\frac{p(x)}{d-2}, \ldots, \frac{p(x)}{d-2}, \underbrace{1-p(x)}_i, \frac{p(x)}{d-2}, \ldots, \underbrace{0}_j, \frac{p(x)}{d-2}, \ldots, \frac{p(x)}{d-2})$ and $f_2 =$

   $(\frac{p(x)}{d-2}, \ldots, \frac{p(x)}{d-2}, \underbrace{0}_i, \frac{p(x)}{d-2}, \ldots, \underbrace{1-p(x)}_j, \frac{p(x)}{d-2}, \ldots, \frac{p(x)}{d-2})$ for $p(x) \in [0,1]^{\mathcal{X}}$, then we have

   $$\mathbb{E}[\langle f_1, \psi_0 \rangle] - \mathbb{E}[\langle f_1, \sum_{i=1}^{m} k_i \psi_i \rangle] = \mathbb{E}[\langle f_2, \psi_0 \rangle] - \mathbb{E}[\langle f_2, \sum_{i=1}^{m} k_i \psi_i \rangle],$$

   or equivalently

   $$\mathbb{E}[(1-p(x))(\psi_0^i(x) - \sum_{t=1}^{m} k_t \psi_t^i(x) - \psi_{m+1}^j(x) + \sum_{t=1}^{m} k_t \psi_t^j(x))] = 0,$$

   for all function $p(x) \in \Delta_d^{\mathcal{X}}$. A similar result can be achieved for $d = 2$ and by setting $f_1 = (p(x), 1-p(x))$ and $f_2 = (1-p(x), p(x))$. As a result, we have

   $$\psi_0^i(x) - \sum_{t=1}^{m} k_t \psi_t^i(x) = \psi_0^j(x) - \sum_{t=1}^{m} k_t \psi_t^j(x),$$

   for all $i \neq j \in [1:d]$, and consequently

   $$\psi_0^i(x) - \sum_{t=1}^{m} k_t \psi_t^i(x) = \max_{j \in [1:d]} \{\psi_0^j(x) - \sum_{t=1}^{m} k_t \psi_t^j(x)\},$$

   for all $i \in [1:n+1]$. As a result, there is a set of $k_1, \ldots, k_m$ such that $\psi_0(x) - \sum_{i=1}^{m} k_i \psi_i(x)$ has equal components almost everywhere. As a result, due to the freedom of choice for $\tau(\psi_0(x) - \sum_{i=1}^{m} k_i \psi_i(x), x)$ where $\tau \in \mathcal{S}_d$ and when we have more than one maximizer component, then, without loss of generality we can assume that every prediction function $f$ almost everywhere is in form of $\tau(\psi_{m+1}(x) - \sum_{i=1}^{m} k_i \psi_i(x), x)$.

2. $\delta^{**} > \delta^*$: In such case, for all $\delta_0 \in [\delta^*, \delta^{**}]$, we can show that $v = (\delta_0, \ldots, \delta_m)$ is an interior point of $\mathcal{M}$. To show that, we first find $m$ points $v'_1, \ldots, v'_m \in \mathcal{N}$ that are linearly independent and such that their convex hull include $(\delta_1, \ldots, \delta_m)$. Using the definition of $\mathcal{M}$, for each of these points $v'_i = (\delta'^i_1, \ldots, \delta'^i_m)$, there exists $h'_i \in \mathbb{R}$ such that $v''_i = (h'_i, \delta'^i_1, \ldots, \delta'^i_m)$ is within $\mathcal{M}$. Now, we add the two points $v^{**}$ and $v^*$ to these sets of points. It is easy to see that $v''_i$s are linearly independent. Furthermore, we know that $(\delta_1, \ldots, \delta_m)$ is a convex combination of $v'_i$s, i.e., $\sum_i a_i v'_i = (\delta_1, \ldots, \delta_m)$. As a result, if $\sum_i b_i v''_i - v^{**} = (0, \ldots, 0)$, then we have $b_i = a_i$ for $i \in [1:m]$. Furthermore, we have $\sum a_i h'_i = \sum b_i h'_i = \delta^{**}$. Similarly, if $\sum_i c_i v''_i - v^* = (0, \ldots, 0)$ we have $c_i = a_i$ and $\sum a_i h'_i = \sum c_i h'_i = \delta^*$. As a result, since $\delta^* \neq \delta^{**}$ at least one of these cases would not occur, or equivalently, the dimension of the convex hull of $v''_1, \ldots v''_m, v^{**}, v^*$ is of dimension $m + 1$. As a result, $v$ is an interior point of

this convex hull, and because the convex hull is $(m + 1)$-dimensional, it is an interior point of $\mathcal{M}$.

Now, since $v^{**}$ is a border point in $\mathcal{M}$ and due to the convexity of $\mathcal{M}$ there is a hyperplane $\mathcal{P}$ such that it passes $v^{**}$ and it lays above all points of $\mathcal{M}$. Since $v$ is an interior point of $\mathcal{M}$, a neighborhood of $v$ is laid under the hyperplane $\mathcal{P}$, hence $v$ cannot be laid on the hyperplane. Therefore, if the generating formula of such hyperplane is $\sum_{i=0}^{m} k_i x_i = \sum_{i=1}^{m} k_i \delta_i + k_0 \delta^{**}$, since $v$ is not laid on the hyperplane we assure that $\sum_{i=1}^{m} k_i \delta_i + k_0 \delta_0 \neq \sum_{i=1}^{m} k_i \delta_i + k_0 \delta^{**}$, or equivalently $k_0 \neq 0$. Hence, without loss of generality assume that $k_0 = -1$. This shows that for all points in $(u_0, \ldots, u_m) \in \mathcal{M}$ we have

$$u_0 - \sum_{i=1}^{m} k_i u_i \leq \delta^{**} - \sum_{i=1}^{m} k_i \delta_i,$$

or equivalently, by the definition of $\mathcal{M}$, for all prediction function $f$, we have

$$\mathbb{E}\big[\langle f(x), \psi_0(x) - \sum_{i=1}^{m} k_i \psi_i(x)\rangle\big] \leq \delta^{**} - \sum_{i=1}^{m} k_i \delta_i.$$

Assuming that $\hat{f} \in \mathcal{C}$ maximizes the objective, we have

$$\mathbb{E}\big[\langle f(x), \psi_0(x) - \sum_{i=1}^{m} k_i \psi_i(x)\rangle\big] \leq \mathbb{E}\big[\langle \hat{f}(x), \psi_0(x) - \sum_{i=1}^{m} k_i \psi_i(x)\rangle\big]. \tag{37}$$

This shows that almost everywhere when there is a unique maximizing component $j$ in $\psi_0(x) - \sum_{i=1}^{m} k_i \psi_i(x)$, then $\hat{f}_j(x) = 1$. The reason is that otherwise and if there is a set $A$ such that $\mu(A) > 0$ and for $x \in A$ and a choice of $l \in [0, 1)$, $\epsilon \in \mathbb{R}$, and all $t \neq j$ we have $\psi_{m+1}^{j}(x) - \sum_{i=1}^{m} k_i \psi_i^{j}(x) \geq \epsilon + \psi_{m+1}^{t}(x) - \sum_{i=1}^{m} k_i \psi_i^{t}(x)$ while $f_j \leq 1 - l$, then we can make a function $f(x)$ that is $f(x) = \hat{f}(x)$ for $x \in \mathcal{X} \setminus A$ and $f(x) = [0, \ldots, \underbrace{1}_{j}, \ldots, 0]$ for $x \in A$. Such function leads to

$$\mathbb{E}\big[\langle f(x), \psi_0(x) - \sum_{i=1}^{m} k_i \psi_i(x)\rangle\big] \geq \epsilon l \mu(A) + \mathbb{E}\big[\langle \hat{f}(x), \psi_0(x) - \sum_{i=1}^{m} k_i \psi_i(x)\rangle\big],$$

that is in contradiction with (37). This completes the proof of this step.

## J   Proof of Theorem 4.2

In the following, we introduce a few lemmas that are useful in our proofs.

**Lemma J.1.** *For every random variable $X$ on $\mathbb{R}$ we have*

$$\lim_{\tau \to t^{-}} \Pr(\tau \leq X < t) = \lim_{\tau \to t^{+}} \Pr(t < X < \tau) = 0$$

*Proof.* For each increasing sequence $\{\tau_i\}_{i=1}^{\infty}$ we show that the first limit is zero, which proves the claim that the function of $\tau$ has a zero limit.
We define

$$\mathcal{S}_i = [\tau_i, t),$$

and notice that

$$\mathcal{S}_1 \supseteq \mathcal{S}_2 \supseteq \ldots.$$

Further, we note that

$$\bigcap_{i=1}^{\infty} \mathcal{S}_i = \emptyset.$$

As a result

$$\mathcal{S}_1^{c} \subseteq \mathcal{S}_2^{c} \subseteq \ldots,$$

and

$$\bigcup_{i=1}^{\infty} \mathcal{S}_i^c = \mathbb{R}.$$

Next, because probability measure is $\sigma$-additive, we conclude its lower-semicontinuity [38, Theorem 13.6], and therefore we have

$$\lim_{i \to \infty} \Pr(X \in \mathcal{S}_i^c) = \Pr(X \in \cup_{i=1}^{\infty} \mathcal{S}_i^c) = 1,$$

which proves $\lim_{i \to \infty} \Pr(X \in \mathcal{S}_i) = 0$.
We could take similar steps to show that since $\bigcap_{i=1}^{\infty}(t, \tau_i') = \emptyset$ for decreasing $\tau_i'$ we have

$$\lim_{i \to \infty} \Pr(X \in (t, \tau_i')) = 0.$$

$\square$

**Lemma J.2.** *Let $\psi_1 : \mathcal{X} \to \mathbb{R}^d$ be a bounded function. Further, we define two functions $C(k) = \mathbb{E}\big[\langle f_{k,0}^*(x), \psi_1(x)\rangle\big]$, $D(k) = \mathbb{E}\big[\langle f_{k,1}^*(x), \psi_1(x)\rangle\big]$, and $F(k) = \mathbb{E}\big[\langle f_{k,1}^*(x), \psi_0(x)\rangle\big]$, where $f_{k,p}^*$ is defined in Theorem 4.2. Then,*

1. *$C(k)$ is monotonically non-increasing,*
2. *$C(k)$ is upper semi-continuous,*
3. *$F(k)$ is monotonically non-decreasing,*
4. *$D(k)$ is lower semi-continuous, and we have*
5. *$\lim_{k' \uparrow k} C(k) = \lim_{k' \uparrow k} D(k)$*

*Proof.* 1. Firstly, let us define $\ell_k(x) = \psi_0(x) - k\psi_1(x)$. For the setting where $p = 0$, the prediction function $f_{k,p}^*(x)$ is defined as

$$f_{k,0}^*(x, p) = \begin{cases} 1 & i = \min\{\underset{j \in \operatorname{argmax} \ell_k(x)}{\operatorname{argmin}} \big(\psi_1(x)\big)(j)\} \\ 0 & \text{otherwise} \end{cases}. \tag{38}$$

Further, for $k_1, k_2$ such that $k_1 \le k_2$, let us define $j_1$ and $j_2$ as the only non-zero index of $f_{k_1,0}^*(x, p)$ and $f_{k_2,0}^*(x, p)$, respectively. To show that $C(k)$ is monotonically non-increasing we only need to show that $\big(\psi_1(x)\big)(j_1) = \langle f_{k_1,0}^*(x), \psi_1(x)\rangle \ge \langle f_{k_2,0}^*(x), \psi_1(x)\rangle = \big(\psi_1(x)\big)(j_2)$. Assume that this does not occur, or equivalently $\big(\psi_1(x)\big)(j_1) < \big(\psi_1(x)\big)(j_2)$. In such case we have

$$\begin{aligned}
\max \ell_{k_2}(x) &\overset{(a)}{=} \big(\ell_{k_2}(x)\big)(j_2) \\
&= \big(\ell_{k_1}(x) - (k_2 - k_1)\psi_1(x)\big)(j_2) \\
&\le (k_1 - k_2)\big(\psi_1(x)\big)(j_2) + \max_j \big(\ell_{k_1}(x)\big)(j) \\
&\overset{(b)}{=} (k_1 - k_2)\big(\psi_1(x)\big)(j_2) + \big(\ell_{k_1}(x)\big)(j_1) \\
&\overset{(c)}{<} (k_1 - k_2)\big(\psi_1(x)\big)(j_1) + \big(\ell_{k_1}(x)\big)(j_1) \\
&= \big(\ell_{k_2}(x)\big)(j_2),
\end{aligned} \tag{39}$$

where $(a)$ and $(b)$ holds due to the definition of $j_1$ and $j_2$, and $(c)$ holds due to the assumption $\big(\psi_1(x)\big)(j_1) < \big(\psi_1(x)\big)(j_2)$. The last inequality is clearly a contradiction, and shows that $\langle f_{k_1,0}^*(x), \psi_1(x)\rangle \ge \langle f_{k_2,0}^*(x), \psi_1(x)\rangle$, and therefore $C(k_1) \ge C(k_2)$.

2. Let us divide the space $\mathcal{X}$ into two subsets

$$A_k = \Big\{ x \in \mathcal{X} : \big|\operatorname*{argmax}_i(\ell_k(x))(i)\big| = d \Big\},$$

$$B_k = \Big\{ x \in \mathcal{X} : \big|\operatorname*{argmax}_i(\ell_k(x))(i)\big| \in [1 : d - 1] \Big\}.$$

For each $x \in A_k$ we know

$$\big(f_{k,0}^*(x)\big)(i) = \begin{cases} 1 & i = \min\{\underset{j}{\arg\min}\,\big(\psi_1(x)\big)(j)\} \\ 0 & \text{otherwise} \end{cases}$$

Using previous part, we know that by increasing $k$ we have no increase in $\langle f_{k,0}^*(x), \psi_1(x)\rangle$, and in this case since $\langle f_{k,0}^*(x), \psi_1(x)\rangle = \min_j \big(\psi_1(x)\big)(j)$, then this value cannot reduce with the change of $k$. Therefore, $\langle f_{k,0}^*(x), \psi_1(x)\rangle$ is a constant function for all $k' \geq k$, and consequently $\mathbb{E}\big[\langle f_{k',0}^*(x), \psi_1(x)\rangle | x \in A_k\big] \Pr(x \in A_k)$ is a constant function for $k' \geq k$. If $x \in B_k$, then for $j \notin \underset{i}{\arg\max}\,\big(\ell_k(x)\big)(i)$ and $l \in \underset{i}{\arg\max}\,\big(\ell_k(x)\big)(i)$, we have $\big(\ell_k(x)\big)(j) < \big(\ell_k(x)\big)(l)$. Define the set $C_\delta$ for $\delta \geq 0$ as

$$C_\delta = \{x \in B_k : \big(\ell_k(x)\big)(j) \leq \big(\ell_k(x)\big)(l) - \delta\}.$$

Using Lemma J.1 we know that

$$\lim_{\delta \to 0} \Pr(B_k \setminus C_\delta) = 0,$$

or equivalently for all $\epsilon \geq 0$, there exists $\delta$ such that

$$\Pr(B_k \setminus C_\delta) \leq \epsilon'.$$

Therefore, if without loss of generality, we assume that $\psi_1(x)$ is bounded by 1, then there exists $\delta \geq 0$ such that we have

$$\mathbb{E}\big[\langle f_{k',0}^*(x), \psi_1(x)\rangle | x \in B_k \setminus C_\delta\big] \Pr(x \in B_k \setminus C_\delta)$$
$$\overset{(a)}{\leq} \|\psi_1(x)\|_\infty \Pr(x \in B_k \setminus C_\delta) \leq \epsilon/2,$$

where $(a)$ holds due to Hölder's inequality.
If $x \in C_\delta$, and because we assumed $\|\psi_1(x)\|_\infty \leq 1$, then we know that by increasing $k$ to $k' \in [k - \delta/2, k + \delta/2)$, we have

$$\mathcal{I} = \arg\max \ell_{k'}(x) \subseteq \arg\max \ell_k(x) = \mathcal{J}. \tag{40}$$

This means that

$$\langle f_{k,0}^*(x), \psi_1(x)\rangle = \min_{j \in \mathcal{J}} \big(\psi_1(x)\big)(j) \leq \min_{j \in \mathcal{I}} \big(\psi_1(x)\big)(j) = \langle f_{k',0}^*(x), \psi_1(x)\rangle.$$

This, together with the previous part in which we showed $\langle f_{k,0}^*(x), \psi_0(x)\rangle \geq \langle f_{k',0}^*(x), \psi_0(x)\rangle$, concludes that $\langle f_{k,0}^*(x), \psi_0(x)\rangle = \langle f_{k',0}^*(x), \psi_0(x)\rangle$. This means that $\mathbb{E}\big[\langle f_{k',0}^*(x), \psi_0(x)\rangle | x \in C_\delta\big] \Pr(x \in C_\delta)$ is a constant function for all $k' \geq k$.
Finally, since we have

$$C(k') = \mathbb{E}\big[\langle f_{k',0}^*(x), \psi_1(x)\rangle\big] = \mathbb{E}\big[\langle f_{k',0}^*(x), \psi_1(x)\rangle | x \in A_k\big] \Pr(x \in A_k)$$
$$+ \mathbb{E}\big[\langle f_{k',0}^*(x), \psi_1(x)\rangle | x \in B_k \setminus C_\delta\big] \Pr(x \in B_k \setminus C_\delta)$$
$$+ \mathbb{E}\big[\langle f_{k',0}^*(x), \psi_1(x)\rangle | x \in C_\delta\big] \Pr(x \in C_\delta),$$

and because the first term and the third term in RHS are constant in terms of $k'$ and for $k' \geq k$, and the second term is diminishing, then we have

$$\big|C(k') - C(k)\big| = \big|\mathbb{E}\big[\langle f_{k',0}^*(x), \psi_1(x)\rangle | x \in B_k \setminus C_\delta\big] \Pr(x \in B_k \setminus C_\delta)$$
$$- \mathbb{E}\big[\langle f_{k,0}^*(x), \psi_1(x)\rangle | x \in B_k \setminus C_\delta\big] \Pr(x \in B_k \setminus C_\delta)\big| \leq \epsilon/2 + \epsilon/2,$$

which is equivalent to say that $\lim_{k' \uparrow k} C(k') = C(k)$.

3. For $p = 1$, we know that the prediction function $f_{k,p}^*(x)$ is obtained as

$$f_{k,1}^*(x) = \begin{cases} 1 & i = \min\{\underset{j \in \arg\max \ell_k(x)}{\arg\max}\,\big(\psi_0(x)\big)(j)\} \\ 0 & \text{otherwise} \end{cases}.$$

If we define $\psi_1'(x) := -\psi_0(x)$, then we have

$$f_{k,1}^*(x) = \begin{cases} 1 & i = \min\{\underset{j\in\operatorname{argmax}\ell_k(x)}{\operatorname{argmin}}\ (\psi_1'(x))(j)\} \\ 0 & \text{otherwise} \end{cases}.$$

Since the above is equal to (38), then using the first part of this lemma, we know that $\mathbb{E}\big[\langle f_{k,1}^*(x), \psi_1'(x)\rangle\big] = -\mathbb{E}\big[\langle f_{k,1}^*(x), \psi_0(x)\rangle\big]$ is monotonically non-increasing, which is equivalent to $F(k) = \mathbb{E}\big[\langle f_{k,1}^*(x), \psi_0(x)\rangle\big]$ being monotonically non-decreasing.

4. This part is similar to the second part of the proof. In fact, if $x \in A_k$, then we have

$$\big(f_{k,1}^*(x)\big)(i) = \begin{cases} 1 & i = \min\{\underset{j}{\operatorname{argmax}}\ (\psi_0(x))(j)\} \\ 0 & \text{otherwise} \end{cases}. \tag{41}$$

For $k' \leq k$ and because of the third part of this lemma, we know that $\langle f_{k',1}^*(x), \psi_0(x)\rangle \geq \langle f_{k,1}^*(x), \psi_0(x)\rangle$. Furthermore, because of (41) we know that $\langle f_{k,1}^*(x), \psi_0(x)\rangle = \max \psi_0(x)$, and therefore by reducing $k'$, the prediction function $f_{k',1}^*(x)$ stays constant. As a result, $\mathbb{E}\big[\langle f_{k',1}^*(x), \psi_1(x)\rangle | x \in A_k\big]\Pr(x \in A_k)$ is a constant function for $k' \leq k$. Furthermore, similar to the second part of this lemma, we can show that for each $\epsilon > 0$, there exists $\delta' \geq 0$ such that for all $0 \leq \delta \leq \delta'$ we have

$$\mathbb{E}\big[\langle f_{k',1}^*(x), \psi_1(x)\rangle | x \in B_k \setminus C_\delta\big]\Pr(x \in B_k \setminus C_\delta)$$

$$\overset{(a)}{\leq} \|\psi_1(x)\|_\infty \Pr(x \in B_k \setminus C_\delta) \leq \epsilon/4, \tag{42}$$

Moreover, for the case of $x \in C_\delta$, since in this case $\mathcal{J} \subseteq \mathcal{I}$, then we know that

$$\langle f_{k,1}^*(x), \psi_0(x)\rangle = \max_{j\in\mathcal{J}} (\psi_0(x))(j) \leq \max_{j\in\mathcal{I}} (\psi_0(x))(j) = \langle f_{k',1}^*(x), \psi_0(x)\rangle. \tag{43}$$

Next, using the third part of this lemma, we know that for $k' \leq k$ we have $\langle f_{k',1}^*(x), \psi_0(x)\rangle \leq \langle f_{k,1}^*(x), \psi_0(x)\rangle$, which together with (43) concludes that $\langle f_{k,1}^*(x), \psi_0(x)\rangle = \langle f_{k',1}^*(x), \psi_0(x)\rangle$. Next, because $\big(\psi_0(x) - k\psi_1(x)\big)(i) = \big(\psi_0(x) - k\psi_1(x)\big)(j)$ for $i, j \in \mathcal{J}$, then we know that $\big|\big(\ell_{k'}(x)\big)(i) - \big(\ell_{k'}(x)\big)(j)\big| = \big|(k - k')\big((\psi_1(x))(i) - (\psi_1(x))(j)\big)\big| \leq 2|k - k'|$. Therefore, if for $i, j \in \mathcal{J}$ we know that $\big(\psi_0(x)\big)(i) = \big(\psi_0(x)\big)(j)$, then the difference between $\psi_1$ for those indices is bounded as

$$\big|(\psi_1(x))(i) - (\psi_1(x))(j)\big| \leq \frac{1}{k}\big|(\psi_0(x))(i) - (\psi_0(x))(j)\big|$$
$$+ \big|(\ell_k(x))(i) - (\ell_k(x))(j)\big|$$
$$\leq 2|k - k'|. \tag{44}$$

Now, we know that because $x \in C_\delta$, then $\langle f_{k,1}^*(x), \psi_1(x)\rangle = (\psi_1(x))(i)$ for $i \in \operatorname*{argmax}_{j\in\mathcal{J}} (\psi_0(x))(j)$, and $\langle f_{k',1}^*(x), \psi_1(x)\rangle = (\psi_1(x))(j)$ for $j \in \operatorname*{argmax}_{k\in\mathcal{I}} (\psi_0(x))(j)$. Hence, we can see that $i \in \mathcal{J} \subseteq \mathcal{I}$ and $j \in \mathcal{I}$, and because $(\psi_0(x))(i) = \langle f_{k,1}^*(x), \psi_0(x)\rangle = \langle f_{k',1}^*(x), \psi_0(x)\rangle = (\psi_0(x))(j)$, and due to (44) we have

$$\big|\langle f_{k,1}^*(x), \psi_1(x)\rangle - \langle f_{k',1}^*(x), \psi_1(x)\rangle\big| \leq 2|k - k'|,$$

as long as $k' \in [k - \delta/2, k)$. Therefore, if we set $\delta = \max\{\delta', \epsilon/2\}$ we have

$$\big|\langle f_{k,1}^*(x), \psi_1(x)\rangle - \langle f_{k',1}^*(x), \psi_1(x)\rangle\big| \leq \epsilon/2,$$

and therefore

$$\Big|\mathbb{E}\big[\langle f_{k',1}^*(x), \psi_1(x)\rangle | x \in C_\delta\big] - \mathbb{E}\big[\langle f_{k,1}^*(x), \psi_1(x)\rangle | x \in C_\delta\big]\Big|$$

$$\leq \mathbb{E}\Big[\big\|\langle f_{k,1}^*(x), \psi_0(x)\rangle - \langle f_{k',1}^*(x), \psi_0(x)\rangle\big\|\Big] \leq \epsilon/2 \tag{45}$$

Finally, we can rewrite $D(k')$ as

$$D(k') = \mathbb{E}\big[\langle f^*_{k',1}(x), \psi_0(x)\rangle\big] = \mathbb{E}\big[\langle f^*_{k',1}(x), \psi_0(x)\rangle | x \in A_k\big] \Pr(x \in A_k)$$
$$+ \mathbb{E}\big[\langle f^*_{k',1}(x), \psi_0(x)\rangle | x \in B_k \setminus C_\delta\big] \Pr(x \in B_k \setminus C_\delta)$$
$$+ \mathbb{E}\big[\langle f^*_{k',1}(x), \psi_0(x)\rangle | x \in C_\delta\big] \Pr(x \in C_\delta),$$

and because of (42) and (45), and since the first term is a constant function in terms of $k'$ and for all $k' \in [k - \delta/2, k]$, then we have

$$|D(k') - D(k)| \leq \epsilon/4 + \epsilon/4 + \epsilon/2 = \epsilon. \tag{46}$$

This shows that $D(k')$ is lower semi-continuous around $k' = k$.

5. To prove this part, we first divide $\mathcal{X}$ into two subsets

$$G_{k'} = \Big\{ x \in \mathcal{X} : \big| \operatorname*{argmax}_{i}(\ell_{k'}(x))(i) \big| = 1 \Big\}, \tag{47}$$

and $H_{k'} = \mathcal{X} \setminus G_{k'}$. We know that for $x \in G_{k'}$ we have

$$f^*_{k',0}(x) = f^*_{k',1}(x) = \begin{cases} 1 & i = \min\{j \in \operatorname{argmax} \ell_{k'}(x)\} \\ 0 & \text{otherwise} \end{cases} \tag{48}$$

This concludes that

$$\mathbb{E}\big[\langle f^*_{k',0}(x), \psi_1(x)\rangle | x \in G_k\big] = \mathbb{E}\big[\langle f^*_{k',1}(x), \psi_1(x)\rangle | x \in G_k\big]. \tag{49}$$

Moreover, let us define the set $\Psi_1^{k'} = \{x \in \mathcal{X} : \exists c \in \mathbb{R}, \forall j \in \operatorname{argmax} \ell_{k'}(x), (\psi_1(x))(j) = c\}$. We show that sum of the probabilities of $H_{k'} \setminus \Psi_1^{k'}$ is always bounded by $2^d$ for a set of choices for $k'$, or equivalently

$$\sum_{k' \in \mathcal{K}} \Pr(x \in H_{k'} \setminus \Psi_1^{k'}) \leq 2^d, \tag{50}$$

for all finite or countably infinite choice of $\mathcal{K} \subseteq \mathbb{R}^+$. In fact, we know that for each instance $x$, $\operatorname*{argmax}_{j \in [1:d]} (\ell_k(x))(j)$ can take up to $2^d$ cases of all subsets of $\{1, \ldots, d\}$. Therefore, we need to show that there cannot exist two values of $k, k'$ such that for $x \in (H_k \setminus \Psi_1^k) \cap (H_{k'} \setminus \Psi_1^{k'})$ we have

$$\operatorname*{argmax}_{j} (\ell_k(x))(j) = \operatorname*{argmax}_{j} (\ell_{k'}(x))(j). \tag{51}$$

If we prove such identity, then due to pigeonhole principle, we have

$$\sum_{k' \in \mathcal{K}} \mathbb{1}_{x \in H_{k'} \setminus \Psi_1^{k'}} \leq 2^d, \tag{52}$$

which by integration over all values of $x$ concludes in (50). We prove this claim by contradiction. If we assume $k, k' \in \mathcal{K}$ such that for $x \in (H_k \setminus \Psi_1^k) \cap (H_{k'} \setminus \Psi_1^{k'})$ the identity (51) holds, then because $x \in H_k \cap H_{k'}$, then the size of $\operatorname*{argmax}_{j} (\ell_k(x))(j)$ and $\operatorname*{argmax}_{j} (\ell_{k'}(x))(j)$ is at least 2. This concludes that

$$(\psi_0(x) - k\psi_1(x))(i) = (\psi_0(x) - k\psi_1(x))(j)$$

as well as

$$(\psi_0(x) - k'\psi_1(x))(i) = (\psi_0(x) - k'\psi_1(x))(j)$$

for all choices of $i, j \in \operatorname{argmax} \ell_k(x)$. As a result, we have

$$(k - k')\big((\psi_1(x))(i) - (\psi_1(x))(j)\big) = 0,$$

and because $k' \neq k$, we have

$$\big(\psi_1(x)\big)(i) = \psi_1(x)\big)(j),$$

for all $i, j \in \arg\max \ell_k(x)$. Therefore, $x \in \Psi_1^{k'}$ and that is a contradiction.

Now that we know that the sum of the probabilities of $\Pr(x \in H_{k'} \setminus \Psi_1^{k'})$ is bounded, we can renormalize that and make a probability measure as

$$g(A) = \frac{\sum_{k \in A, \Pr(x \in H_k \setminus \Psi_1^k) > 0} \Pr(x \in H_k \setminus \Psi_1^k)}{\sum_{k : \Pr(x \in H_k \setminus \Psi_1^k) > 0} \Pr(x \in H_k \setminus \Psi_1^k)}. \tag{53}$$

Due to Lemma J.1, for all $\epsilon \geq 0$ we can find a small enough $\delta \geq 0$ such that $g([k - \delta, k)) \leq \epsilon/2^{d+1}$, and therefore for all $k' \in [k - \delta, k)$ we have

$$\Pr(x \in H_{k'} \setminus \Psi_1^{k'}) \leq \sum_{t \in [k-\delta, k), \Pr(x \in H_t \setminus \Psi_1^t) > 0]} \Pr(x \in H_t \setminus \Psi_1^t)$$

$$= g\big([k - \delta, k)\big) \sum_{k : \Pr(x \in H_k \setminus \Psi_1^k) > 0} \Pr(x \in H_k \setminus \Psi_1^k)$$

$$\leq \frac{\epsilon}{2^{d+1}} 2^d = \epsilon/2,$$

where the last inequality holds because of (50).

Now, using this and due to (49), and by defining $g_i(x) = \langle f_{k,i}^*(x), \psi_0(x) \rangle$ for $i = 1, 2$, we can bound the difference of $D(k)$ and $C(k)$ as

$$\big|D(k) - C(k)\big| = \Big|\mathbb{E}\big[g_1(x) - g_0(x)\big| x \in H_{k'}\big] \Pr(x \in H_{k'})\Big|$$

$$\leq \Pr(x \in H_{k'} \setminus \Psi_1^{k'} | x \in H_{k'}) \Big|\mathbb{E}\big[g_1(x) - g_0(x)\big| x \in H_{k'} \setminus \Psi_1^{k'}\big]\Big|$$

$$+ \Pr(x \in H_{k'} \cap \Psi_1^{k'} | x \in H_{k'}) \Big|\mathbb{E}\big[g_1(x) - g_0(x)\big| x \in H_{k'} \cap \Psi_1^{k'}\big]\Big|$$

$$\overset{(a)}{\leq} 2(\epsilon/2) + \Big|\mathbb{E}\big[g_1(x) - g_0(x)\big| x \in H_{k'} \cap \Psi_1^{k'}\big]\Big|$$

$$\overset{(b)}{=} \epsilon,$$

where $(a)$ holds because $\|f_{k,0}^* - f_{k,1}^*\|_1 \leq \|f_{k,0}^*\|_1 + \|f_{k,1}^*\|_1 = 2$ and because of Hölder inequality we have $\big|\langle f_{k,0}^*(x) - f_{k,1}^*, \psi_1(x) \rangle\big| \leq \|f_{k,0}^* - f_{k,1}^*\|_1 \|\psi_1(x)\|_\infty \leq 2$. Moreover, to show that $(b)$ holds we know that for $x \in \Psi_1^{k'}$ we have $\big(\psi_1(x)\big)(i) = \big(\psi_1(x)\big)(j)$ for all $i, j \in \arg\max \ell_{k'}(x)$. Therefore, because we know $g_0(x) = \big(\psi_1(x)\big)(i)$ for $i \in \arg\min_{j \in \arg\max_l (\ell_{k'}(x))(l)} \big(\psi_1(x)\big)(j) \subseteq \arg\max_l \big(\ell_{k'}(x)\big)(l)$ and $g_1(x) = \big(\psi_1(x)\big)(j)$ for $j \in \arg\max_{j \in \arg\max_l (\ell_{k'}(x))(l)} \big(\psi_0(x)\big)(j) \subseteq \arg\max_l \big(\ell_{k'}(x)\big)(l)$, we have $g_0(x) = g_1(x)$. The above inequality proves that the limit of $C(k')$ and $D(k')$ for $k' \uparrow k$ are equal and that completes the proof.

$\square$

To prove this theorem, we take the following steps: (i) We show that the set $\mathcal{K}$ has a non-negative member, (ii) we show that the prediction function $f_{k,p}^*(x)$ achieves the inequality constraint tightly, and by Theorem 4.1 we can conclude that $f_{k,p}^*(x)$ is the optimal solution.

- **step (i)**: It is easy to see that the Bayes optimal solution of the prediction function in (3) without any constraint is

$$\big(f^*(x)\big)(i) = \begin{cases} 1 & \big(\psi_0(x)\big)(i) > \big(\psi_0(x)\big)(j) \text{ for all } j \neq i \\ 0 & \big(\psi_0(x)\big)(i) < \max_j \big(\psi_0(x)\big)(j) \\ p_i(x) & \text{otherwise} \end{cases},$$

where $p_i(x) \in \Delta_d$ is an arbitrary vector. We can see that by setting

$$
\big(p_i(x)\big)(j) = \begin{cases} 1 & j = \min\{ \underset{t \in \mathrm{argmax}\, \ell_0(x)}{\mathrm{argmin}} \big(\psi_1(x)\big)(t)\} \\ 0 & \text{otherwise} \end{cases},
$$

then the two prediction functions $f^*(x)$ and $f^*_{0,0}(x)$ are equal (See statement of Theorem 4.2).

Now, in the first and second part of Lemma J.2 we have shown that $\mathbb{E}\big[\langle f^*_{k,0}(x), \psi_1(x)\rangle\big]$ is upper semi-continuous and monotonically non-increasing. Therefore, for all $k \in \mathbb{R}^+$ we have

$$
\mathbb{E}\big[\langle f^*_{k,0}(x), \psi_1(x)\rangle\big] \le \mathbb{E}\big[\langle f^*_{0,0}(x), \psi_1(x)\rangle\big] = \mathbb{E}\big[\langle f^*(x), \psi_1(x)\rangle\big].
$$

Similarly, we can show that for $k \to \infty$, the solution is equivalent to the Bayes minimizer of

$$
f^{**}(x) = \underset{f \in \Delta_d^{\mathcal{X}}}{\mathrm{argmin}}\, \mathbb{E}\big[\langle f(x), \psi_1(x)\rangle\big].
$$

Therefore, since $\delta$ is an interior point of all possible values, it lays on the interval $\big(\mathbb{E}\big[\langle f^{**}(x), \psi_1(x)\rangle\big], \mathbb{E}\big[\langle f^*(x), \psi_1(x)\rangle\big]\big)$, due to the montonicity and upper semi-continuity of $\mathbb{E}\big[\langle f^*_{k,0}, \psi_1(x)\rangle\big]$, we can find $t$ such that

$$
\mathbb{E}\big[\langle f^*_{t,0}(x), \psi_1(x)\rangle\big] \le \delta \le \lim_{\tau \uparrow t} \mathbb{E}\big[\langle f^*_{k,0}(x), \psi_1(x)\rangle\big]. \tag{54}
$$

Moreover, this $t$ should be a positive scalar, since otherwise we have

$$
\mathbb{E}\big[\langle f^*_{t,0}(x), \psi_1(x)\rangle\big] \ge \mathbb{E}\big[\langle f^*_{0,0}(x), \psi_1(x)\rangle\big] = \mathbb{E}\big[\langle f^*(x), \psi_1(x)\rangle\big] > \delta,
$$

which is a contradiction to (54).

- **step (ii)**: In this step, we consider the following two cases:
  - $C(t)$ **is continuous at** $t$: In this case, (54) is equivalent to $\delta = C(t) = \mathbb{E}\big[\langle f^*_{t,0}(x), \psi_0(x)\rangle\big]$, which means that the prediction function $f^*_{k,0}(x)$ achieves the constraint tightly, and therefore using Theorem 4.1 $f^*_{k,0}(x)$ is the optimal solution.
  - $C(t)$ **is discontinuous at** $t$: To show that we can achieve the highest constraint in this case, we first condition the constraint into two events $x \in G_k$ and $x \in \mathcal{X} \setminus G_k$, where $G_k$ is defined in (47). We know that in the latter case $x \in \mathcal{X} \setminus G_k$, the prediction function $f^*_{k,p}$ can be decomposed into two components

    $$
    f^*_{k,p}(x) = p f^*_{k,1}(x) + (1-p) f^*_{k,0}(x), \tag{55}
    $$

    while for $x \in G_k$ the prediction function $f^*_{k,p}(x) = f^*_{k,0}(x) = f^*_{k,1}(x)$ for all $p \in [0,1]$. Therefore, in both cases (55) holds, and we have

    $$
    \begin{aligned}
    \mathbb{E}\big[\langle f^*_{k,p}(x), \psi_1(x)\rangle\big] &= \mathbb{E}\big[\langle p f^*_{k,1}(x) + (1-p) f^*_{k,0}(x), \psi_1(x)\rangle\big] \\
    &= p\mathbb{E}\big[\langle f^*_{k,1}(x), \psi_1(x)\rangle\big] + (1-p)\mathbb{E}\big[\langle f^*_{k,0}(x), \psi_1(x)\rangle\big] \\
    &= pD(k) + (1-p)C(k), \tag{56}
    \end{aligned}
    $$

    where $C(\cdot)$ and $D(\cdot)$ are defined in Lemma J.2. Using this lemma, we know that $D(\cdot)$ is lower semi-continuous, and $\lim_{k' \uparrow k} C(k) = \lim_{k' \uparrow k} D(k)$. Therefore, together with (56) and the definition of $p$ in the statement of theorem, we have

    $$
    \begin{aligned}
    \mathbb{E}\big[\langle f^*_{k,p}(x), \psi_0(x)\rangle\big] =& p \lim_{k' \uparrow k} C(k') + (1-p)C(k) \\
    =& \frac{C(k) - c}{C(k) - \lim_{k' \uparrow k} C(k')} \lim_{k' \uparrow k} C(k') \\
    & + \frac{c - \lim_{k' \uparrow k} C(k')}{C(k) - \lim_{k' \uparrow k} C(k')} C(k) = c. \tag{57}
    \end{aligned}
    $$

    Equivalently, the prediction function achieves the constraint inequality tightly, and therefore by Theorem 4.1 this is sufficient to be the optimal solution to the constrained optimization problem.

# K  Proof of Theorem 5.1

Through the proof of this theorem, we use [6, Lemma 3.2.3] that implies that the class of multi-plications of $k$ binary functions $f_i(x)$ for $i \in [1 : k]$ within a hypothesis class with VC dimension $VC(f_i) = d$ itself has a VC dimension that is bounded as

$$\underbrace{VC(\{\prod_{i=1}^{k} f_i : f_i \in \mathcal{H}_i, VC(\mathcal{H}_i) = d\})}_{\mathcal{H}'} \leq 2dk \log 3k. \tag{58}$$

In fact, we use a simple extension to this lemma for which the VC dimension of the functions is not $d$ itself but is bounded above by $d$. In such case we claim that (58) still holds. The starting point for the proof to this lemma is bounding the size of the restriction $\Pi_{\mathcal{H}}(S) = |\{h \cap S : h \in \mathcal{H}\}|$ for the hypothesis class $\mathcal{H}$ by

$$\Pi_{\mathcal{H}}(S) \leq \left(\frac{em}{d}\right)^d, \tag{59}$$

where $VC(\mathcal{H}) = d$ and $m = |S|$. However, this inequality holds for the hypothesis classes that have VC dimensions that are bounded by $d$. The reason is increasingly monotonicity of RHS of (59). In fact, by obtaining the gradient of $\left(\frac{em}{d}\right)^d$ in terms of $d$ we have

$$\frac{\partial \left(\frac{em}{d}\right)^d}{\partial d} = \frac{\partial \left(e^{d \log em/d}\right)}{\partial d} = (\log em/d - 1)\left(\frac{em}{d}\right)^d,$$

which is nonnegative as long as $m \geq d$. If we particularly set $m^* = 2dk \log 3k$, then $m^* \geq d$ and therefore (59) holds. Next, similar to the proof of [6, Lemma 3.2.3], we can show that for the set $S$ with size $m^*$ we have

$$\Pi_{\mathcal{H}'}(S) \leq \Pi_{\mathcal{H}_1}^k(S) \leq \left(\frac{em^*}{d}\right)^{dk} \leq 2^{m^*},$$

which means that $S$ cannot be shattered by $\mathcal{H}'$, and therefore the VC dimension of this hypothesis class must be bounded by $m^*$.
We further use the following lemma:

$\square$

**Lemma K.1.** *For arbitrary sets of functions $\{\phi_1^i(x)\}_{i=1}^n$ and $\{\phi_2^i(x)\}_{i=1}^n$ on $\mathbb{R}$ and for a given $d \in \mathbb{R}$ the hypothesis class*

$$\mathcal{H} = \{\prod_{i=1}^{n} \text{sgn}\big(\phi_1^i(x) - k\phi_2^i(x) - d\big) : k \in \mathbb{R}\},$$

*has the VC dimension of at most* 4.

*Proof.* To prove this lemma, we show that the form of the product in the definition of $\mathcal{H}$ reduces to the form of an interval on $\mathbb{R}$, which is known to have VC dimension of 2. In fact, each term $\text{sgn}(\phi_1^i(x) - k\phi_2^i(x) - d)$ can be rewritten as

$$\text{sgn}(\phi_1^i(x) - k\phi_2^i(x) - d) = \text{sgn}(\tfrac{\phi_1^i(x)-d}{\phi_2^i(x)} - k)\text{sgn}(\phi_2^i(x)) + \text{sgn}(k - \tfrac{\phi_1^i(x)-d}{\phi_2^i(x)})\text{sgn}(-\phi_2^i(x))$$
$$+ \text{sgn}(\phi_1^i(x) - d)\mathbb{I}_{\phi_2^i(x)=0}.$$

As a result, by multiplying all terms we have

$$\prod_{i=1}^{n} \text{sgn}\big(\phi_1^i(x) - k\phi_2^i(x) - d\big) = \text{sgn}(\min_{i \in \mathcal{A}_x} \tfrac{\phi_1^i(x)-d}{\phi_2(x)} - k)\text{sgn}(k - \max_{i \in \mathcal{B}_x} \tfrac{\phi_1^i(x)-d}{\phi_2(x)}) \prod_{i \in \mathcal{C}_x} \text{sgn}(\phi_1^i(x) - d),$$
$$\tag{60}$$

where $\mathcal{A}_x$, $\mathcal{B}_x$, and $\mathcal{C}_x$ are defined as $\mathcal{A}_x = \{i \in [1 : n] : \phi_2^i(x) > 0\}$, $\mathcal{B}_x = \{i \in [1 : n] : \phi_2^i(x) < 0\}$, and $\mathcal{C}_x = \{i \in [1 : n] : \phi_2^i(x) = 0\}$. Now, we see that the first two terms define an interval for $k \in \big(f_1(x), f_2(x)\big)$ where $f_1(x) = \max_{i \in \mathcal{B}_x} \tfrac{\phi_1^i(x)-d}{\phi_2^i(x)}$ and $f_2(x) = \min_{i \in \mathcal{A}_x} \tfrac{\phi_1^i(x)-d}{\phi_2^i(x)}$. Next,

we prove that the VC dimension of the hypothesis class of all such functions is less than the VC dimension of $\mathcal{G} = \{f : \mathbb{R} \times \mathbb{R} \to \{0, 1\} : f(x, y) = \text{sgn}(x - k_1)\text{sgn}(k_2 - y), k_1, k_2 \in \mathbb{R}\}$. The reason is that if the aforementioned interval can shatter a set $\mathcal{S}$, then we can find the corresponding values of $f_1(x)$ and $f_2(x)$ for each $x \in \mathcal{S}$, and then form the pair $(x_i, y_i)$ where $x_i = f_1(x)$ and $y_i = f_2(x)$, and by setting $k_1 = k_2 = k$, we can shatter the set $\{(x_i, y_i)\}_{i=1}^{|\mathcal{S}|}$ with $\mathcal{G}$. Note that here all pairs are identical. The reason is that if not, i.e., if $f_1(x) = f_1(x')$ and $f_2(x) = f_2(x')$ for $x, x' \in \mathcal{S}$ and $x \neq x'$, then, for all possible $k$, we have $\text{sgn}(k - f_1(x))\text{sgn}(f_2(x) - k) = \text{sgn}(k - f_1(x'))\text{sgn}(f_2(x') - k)$, and therefore we cannot shatter $\mathcal{S}$ by $\text{sgn}(k - f_1(x))\text{sgn}(f_2(x) - k)$. Therefore, the set $\{(x_i, y_i)\}_{i=1}^{|\mathcal{S}|}$ has the same cardinality of $\mathcal{S}$, which in consequence proves that the VC dimension of all $\text{sgn}(k - f_1(x))\text{sgn}(f_2(x) - k)$ is bounded by $VC(\mathcal{G})$. Moreover, $VC(\mathcal{G}) \leq 4$, since for each 5 points in two-dimensional space, one is in the convex hull of the others, and in case that all others are labeled as 1, the one in the convex hull also must be labeled as 1. As a result, $\mathcal{G}$ cannot shatter 5 points, and therefore $VC(\mathcal{G}) \leq 4$.

Up to now, we have shown that the class of functions equal to the first two terms of (60) has a VC dimension that is bounded by 4. Next, we show that multiplying a hypothesis class $\mathcal{H}$ with a binary function $\phi(x)$ does not increase the VC dimension of that class. More formally, if we define

$$\mathcal{H} = \{\phi(x)f(x) : f \in \mathcal{H}'\},$$

then $VC(\mathcal{H}) \leq VC(\mathcal{H}')$. The reason is that if we can shatter a set $\mathcal{S}$ using $\mathcal{H}$, then for each member $x \in \mathcal{S}$ there exists two members $f_1, f_2$ of $\mathcal{H}'$ such that $f_1(x) = 1$ and $f_2(x) = 0$. This means that $\phi(x) \neq 0$, because otherwise $f_1(x) = 1$ would not be achievable. Therefore, $\phi(x) = 1$ for all $x \in \mathcal{S}$, and as a result similarly $\mathcal{H}'$ can shatter $\mathcal{S}$, which proves that $VC(\mathcal{H}) \leq VC(\mathcal{H}')$.

Finally, since we know that the class of all functions in $\mathcal{H}$ is in form of $\text{sgn}(k - f_1(x))\text{sgn}(f_2(x) - k)$ multiplied with a binary function, then we conclude that $VC(\mathcal{H}) \leq 4$. $\qquad \square$

To prove the rest of the theorem, we need to show that for all choices of $\hat{k}$ and $\hat{p}$ the difference of the empirical and the true loss is bounded. In fact, we should find a bound in form of

$$\Pr\left(\sup_{k,p} \left|\mathbb{E}_{S^n}\left[\langle f_{k,p}^*(x), \psi_0(x) \rangle\right] - \mathbb{E}_\mu\left[\langle f_{k,p}^*(x), \psi_0(x) \rangle\right]\right| \leq d_n\right) \geq 1 - \epsilon.$$

Here, we divide the class $\mathcal{X}$ into two subsets $G_k$ and $H_k = \mathcal{X} \setminus G_k$, where $G_k$ is defined in (47). Now, using the definition of $f_{k,p}^*(x)$, we know that within $G_k$, the inner-product $\langle f_{k,p}^*(x), \psi_1(x) \rangle$ can be rewritten as

$$\langle f_{k,p}^*(x), \psi_1(x) \rangle = \left(\psi_1(x)\right)\left(\underset{i}{\text{argmax}}\left(\ell_k(x)\right)(i)\right)$$

$$= \sum_{j=1}^{d} \left(\psi_1(x)\right)(j) \prod_{i \neq j} \text{sgn}\left(\left(\ell_k(x)\right)(j) - \left(\ell_k(x)\right)(i)\right)$$

$$= \sum_{j=1}^{d} \left(\psi_1(x)\right)(j) \underbrace{\prod_{i \neq j} \text{sgn}\left(\left(\psi_0(x)\right)(j) - \left(\psi_0(x)\right)(0) - k\left[\left(\psi_1(x)\right)(j) - \left(\psi_1(x)\right)(i)\right]\right)}_{\Phi_j^k(x)}.$$

Now, we can condition $x$ on being a member of $G_k$, and therefore the maximum difference between the two empirical and true expectation is as

$$\sup_{k,p} \left|\mathbb{E}_{S^n}\left[\langle f_{k,p}^*(x), \psi_1(x) \rangle \mid x \in G_k\right] - \mathbb{E}_\mu\left[\langle f_{k,p}^*(x), \psi_1(x) \rangle \mid x \in G_k\right]\right|$$

$$\leq \sum_{j=1}^{d} \sup_{k,p} \left|\mathbb{E}_{S^n}\left[\left(\psi_1(x)\right)(j) \cdot \Phi_j^k(x) \mid x \in G_k\right] - \mathbb{E}_\mu\left[\left(\psi_1(x)\right)(j) \cdot \Phi_j^k(x) \mid x \in G_k\right]\right|. \quad (61)$$

Now, we bound the inner term of (61) in a high probability setting. To that end, we use Rademacher's inequality in [66, Theorem 26.5], which shows that maximum difference between the expected value of a function $h \in \mathcal{H}$ over empirical distribution and the true distribution is $2R(\mathcal{H}) + 4c\sqrt{\frac{\ln 4/\epsilon}{n}}$ where $R(\mathcal{H})$ is the Rademacher's complexity of the class of function $\mathcal{H}$ and $c$ is maximum value that $h$ can take. By defining

$$h(x) := \left(\psi_1(x)\right)(j) \cdot \Phi_j^k(x),$$

we have $c = \|(\psi_1(x))(j)\|_\infty \leq 1$. Therefore, we have for all $h$,

$$\sup_{h \in \mathcal{H}} \mathbb{E}_{S^n}[h(x)] - \mathbb{E}_\mu[h(x)] \leq 2R(\mathcal{H}) + 4\sqrt{\frac{\ln 4d/\epsilon}{n}}, \tag{62}$$

with probability at least $1 - \frac{\epsilon}{d}$. Now, we can use contraction Lemma [66, Lemma 26.9] to show that since $\|(\psi_1(x))(j)\|_\infty \leq 1$, then $R(\mathcal{H}) \leq R(\mathcal{F})$, where $\mathcal{F} = \{\Phi_j^k(x), k \in \mathbb{R}\}$. Moreover, $\mathcal{F}$ contains functions that are all multiplication of $d - 1$ binary functions all in form of

$$\text{sgn}\Big((\psi_1(x))(j) - (\psi_1(x))(0) - k\big[(\psi_0(x))(j) - (\psi_0(x))(i)\big]\Big).$$

Lemma K.1 shows that the hypothesis class that contains products of all such function has a VC-dimension that is bounded by $4$. As a result, the Rademacher's complexity of $\mathcal{F}$ is bounded using [47, Corollary 3.8, Corollary 3.18] as

$$R(\mathcal{F}) \leq \sqrt{\frac{4 \log en/4}{n}},$$

and therefore together with (62) for all $h \in \mathcal{H}$ we have

$$\mathbb{E}_{S^n}[h(x)] - \mathbb{E}_\mu[h(x)] \leq 2\sqrt{\frac{4 \log en/4}{n}} + 4\sqrt{\frac{\ln 4d/\epsilon}{n}},$$

with probability at least $1 - \frac{\epsilon}{d}$. Hence, using (61) we have

$$\sup_{k,p} \left| \mathbb{E}_{S^n}\big[\langle f_{k,p}^*(x), \psi_1(x)\rangle \mid x \in G_k\big] - \mathbb{E}_\mu\big[\langle f_{k,p}^*(x), \psi_1(x)\rangle \mid x \in G_k\big] \right|$$

$$\leq 2d\sqrt{\frac{4 \log el/4}{l}} + 4d\sqrt{\frac{\ln 4d/\epsilon}{l}}, \tag{63}$$

with probability at least $1 - \epsilon$. In the last inequality, we used Bonferroni's inequality on $\epsilon/d$ bad events that each summand of (61) is not within the concentration bound.

Next, we consider the region $H_k$ in which there are at least two maximizer components of $\ell_k(x)$. In this case, by definition of $\hat{f}_{k,p}(x)$, among these maximizers, we choose the first maximizer of $\psi_0(x)$ with probability $p$ and the first minimizer of $\psi_1(x)$ with probability $1 - p$. Therefore, by condition on these cases, and if we define

$$E(k,p) := \left| \mathbb{E}_{S^n}\big[\langle \hat{f}_{k,p}(x), \psi_1(x)\rangle \mid x \in H_k\big] - \mathbb{E}_\mu\big[\langle \hat{f}_{k,p}(x), \psi_1(x)\rangle \mid x \in H_k\big] \right|, \tag{64}$$

then we have

$$\sup_{k,p} E(k,p) \leq \sup_{k,p} pE(k,1) + (1-p)E(k,0) \leq \sup_{k,p} E(k,1) + \sup_{k,p} E(k,0). \tag{65}$$

Now, to bound $E(k,1)$, we first rewrite the closed-form solution of $\hat{f}_{k,1}(x)$ as

$$\big(\hat{f}_{k,1}(x)\big)(i) = \text{sgn}\Big((\ell_k(x))(i) \geq \max_j (\ell_k(x))(j) - d\Big) \prod_{j<i} l_{ij}(x) \prod_{j>i} u_{ij}(x), \tag{66}$$

where $l_{ij}(x)$ and $u_{ij}(x)$ are defined as

$$l_{ij}(x) := 1 - \mathbb{I}_{\big(\psi_0(x)\big)(i) \leq \big(\psi_0(x)\big)(j)} \mathbb{I}_{\big(\ell_k(x)\big)(j) \geq \max_t \big(\ell_k(x)\big)(t)},$$

and

$$u_{ij}(x) := 1 - \mathbb{I}_{\big(\psi_0(x)\big)(i) < \big(\psi_0(x)\big)(j)} \mathbb{I}_{\big(\ell_k(x)\big)(j) \geq \max_t \big(\ell_k(x)\big)(t)},$$

respectively. Note that the only difference between the definition of $u_{ij}(x)$ and $l_{ij}(x)$ is that $u_{ij}(x)$ permits the equality of $(\psi_0(x))(i)$ with other components, while that is not the case for $l_{ij}(x)$. This difference lets us find the *first* component with the largest value of $\psi_0(x)$.

Now, we can rewrite $\text{sgn}\Big((\ell_k(x))(j) \geq \max_t (\ell_k(x))(t)\Big)$ as the product

$$\text{sgn}\Big((\ell_k(x))(j) \geq \max_t (\ell_k(x))(t) - d\Big) := \prod_{l \in [1:d]} \text{sgn}\Big((\ell_k(x))(j) \geq (\ell_k(x))(l)\Big).$$

As shown in Lemma K.1, the class of such function has VC dimension of at most $4$. Furthermore, multiplying a hypothesis class with a function such as $\text{sgn}\Big(\big(\psi_0(x)\big)(i) \geq \big(\psi_0(x)\big)(j)\Big)$ and $\text{sgn}\Big(\big(\psi_0(x)\big)(i) > \big(\psi_0(x)\big)(j)\Big)$ does not increase the VC dimension (See proof of Lemma K.1, and neither does negation. Therefore, in RHS of (66) we can count $d$ number of functions, each with a hypothesis class with the VC dimension of at most $4$, and therefore using the early discussions in this proof (58), $\big(\hat{f}_{k,1}(x)\big)(i)$ is within a function class with the VC dimension of at most $8d\log(3d)$. Therefore, similar to (63) in previous part, we can bound $\sup_{k,p} E(k,1)$ as

$$\sup_{k,p} E(k,1) \leq 2d\sqrt{\frac{8d\log(3d)\log\big(en/\big(8d\log(3d)\big)\big)}{n}}$$
$$+ 4d\sqrt{\frac{\ln 4d/\epsilon}{n}}, \tag{67}$$

for $l \geq 8d\log(3d)$ with probability at least $1 - \epsilon$.
We can similarly, show that $\sup_{k,p} E(k,0)$ is bounded as

$$\sup_{k,p} E(k,0) \leq 2d\sqrt{\frac{8d\log(3d)\log\big(en/\big((8n+8)\log(3d)\big)\big)}{n}}$$
$$+ 4d\sqrt{\frac{\ln 4d/\epsilon}{n}}, \tag{68}$$

Therefore, using (63), (64), (65), (67), (68), and the application Bonferonni's inequality we have

$$\sup_{k,p} \Big|\mathbb{E}_{S^n}\big[\langle f^*_{k,p}(x), \psi_0(x)\rangle\big] - \mathbb{E}_\mu\big[\langle f^*_{k,p}(x), \psi_0(x)\rangle\big]\Big|$$
$$\leq 6d\sqrt{\frac{8d\log(3d)\log\frac{el}{(8n+8)\log(3d)}}{l}} + 12d\sqrt{\frac{\ln\frac{12d}{\epsilon}}{l}} \tag{69}$$
$$:= d_n(\epsilon), \tag{70}$$

with probability at least $1 - \epsilon$. Therefore, by assuming $\mathbb{E}_{S^n}\big[\langle f^*_{k,p}(x), \psi_1(x)\rangle\big] \leq \alpha - d_n(\epsilon)$, we assure that $\mathbb{E}_\mu\big[\langle f^*_{k,p}(x), \psi_1(x)\rangle\big] \leq \alpha$, with probability at least $1 - \epsilon$, and this completes the proof.

## L   Proof of Theorem 5.3

We first introduce three lemmas that are useful in proving this theorem.

**Lemma L.1.** *If $\delta$ is an $\epsilon$-interior point of the set $\mathcal{C} = \big\{\mathbb{E}_\mu\big[\langle f(x), \psi_1(x)\rangle\big] : f \in \Delta_d^{\mathcal{X}}\big\}$, then $\delta$ is $(\epsilon/2)$-interior point of $\mathcal{D} = \big\{\mathbb{E}_{S^n}\big[\langle f(x), \psi_1(x)\rangle\big] : f \in \Delta_d^{\mathcal{X}}\big\}$ with probability $1 - 2e^{-\frac{l\epsilon^2}{4}}$.*

*Proof.* The proof of this lemma is a direct application of Hoeffding's inequality. In fact, for $\|\psi_1\|_\infty \leq C$ that inequality together with Hölder's inequality imply that

$$\Pr\left(\Big|\mathbb{E}_\mu\big[\langle f(x), \psi_1(x)\rangle\big] - \mathbb{E}_{S^n}\big[\langle f(x), \psi_1(x)\rangle\big]\Big| \geq \epsilon/2\right) \leq e^{-\frac{n\epsilon^2}{4C^2}}.$$

Therefore, if there exists $f_1$ such that $\mathbb{E}_\mu\big[\langle f_1(x), \psi_1(x)\rangle\big] = \epsilon$, then with probability at least $1 - e^{-\frac{n\epsilon^2}{4C^2}}$ we have $\mathbb{E}_{S^n}\big[\langle f_1(x), \psi_1(x)\rangle\big] \in [\epsilon/2, 3\epsilon/2]$. Similarly, if $f_2$ exists such that $\mathbb{E}_\mu\big[\langle f_1(x), \psi_1(x)\rangle\big] = -\epsilon$, then with probability $1 - e^{-\frac{n\epsilon^2}{4C^2}}$ we have $\mathbb{E}_{S^n}\big[\langle f_2(x), \psi_1(x)\rangle\big] \in [-3\epsilon/2, -\epsilon/2]$. As a result of Bonferroni's inequality, with probability at least $1 - 2e^{-\frac{n\epsilon^2}{4C^2}}$ both these events happen, and because of the convexity of the set $\mathcal{D}$ we can say that with such probability all values between $a_0 \in [-3\epsilon/2, -\epsilon/2]$ and $a_1 \in [\epsilon/2, 3\epsilon/2]$ are in $\mathcal{D}$ too. This, of course at least contains the interval $[-\epsilon/2, \epsilon/2]$. $\qquad\square$

**Lemma L.2.** *Assume that we have an approximation $\hat{\psi}_1(x)$ of $\psi_1(x)$ with the error bounded as $\|\hat{\psi}_1(x) - \psi_1(x)\|_\infty \leq \epsilon$. Further let $\epsilon' \in \mathbb{R}^+$ such that $\epsilon' \geq \epsilon$. Now, if for $\sigma \in \{-\epsilon', \epsilon'\}$ there exists a rule $f \in \Delta_d^{\mathcal{X}}$ such that $\mathbb{E}_\mu\big[\langle f(x), \psi_1(x)\rangle\big] = \delta + \sigma$, then there exists $k \in \mathbb{R}$ as well as $p \in [0, 1]$ such that $\mathbb{E}_\mu\big[\langle \hat{f}_{k,p}(x), \hat{\psi}_1(x)\rangle\big] = \delta + \frac{\epsilon'-\epsilon}{2}$.*

*Proof.* Firstly, because of Hölder's inequality we know that

$$\left| \mathbb{E}_\mu\big[\langle f(x), \psi_1(x)\rangle\big] - \mathbb{E}_\mu\big[\langle f(x), \hat{\psi}_1(x)\rangle\big] \right| \leq \epsilon \|f_{k,p}^*(x)\|_1 = \epsilon,$$

for all $f \in \Delta_d^{\mathcal{X}}$. Therefore, by setting $\sigma = \epsilon'$ and $\sigma = -\epsilon'$, we can show that for $f_1 \in \Delta_d^{\mathcal{X}}$ such that

$$\mathbb{E}_\mu\big[\langle f_1(x), \psi_1(x)\rangle\rangle\big] = \delta + \epsilon',$$

then

$$\mathbb{E}_\mu\big[\langle f_1(x), \hat{\psi}_1(x)\rangle\big] \geq \delta + \epsilon' - \epsilon,$$

and where for $f_2 \in \Delta_d^{\mathcal{X}}$

$$\mathbb{E}_\mu\big[\langle f_2(x), \psi_1(x)\rangle\rangle\big] = \delta - \epsilon',$$

then

$$\mathbb{E}_\mu\big[\langle f_2(x), \hat{\psi}_1(x)\rangle\big] \leq \delta - \epsilon' + \epsilon.$$

Now, because of step (iii) of the proof of Theorem 4.1, we know that the set of constraints for all rules within $\Delta_d^{\mathcal{X}}$ is convex. Therefore, since we can achieve two points $f_1, f_2$ such that the constraint $\mathbb{E}_\mu\big[\langle f_i(x), \hat{\psi}_1(x)\rangle\big]$ can achieve two points above $\delta + \epsilon' - \epsilon$ and below $\delta - \epsilon' + \epsilon$, then for each $c \in [\delta - \epsilon' + \epsilon, \delta + \epsilon' - \epsilon]$ there exists $f \in \Delta_d^{\mathcal{X}}$ such that $\mathbb{E}_\mu\big[\langle f(x), \hat{\psi}_1(x)\rangle\big] = c$. Now, let $c = \delta + \frac{\epsilon' - \epsilon}{2}$. In the following, we show that there exists $k \in \mathbb{R}$ and $p \in [0,1]$ such that further $\mathbb{E}_\mu\big[\langle \hat{f}_{k,p}(x), \hat{\psi}_1(x)\rangle\big] = c$.

To that end, we first remind that Lemma J.2 shows that $\mathbb{E}_\mu\big[\langle \hat{f}_{k,0}(x), \hat{\psi}_1(x)\rangle\big]$ is monotonically non-increasing in terms of $k$. We show that for $k \in \mathbb{R}^-$ we have $\max \hat{\psi}_1(x) - \langle \hat{f}_{k,0}(x), \hat{\psi}_1(x)\rangle \leq -\frac{2}{k}$. The reason is that if $j \in \underset{l}{\arg\max} \big(\hat{\psi}_0(x) - k\hat{\psi}_1(x)\big)(l)$ and $j' \in \underset{l}{\arg\max} \big(\hat{\psi}_1(x)\big)(l)$, then we have

$$\big(\hat{\psi}_0(x) - k\hat{\psi}_1(x)\big)(j) \geq \big(\hat{\psi}_0(x) - k\hat{\psi}_1(x)\big)(j'),$$

which concludes that

$$-k\big[\big(\hat{\psi}_1(x)\big)(j) - \big(\hat{\psi}_1(x)\big)(j')\big] \geq \big(\hat{\psi}_0(x)\big)(j') - \big(\hat{\psi}_0(x)\big)(j) \geq -2.$$

Therefore, since

$$\mathbb{E}_\mu\big[\langle \arg\max \hat{\psi}_1(x), \hat{\psi}_1(x)\rangle\big] = \max_{f \in \Delta_d^{\mathcal{X}}} \mathbb{E}_\mu\big[\langle f(x), \hat{\psi}_1(x)\rangle\big] \geq \delta + \epsilon' - \epsilon,$$

where the last inequality holds due to the existence of $f_1$, then for $k \leq -8/(\epsilon' - \epsilon)$ we have

$$\mathbb{E}_\mu\big[\langle \hat{f}_{k,0}(x), \hat{\psi}_1(x)\rangle\big] \geq \delta + \epsilon' - \epsilon - \frac{2}{-8/(\epsilon' - \epsilon)} \geq \delta + 3\frac{\epsilon' - \epsilon}{4}.$$

Similarly, if we let $k \geq 8/(\epsilon' - \epsilon)$ we can prove that

$$\mathbb{E}_\mu\big[\langle \hat{f}_{k,0}(x), \hat{\psi}_1(x)\rangle\big] \leq \delta - \epsilon' + \epsilon + 2l \leq \delta - 3\frac{\epsilon' - \epsilon}{4}.$$

As a result, the set $\mathcal{C} = \{k : \mathbb{E}_\mu\big[\langle \hat{f}_{k,0}(x), \hat{\psi}_1(x)\rangle\big] \geq c\}$ is non-empty and bounded below by $-\frac{8}{\epsilon' - \epsilon}$. Therefore, its infimum exists and is also bounded below by $-\frac{8}{\epsilon' - \epsilon}$. Let us name that infimum $\hat{k}$. Now, if $\mathbb{E}_\mu\big[\langle \hat{f}_{k,0,0}(x), \hat{\psi}_1(x)\rangle\big]$ is continuous at $k = \hat{k}$, then we can show that $\mathbb{E}_\mu\big[\langle \hat{f}_{\hat{k},0}(x), \hat{\psi}_1(x)\rangle\big] = c$. If not, then as shown in step (ii) of the proof of Theorem 4.1, and in particular in (57), there exists $p$ such that $\mathbb{E}_\mu\big[\langle \hat{f}_{\hat{k},p}(x), \hat{\psi}_1(x)\rangle\big] = c$. This completes the proof. $\square$

**Lemma L.3.** *If* $\|\hat{\psi}_0 - \psi_0\|_\infty \leq \delta_0$ *and* $\|\hat{\psi}_1 - \psi_1\|_\infty \leq \delta_1$, *and for* $k \in [-K, K]$, *and* $k' \leq k - \frac{2(\delta_0 + K\delta_1)}{T}$ *for* $T \in \mathbb{R}^+$, *then we have*

$$\mathbb{E}\big[\langle \hat{f}_{k,0,0}(x) - f_{k',0}^*(x), \psi_1(x)\rangle\big] \leq T.$$

*Proof.* The proof of this lemma bears similarity to that of Lemma J.2. Here too, we define $\hat{\ell}_k(x) = \hat{\psi}_0(x) - k\hat{\psi}_1(x)$. Next, we have

$$\hat{f}_{k,0}(x) = \begin{cases} 1 & i = \min\{\underset{i \in \underset{l}{\operatorname{argmax}} \left(\hat{\ell}_k(x)\right)(l)}{\operatorname{argmin}} \hat{\psi}_1(x)\} \\ 0 & \text{otherwise} \end{cases}. \tag{71}$$

Next, we need to show that $(\psi_1(x))(j_1) = \langle r_{k',0}(x), \psi_1(x)\rangle \geq \langle \hat{f}_{k,0,0}(x), \psi_0(x)\rangle - T = (\psi_0(x))(j_2) - T$. Assume otherwise, meaning that $(\psi_1(x))(j_1) < (\psi_1(x))(j_2) - T$. In this case, we have

$$\begin{aligned}
\max \hat{\ell}_k(x) &\overset{(a)}{=} \left(\hat{\ell}_k(x)\right)(j_2) \\
&= \left(\ell_k(x)\right)(j_2) + \left(\hat{\psi}_0(x) - \psi_0(x)\right)(j_2) - k\left(\hat{\psi}_1(x) - \psi_1(x)\right)(j_2) \\
&= \left(\ell_{k'}(x)\right)(j_2) - (k - k')\left(\psi_1(x)\right)(j_2) + \left(\hat{\psi}_0(x) - \psi_0(x)\right)(j_2) - k\left(\hat{\psi}_1(x) - \psi_1(x)\right)(j_2) \\
&\overset{(b)}{\leq} \left(\ell_{k'}(x)\right)(j_2) - (k - k')\left(\psi_1(x)\right)(j_2) + (\delta_0 + K\delta_1) \\
&\overset{(c)}{<} \left(\ell_{k'}(x)\right)(j_2) - (k - k')\left(\psi_1(x)\right)(j_1) - (k - k')T + (\delta_0 + K\delta_1) \\
&\overset{(d)}{\leq} \left(\ell_{k'}(x)\right)(j_1) - (k - k')\left(\psi_1(x)\right)(j_1) - (k - k')T + (\delta_0 + K\delta_1) \\
&\overset{(e)}{\leq} \left(\ell_{k'}(x)\right)(j_1) - (k - k')\left(\psi_1(x)\right)(j_1) - 2\frac{\delta_0 + K\delta_1}{T}T + (\delta_0 + K\delta_1) \\
&= \left(\ell_{k'}(x)\right)(j_1) - (k - k')\left(\psi_1(x)\right)(j_1) - (\delta_0 + K\delta_1) \\
&= \left(\ell_k(x)\right)(j_1) - (\delta_0 + K\delta_1) \\
&= \left(\hat{\ell}_k(x)\right)(j_1) - (\delta_0 + K\delta_1) - \left(\hat{\psi}_0(x) - \psi_0(x)\right)(j_1) + k\left(\hat{\psi}_1(x) - \psi_1(x)\right)(j_1) \\
&\overset{(f)}{\leq} \left(\hat{\ell}_k(x)\right)(j_1) - (\delta_0 + K\delta_1) + (\delta_0 + K\delta_1) = \left(\hat{\ell}_k(x)\right)(j_1),
\end{aligned}$$

which is a contradiction. Note that $(a)$ holds because of definition of $j_2$ and (71), $(b)$ holds due to approximation assumptions $\|\hat{\psi}_0 - \psi_0\|_\infty \leq \delta_0$ and $\|\hat{\psi}_1 - \psi_1\|_\infty \leq \delta_1$, $(c)$ holds because of the assumption $(\psi_1(x))(j_1) < (\psi_1(x))(j_2) - T$, $(d)$ is followed by the definition of $j_1$ on maximizing $\ell_{k'}(x)$, and $(e)$ holds because $k \geq k' + \frac{2(\delta_0 + K\delta_1)}{T}$, and $(f)$ is followed by approximation assumptions. $\qquad\square$

We first formally express Theorem 5.3 as following:

**Theorem L.4.** *Assume that* $(\delta - \epsilon_l, \delta + \epsilon_u)$ *is a subset of of all achievable constraints* $\mathbb{E}\left[\langle f(x), \psi_1(x)\rangle\right]$, *and that* $\|\psi_i(x)\|_\infty \leq 1$ *for* $i = 1, 2$. *Further, let the size* $n$ *of validation data be large enough such that* $d_n(\delta/3) \leq \frac{\epsilon_l}{2}$. *Now, if the optimal predictor* $f_{k,0}^*(x)$ *is* $(\gamma, \Delta)$-sensitive around optimal* $k^*$ *for* $\Delta \geq \frac{\left(2\max\{d_n(\delta/3), \delta_1\} + \sqrt{2\gamma C(\delta_0 + K\delta_1)}\right)^{1/\gamma}}{C}$ *and* $\gamma \leq 1$, *then for* $n \geq \frac{16}{\epsilon_l^2}\log\frac{3}{\delta}$, *and with probability at least* $1 - \delta$, *the optimal empirical classifier, as of Algorithm 1 has an objective that is at most* $D_0$-*far from the true optimal objective where* $D_0$ *is defined as*

$$\mathbb{E}\left[\langle f_{k^*, p^*}^*(x), \psi_0(x)\rangle\right] - \mathbb{E}\left[\langle \hat{f}_{\hat{k}, \hat{p}}(x), \psi_0(x)\rangle\right] \leq 2\left(\frac{2\max\{d_n(\delta/3), \delta_0\}}{C}\right)^{1/\gamma} + 4\sqrt{\frac{2(\delta_0 + K\delta_1)}{\gamma C}} \\
+ 2(\delta_0 + K\delta_1) + 2Kd_n(\delta/3), \tag{72}$$

*where* $K$ *is an upper-bound to the absolute value of* $k^*$.

In order to prove this theorem, we first define a measure of distance between two rules $f_1, f_2 \in \Delta_d^{\mathbb{R}}$ as

$$D_k(f_1, f_2) := \mathbb{E}\left[\langle f_1(x) - f_2(x), \psi_0(x) - k\psi_1(x)\rangle\right]. \tag{73}$$

Using this measure of distance, the difference of objectives between two rules $f_1$ and $f_2$ can be written as

$$\mathbb{E}\big[\langle f_1(x), \psi_0(x)\rangle\big] - \mathbb{E}\big[\langle f_2(x), \psi_0(x)\rangle\big] = D_{k^*}(f_1, f_2) \\ + k^*\Big(\mathbb{E}\big[\langle f_1(x), \psi_1(x)\rangle\big] - \mathbb{E}\big[\langle f_2(x), \psi_1(x)\rangle\big]\Big). \quad (74)$$

Therefore, if two rules achieve similar constraints, and if $D_k(f_1, f_2)$ is small enough, we can prove that the two rules achieve similar objectives too, since $k$ is bounded above by $K$.

In fact, if we let $f_1(x) = f^*_{k,p}(x)$ and $f_2(x) := \hat{f}_{\hat{k},\hat{p}}$, where $k$ and $p$ are optimal solutions as in Theorem 4.2, then due to this optimality, and because $\mathbb{E}\big[\langle \hat{f}_{\hat{k},\hat{p}}, \psi_1(x)\rangle\big] \leq \delta$ with probability at least $1 - \epsilon$ as shown in Theorem 5.1, then LHS of (74) is positive with at least the same probability. In this proof, we show that how large is that term, and therefore, we show that how sub-optimal is $\hat{f}_{\hat{k},\hat{p}}$ in terms of the objective.

To that end, we first bound the difference between constraints. This bound can be achieved similar to the proof of Theorem 5.1. In fact, there we showed that if the empirical constraint $\mathbb{E}_{S^n}\big[\langle \hat{f}_{\hat{k},\hat{p}}, \psi_1(x)\rangle\big] \leq \delta - d_n(\pi)$, then using (69) the true expectation is bounded as $\mathbb{E}_\mu\big[\langle \hat{f}_{\hat{k},\hat{p}}, \psi_1(x)\rangle\big] \leq \delta$ with probability at least $1 - \pi$. However, (69) is symmetric in empirical and true constraint, i.e., if we show that $\mathbb{E}_{S^n}\big[\langle \hat{f}_{\hat{k},\hat{p}}, \psi_1(x)\rangle\big] \geq \delta - d_n(\pi)$, then we have $\mathbb{E}_\mu\big[\langle \hat{f}_{\hat{k},\hat{p}}, \psi_1(x)\rangle\big] \geq \delta - 2d_n(\pi)$ with probability at least $1 - \pi$.

To show $\mathbb{E}_{S^n}\big[\langle \hat{f}_{\hat{k},\hat{p}}, \psi_1(x)\rangle\big] \geq \delta - d_n(\pi)$, we follow three steps, (i) because $\delta$ is $(\epsilon_l, \epsilon_u)$-interior point of the set of constraints, i.e., $(\delta - \epsilon_l, \delta + \epsilon_u)$ is a subset of all plausible constraints, then $\delta - d_n(\pi)$ is $(\epsilon_l - d_n(\pi), \epsilon_u + d_n(\pi))$-interior point. Now, using Lemma L.1 and by setting $\epsilon' = \min\{\epsilon_l - d_n(\pi), \epsilon_u + d_n(\pi)\}$ we can show that $\delta - d_n(\pi)$ is $\epsilon'/2$-interior point of the empirical constraints with probability at least $1 - 2e^{-\frac{n\epsilon'^2}{4}}$, (ii) using the first step and assuming $d_n(\pi) \leq \epsilon_l/2$ we conclude that $\delta - d_n(\pi)$ is $d_n(\pi)/2$-interior point of the empirical constraints with the aforementioned probability, (iii) because of Lemma L.2, we conclude that for $\epsilon = d_n(\pi)/2$, and with probability at least $1 - 2e^{-\frac{n\epsilon'^2}{4}}$ there exists $k \in \mathbb{R}$ and $p \in [0,1]$ such that $\mathbb{E}_{S^n}\big[\langle \hat{f}_{k,p}(x), \hat{\psi}_1(x)\rangle\big] = \delta - d_n(\pi) + \frac{d_n(\pi)/2 - \epsilon}{2} = \delta - d_n(\pi)$. As a result of the above discussion we conclude that with probability at least $1 - \pi - 2e^{-\frac{n\epsilon'^2}{4}}$ there exists $k$ and $p$ such that $\delta \geq \mathbb{E}\big[\langle \hat{f}_{k,p}(x), \psi_1(x)\rangle\big] \geq \delta - 2d_n(\pi)$. Now, since we know that $\mathbb{E}\big[\langle f_1(x), \psi_1(x)\rangle\big] = \mathbb{E}\big[\langle f^*_{k,p}(x), \psi_1(x)\rangle\big] = \delta$, then we have

$$0 \leq \mathbb{E}\big[\langle f_1(x), \psi_1(x)\rangle\big] - \mathbb{E}\big[\langle f_2(x), \psi_1(x)\rangle\big] \leq 2d_l(\pi), \quad (75)$$

with probability at least $1 - \pi - 2e^{-\frac{n\epsilon'^2}{4}}$.

The above discussion together with (74) and the assumption of boundedness of $k$ shows that the difference of objectives is bounded with a high probability, if we bound $D_k(f_1, f_2)$. However, before we proceed with bounding that term, we should derive a relationship between $\hat{k}$ and $k^*$ for the reasons that we see in proving boundedness of $D_k(f_1, f_2)$.

We have already shown that there exists $\hat{p} \in [0,1]$ such that $\delta \geq \mathbb{E}\big[\langle \hat{f}_{\hat{k},\hat{p}}, \psi_1(x)\rangle\big] \geq \delta - 2d_l(\pi)$. Here, Lemma L.3 shows that for $k' = k - \frac{2(\delta_0 + K\delta_1)}{T}$ we have $\mathbb{E}\big[\langle f^*_{k',0}, \psi_1(x)\rangle\big] \geq \delta - 2d_l(\pi) - T$ with probability at least $1 - \pi - 2e^{-\frac{n\epsilon'^2}{4}}$. Moreover, using symmetry in Lemma L.3 and for $k'' = k + \frac{2(\delta_0 + K\delta_1)}{T}$ we have $\mathbb{E}\big[\langle f^*_{k'',0}(x) - \hat{f}_k(x), \hat{\psi}_1(x)\rangle\big] \leq T$. Now, since $\|\psi_1(x) - \hat{\psi}_1(x)\|_\infty \leq \delta_0$, using Hölder's inequality we conclude that $\mathbb{E}\big[\langle f^*_{k'',0}(x) - \hat{f}_k(x), \hat{\psi}_1(x)\rangle\big] \leq T + 2\delta_1$, and consequently $\mathbb{E}\big[\langle f^*_{k'',0}(x), \hat{\psi}_1(x)\rangle\big] \leq \delta + T + 2\delta_0$

Now that we have found a lower-bound on constraint of the rule $f^*_{k-q}(x)$ for $q = \frac{2(\delta_0 + K\delta_1)}{T}$, then if we find an upper bound on the constraint of the rule $f^*_{k^*+e}(x)$ for an $e \in \mathbb{R}^+$, then we can use monotonicity of the constraint of $f^*_k$ in terms of $k$ and prove a relationship between $k$ and $k^*$. To that end, we use detection assumption with which we can show that

$$\mathbb{E}\big[\langle f^*_{k^* + \frac{1}{C}(2d_n(\pi) + T)^{1/\gamma}}, \psi_1(x)\rangle\big] \leq \delta - 2d_n(\pi) - T,$$

where we assume that $d_n(\pi) \leq \frac{(C\Delta)^\gamma - T}{2}$. Now, using previous discussions conclude that

$$\mathbb{E}\big[\langle f^*_{k^* + \frac{1}{C}(2d_n(\pi)+T)^{1/\gamma}}, \psi_1(x)\rangle\big] \leq \mathbb{E}\big[\langle f^*_{k',0}, \psi_1(x)\rangle\big],$$

with probability at least $1 - \pi - 2e^{-\frac{n\epsilon'^2}{4}}$. This together with the first part of Lemma J.2 shows that $k' \leq k^* + \frac{1}{C}(2d_n(\pi) + T)^{1/\gamma}$, or equivalently $k \leq k^* + \frac{2(\delta_0 + K\delta_1)}{T} + \frac{1}{C}(2d_n(\pi) + T)^{1/\gamma}$ with probability at least $1 - \pi - 2e^{-\frac{n\epsilon'^2}{4}}$. Since we know that $\gamma$ is clamped above by 1, and using the inequality $(1+x)^a \leq 1 + ax$ for $a \geq 1$ we can substitute the above inequality with $k \leq k^* + \frac{2(\delta_0 + K\delta_1)}{T} + \frac{\big(2d_n(\pi)\big)^{1/\gamma}}{C}\big(1 + \frac{T}{\gamma(2d_n(\pi))^{1/\gamma}}\big)$. Now optimizing over $T$ leads in $T = \sqrt{2\gamma C(\delta_0 + K\delta_1)}$, which concludes that $k \leq k^* + \Delta_u k$ with the aforementioned probability, where $\Delta_u k = \frac{\big(2d_n(\pi)\big)^{1/\gamma}}{C} + 2\sqrt{\frac{2(\delta_0 + K\delta_1)}{\gamma C}}$, if we have $d_n(\pi) \leq \frac{(C\Delta)^\gamma - \sqrt{2\gamma C(\delta_0 + K\delta_1)}}{2}$

Similarly, using sensitivity assumption, we have

$$\mathbb{E}\big[\langle f^*_{k^* + \frac{1}{C}(2\delta_1 + T)^{1/\gamma}}(x), \psi_1(x)\rangle\big] \geq \delta + 2\delta_1 + T,$$

where $\frac{(2\delta_1 + T)^{1/\gamma}}{C} \leq \Delta$. Next, using previous discussions conclude that

$$\mathbb{E}\big[\langle f^*_{k^* + \frac{1}{C}(2\delta_1 + T)^{1/\gamma}}(x), \psi_1(x)\rangle\big] \geq \mathbb{E}\big[\langle f^*_{k'',0}(x), \psi_1(x)\rangle\big],$$

with the aforementioned probability. This, again, together with the first part of Lemma J.2 shows that $k'' \geq k^* - \frac{1}{C}(2\delta_1 + T)^{1/\gamma}$, or equivalently $k \geq k^* - \frac{1}{C}(2\delta_1 + T)^{1/\gamma} - \frac{2(\delta_0 + K\delta_1)}{T}$. Therefore, by setting $T = \sqrt{2\gamma C(\delta_0 + K\delta_1)}$ we conclude that $k \geq k^* - \Delta_l k$ where $\Delta_l k = \frac{(2\delta_1)^{1/\gamma}}{C} + 2\sqrt{\frac{2(\delta_0 + K\delta_1)}{\gamma C}}$, and assuming $\frac{\big(2\delta_1 + \sqrt{2\gamma C(\delta_0 + K\delta_1)}\big)^{1/\gamma}}{C} \leq \Delta$.

Next, we turn into bounding $D_{k^*}(f_1, f_2)$. To that end, we first note that

$$t_x(k^*) := \langle f^*_{k^*, p}(x), \psi_0(x) - k^* \psi_1(x)\rangle = \max_i \big(\psi_0(x) - k^*\psi_1(x)\big)(i), \tag{76}$$

for all $p \in [0,1]$. This is followed by the definition of $f^*_{k^*, p}(\cdot)$. Similarly, we can show that

$$\hat{t}_x(\hat{k}) := \langle \hat{f}_{\hat{k}, p}(x), \hat{\psi}_0(x) - k^* \hat{\psi}_1(x)\rangle = \max_i \big(\hat{\psi}_0(x) - \hat{k}\hat{\psi}_1(x)\big)(i),$$

for all $p \in [0,1]$. Now, we can rewrite $D_{k^*}(f_1, f_2)$ as

$$\begin{aligned}
D_{k^*}(f_1, f_2) &= \mathbb{E}\big[\langle f^*_{k^*, p}(x) - \hat{f}_{\hat{k}, \hat{p}}(x), \psi_0 - k^*\psi_1(x)\rangle\big] \\
&= \mathbb{E}[t_x(k^*)] - \mathbb{E}\big[\langle \hat{f}_{\hat{k}, \hat{p}}(x), \psi_0 - k^*\psi_1(x)\rangle\big] \\
&= \mathbb{E}[t_x(k^*)] - \mathbb{E}\big[\langle \hat{f}_{\hat{k}, \hat{p}}(x), \hat{\psi}_0 - k^*\hat{\psi}_1(x)\rangle\big] \\
&\quad - \mathbb{E}\big[\langle \hat{f}_{\hat{k}, \hat{p}}(x), (\psi_0(x) - \hat{\psi}_0(x)) - k^*(\psi_1(x) - \hat{\psi}_1(x))\rangle\big] \\
&\overset{(a)}{\leq} \mathbb{E}[t_x(k^*)] - \mathbb{E}\big[\langle \hat{f}_{\hat{k}, \hat{p}}(x), \hat{\psi}_0 - k^*\hat{\psi}_1(x)\rangle\big] + \delta_0 + K\delta_1 \\
&= \mathbb{E}[t_x(k^*)] - \mathbb{E}[\hat{t}_x(\hat{k})] + (k^* - k)\mathbb{E}\big[\langle \hat{f}_{\hat{k}, \hat{p}}(x), \hat{\psi}_0(x)\rangle\big] + \delta_0 + K\delta_1 \\
&\overset{(b)}{\leq} \mathbb{E}[t_x(k^*)] - \mathbb{E}[\hat{t}_x(\hat{k})] + |k^* - \hat{k}| + \delta_0 + K\delta_1, \tag{77}
\end{aligned}$$

where $(a)$ and $(b)$ hold due to Hölder's inequality.

Next, we show Lipschitzness of $t(k)$ using its structure. In fact, due to its definition, $t(k)$ is the maximum of a set of lines with $\{t_i = \big(\psi_0(x)\big)(i) - k\big(\psi_1(x)\big)(i)\}_{i=1}^{n+1}$ in terms of $k$ with slope $m_i = -\big(\psi_1(x)\big)(i)$ and $y$-intercept of $b_i = \big(\psi_0(x)\big)(i)$. Therefore, such piecewise-linear function has a Lipschitz factor equal to the maximum slope of the lines, which in here is equal to $\max_i m_i = \max_i \big|\big(\psi_1(x)\big)(i)\big| \leq 1$. Therefore, $t(k)$ is a 1-Lipschitz function. Therefore, using (77) we can bound $D_{k^*}(f_1, f_2)$ as

$$\begin{aligned}
D_{k^*}(f_1, f_2) &\leq \mathbb{E}[t_x(\hat{k}) - \hat{t}_x(\hat{k})] + 2|k^* - \hat{k}| + \delta_0 + K\delta_1 \\
&= \mathbb{E}[\max_i \big(\psi_0(x) - \hat{k}\psi_1(x)\big)(i) - \max_i \big(\hat{\psi}_0(x) - \hat{k}\hat{\psi}_1(x)\big)(i) + 2|k^* - \hat{k}| + \delta_0 + K\delta_1 \\
&\overset{(a)}{\leq} 2|k^* - \hat{k}| + 2(\delta_0 + K\delta_1),
\end{aligned}$$

where $(a)$ holds because each component of $\big(\psi_0(x) - \hat{k}\psi_1(x)\big)$ and $\big(\hat{\psi}_0(x) - \hat{k}\hat{\psi}_1(x)\big)$ is bounded by $\delta_0 + K\delta_1$, and because the maximum operator is a norm, and therefore satisfies sub-additivity. Finally, since we have bounded $\Delta \le k^* - \hat{k} \le \Delta_l$ with probability at least $1 - \pi - 2e^{-n\epsilon'^2/4}$, then we have

$$D_k(f_1, f_2) \le 2\max\{\Delta, \Delta_l\} + 2(\delta_0 + K\delta_1)$$

$$= 2\frac{\big(2\max\{d_n(\pi), \delta_1\}\big)^{1/\gamma}}{C} + 4\sqrt{\frac{2(\delta_0 + K\delta_1)}{\gamma C}} + 2(\delta_0 + K\delta_1),$$

with such probability. This, together with (74) and (75) shows that

$$\mathbb{E}\big[\langle f_1(x), \psi_0(x)\rangle\big] - \mathbb{E}\big[\langle f_2(x), \psi_0(x)\rangle\big] \le 2\frac{\big(2\max\{d_n(\pi), \delta_1\}\big)^{1/\gamma}}{C} + 4\sqrt{\frac{2(\delta_0 + K\delta_1)}{\gamma C}}$$

$$+ 2(\delta_0 + K\delta_1) + 2Kd_n(\pi),$$

which completes the proof.

## M Proof of Theorem G.1

To prove this theorem, we first prove the following auxiliary lemma

**Lemma M.1.** *For $\alpha, \epsilon \ge 0$, the following holds*

$$\min_{r_i \ge 0, \sum_{i=1}^n r_i \le \alpha} \sum_{i=1}^n r_i d_i - \min_{r_i \ge 0, \sum_{i=1}^n r_i \le \alpha + \epsilon} \sum_{i=1}^n r_i d_i \le \epsilon \cdot \max_{i \in [1:n]} |d_i|$$

*Proof of lemma.* We know that for every positive vector $\mathbf{r}$ with $\sum_{i=1}^n r_i \le \alpha + \epsilon$, we could rewrite that as a sum of two vectors $\mathbf{r} = \mathbf{r}' + \mathbf{r}''$ for which

$$\sum_{i=1}^n r_i' \le \alpha,$$

and

$$\sum_{i=1}^n r_i'' \le \epsilon.$$

As a result, we can rewrite $\min_{r_i \ge 0, \sum_{i=1}^n r_i \le \alpha + \epsilon} \sum_{i=1}^n r_i d_i$ as

$$\min_{r_i \ge 0, \sum_{i=1}^n r_i \le \alpha + \epsilon} \sum_{i=1}^n r_i d_i \ge \min_{r_i' \ge 0, \sum_{i=1}^n r_i' \le \alpha} \min_{r_i'' \ge 0, \sum_{i=1}^n r_i'' \le \epsilon} \sum_{i=1}^n (r_i' + r_i'') \cdot d_i$$

$$= \min_{r_i' \ge 0, \sum_{i=1}^n r_i' \le \alpha} r_i' d_i + \min_{r_i'' \ge 0, \sum_{i=1}^n r_i'' \le \epsilon} \sum_{i=1}^n r_i'' d_i.$$

Hence, we have that

$$\min_{r_i \ge 0, \sum_{i=1}^n r_i \le \alpha + \epsilon} \sum_{i=1}^n r_i d_i - \min_{r_i' \ge 0, \sum_{i=1}^n r_i' \le \alpha} r_i' d_i \ge -\Big|\min_{r_i'' \ge 0, \sum_{i=1}^n r_i'' \le \epsilon} \sum_{i=1}^n r_i'' d_i\Big|$$

$$\overset{(a)}{\ge} -\sum_{i=1}^n r_i'' \cdot \max_{i \in [1:n]} |d_i| \ge -\epsilon \cdot \max_{i \in [1:n]} |d_i|,$$

where $(a)$ holds using Hölder's inequality. $\qquad\square$

Next, we know that the optimal deterministic deferral policy should satisfy

$$\min_{r(x_i) \in \{0,1\}, \frac{1}{n}\sum_i r(x_i) \le b} \frac{1}{n} \sum_i r(x_i)\ell_H(x_i, y_i, m_i) + \big(1 - r(x_i)\big) \cdot \ell_{AI}(x_i, y_i)$$

$$= \frac{1}{n}\sum_i \ell_{AI}(x_i, y_i) + \min_{r(x_i) \in \{0,1\}, \frac{1}{n}\sum_{i=1}^n r(x_i) \le b} \frac{1}{n}\sum_{i=1}^n r(x_i)\big(\ell_H(x_i, y_i, m_i) - \ell_{AI}(x_i, y_i)\big)$$

$$\overset{(a)}{=} \frac{1}{n}\sum_i \ell_{AI}(x_i, y_i) + \underbrace{\min_{r(x_i) \in \{0,1\}, \sum_{i=1}^n r(x_i) \le \lfloor bn \rfloor} \frac{1}{n}\sum_{i=1}^n r(x_i)\big(\ell_H(x_i, y_i, m_i) - \ell_{AI}(x_i, y_i)\big)}_{B},$$

where $(a)$ holds because $r(x_i) \in \{0, 1\}$ and therefore $\sum r(x_i) \le bn$ if and only if $\sum r(x_i) \le \lfloor bn \rfloor$. Now, we turn to examining $B$. To that end, we first consider the following optimization problem:

$$\min_{r(x_i)\in[0,1],\sum_{i=1}^{n} r(x_i)\le\lfloor bn\rfloor} \frac{1}{n}\sum_{i=1}^{n} r(x_i)\big(\ell_H(x_i,y_i,m_i) - \ell_{AI}(x_i,y_i)\big). \tag{78}$$

For a minimizer $\mathbf{r}^*$ of the above problem, we could form $\hat{\mathbf{r}}$ as

$$\hat{r}_i = \begin{cases} r_i^*(x_i) & \ell_H(x_i,y_i,m_i) - \ell_{AI}(x_i,y_i) \le 0 \\ 0 & \ell_H(x_i,y_i,m_i) - \ell_{AI}(x_i,y_i) > 0 \end{cases}.$$

One can see that $\hat{\mathbf{r}}(x)$ is also a minimizer of the above problem. Hence, without loss of generality, we assume that there is an optimal deferral policy that has only non-zero value when $(x, y, m) \in A = \{(x, y, m) \in \mathcal{D} :\}$. Furthermore, we know that since $\hat{r}(x_i) \le 1$, then $\sum_i \hat{r}(x_i) \le \min\{\lfloor nb \rfloor, |A|\}$. We argue that this inequality does not hold in a strict form, i.e., we have $\sum_i \hat{r}(x_i) = \min\{\lfloor nb \rfloor, |A|\}$. The reason is that otherwise one can find $r'(x) \in [0, 1]^{\mathcal{X}}$ such that $\sum_{(x_i,y_i,m_i)\in A} \hat{f}(x_i) + r'(x_i) = \min\{nb, |A|\}$ and because of negativity of $\ell_H(x_i,y_i,m_i) - \ell_{AI}(x_i,y_i)$, we can strictly reduce the objective function that is a contradiction.

Next, we order $\ell_H(x_i,y_i,m_i) - \ell_{AI}(x_i,y_i)$ increasingly and we name them $d_j$. In fact, we define $k_j$ such that $d_j = \ell_H(x_{k_j},y_{k_j},m_{k_j}) - \ell_{AI}(x_{k_j},y_{k_j})$ and that $d_1 \le d_2 \ldots \le d_{|A|} \le 0$. For the sake of simplicity, we further define $r_j := r(x_{k_j})$. As a result, the optimization problem in (78) can be rewritten as

$$\min_{r_i\in[0,1],\ \sum_{i=1}^{n} r_i=\min\{\lfloor nb\rfloor,|A|\}} \sum_{i=1}^{n} r_i d_i.$$

Here, we show that the optimizer of the above problem is $r_i = \mathbb{1}_{i\le\min\{\lfloor nb\rfloor,|A|\}}$. To show that, we consider $r_i' \in [0, 1]$ such that $\sum_{i=1}^{n} r_i' = \min\{\lfloor nb \rfloor, |A|\}$. Then, we have

$$\sum_{i=1}^{n} \mathbb{1}_{i\le\min\{\lfloor nb\rfloor,|A|\}} d_i - \sum_{i=1}^{n} r_i' d_i = \sum_{i:\,\mathbb{1}_{i\le\min\{\lfloor nb\rfloor,|A|\}}-r_i'<0} (\mathbb{1}_{i\le\min\{\lfloor nb\rfloor,|A|\}} - r_i') \cdot d_i$$
$$+ \sum_{i:\,\mathbb{1}_{i\le\min\{\lfloor nb\rfloor,|A|\}}-r_i'>0} (\mathbb{1}_{i\le\min\{\lfloor nb\rfloor,|A|\}} - r_i') \cdot d_i. \tag{79}$$

Now, since we know that $\sum_i \mathbb{1}_{i\le\min\{\lfloor nb\rfloor,|A|\}} = \sum_i r_i'$, we can define a parameter $Q$ as

$$Q := \sum_{i:\,\mathbb{1}_{i\le\min\{\lfloor nb\rfloor,|A|\}}-r_i'>0} \mathbb{1}_{i\le\min\{\lfloor nb\rfloor,|A|\}} - r_i' = \sum_{i:\,\mathbb{1}_{i\le\min\{\lfloor nb\rfloor,|A|\}}-r_i'<0} r_i' - \mathbb{1}_{i\le\min\{\lfloor nb\rfloor,|A|\}}. \tag{80}$$

Next, by defining $p_i := \frac{\mathbb{1}_{i\le\min\{\lfloor nb\rfloor,|A|\}}-r_i'}{Q}$ for $i$s in which $\mathbb{1}_{i\le\min\{\lfloor nb\rfloor,|A|\}} - r_i' > 0$ and $q_i := \frac{r_i'\mathbb{1}_{i\le\min\{\lfloor nb\rfloor,|A|\}}}{Q}$ for $i$s in which $\mathbb{1}_{i\le\min\{\lfloor nb\rfloor,|A|\}} - r_i' < 0$ and 0 otherwise, we conclude that $\{p_i\}_i$ and $\{q_i\}_i$ are probability mass functions. Hence, using (79) and (80), we have

$$\sum_{i=1}^{n} \mathbb{1}_{i\le\min\{\lfloor nb\rfloor,|A|\}} d_i - \sum_{i=1}^{n} r_i' d_i = Q\Big( \sum_{i=1}^{\min\{\lfloor nb\rfloor,|A|\}} p_i d_i - \sum_{i=\min\{\lfloor nb\rfloor,|A|\}+1}^{n} q_i d_i \Big).$$

The above identity contains the difference of two expected value over random variables that one is always smaller than the other. As a result, we show that

$$\sum_{i=1}^{n} \mathbb{1}_{i\le\min\{\lfloor nb\rfloor,|A|\}} d_i - \sum_{i=1}^{n} r_i' d_i \le 0,$$

which completes the proof.

