# OpenReview forum: "A Unifying Post-Processing Framework for Multi-Objective Learn-to-Defer Problems"
_NeurIPS.cc/2024/Conference — NeurIPS 2024 poster_

### Official Review · Reviewer_4QNG · 2024-07-03

**Soundness:** 3
**Presentation:** 3
**Contribution:** 2
**Rating:** 4
**Confidence:** 5

**Summary:**

While there has been extensive research on L2D, general methods for designing such systems under various constraints (e.g., algorithmic fairness, expert intervention budget, defer of anomaly, etc.) remain largely unexplored. This paper utilizes $d$-dimensional generalization to the fundamental lemma of Neyman and Pearson ($d$-GNP) to obtain Bayes optimal solutions for L2D under various constraint conditions and designs a generic algorithm to estimate these solutions.

**Strengths:**

1. This paper proposes a novel general framework for addressing L2D under various constraint conditions.
2. Different constraint conditions can be simultaneously addressed.
3. The proposed method is a post-processing approach that can be easily extended to existing models.

**Weaknesses:**

1. The general framework provided by the author can address various constraint problems, but it does not seem to be reflected in the experimental section (especially in cases involving multiple constraints), only considering simple single-constraint situations and basic comparisons.

Expert Intervention Budget:

Liu S, Cao Y, Zhang Q, et al. Mitigating Underfitting in Learning to Defer with Consistent Losses. ICML, 2024.

Narasimhan H, Jitkrittum W, Menon A K, et al. Post-hoc estimators for learning to defer to an exper. NeurIPS, 2022.

Long-Tail Classification:

Narasimhan H, Menon A K, Jitkrittum W, et al. Learning to Reject Meets Long-tail Learning. ICLR, 2024.

**Questions:**

1. Post-Processing Framework has many benefits, but it also faces challenges in confidence calibration, which is particularly evident in deep models. Inevitable overfitting makes the model's output probabilities unreliable, indirectly leading to biases in approaching the Bayes optimal solution through post-processing methods. Does the method proposed in this paper include mechanisms to mitigate these issues?
2. There are many studies similar to L2D, such as the dynamic classifier selection literature (DCS). What is the main difference between L2D and DCS?

Cruz R M O, Sabourin R, Cavalcanti G D C. Dynamic classifier selection: Recent advances and perspectives[J]. Information Fusion, 2018, 41: 195-216.

**Limitations:**

N/A.

---

> ### Author Rebuttal · Authors · 2024-08-07
>
> We thank the reviewer for their positive feedbacks. We further address their questions in the following.
> ## Questions
>
>  **Q1** Dealing with overfitting is largely studied in the machine learning literature. In this work, we assume that the scores are estimated using the well-studied methods such as calibrated neural networks and random forests. This is an assumption that a large body of the field, from the first works of L2D (e.g., Mozannar et al. 2020) holds. If the posterior probabilities are not accurate, as the reviewer has mentioned, all the plug-in methods (including Mozannar et al., Verma et al., and Cao et al. for unconstrained L2D problem) would lead to incorrect solutions. Therefore, we believe the study of this issue is beyond the scope of this paper and is of its own interest.
> However, we should add that, as the reviewer has mentioned, the calibration of such probabilities would equip us with an interesting set of properties. In fact, if we know that the neural network $g(x)$ is calibrated, i.e., $P(Y=1|g(X)=p)=p$, then if the only information in our hand is that calibrated estimation, then the plug-in methods are reduced to the plug-in methods for $g(x)$, since in this case $\Pr(Y=1|g(x))=g(x)$, and therefore rules such as $\Pr(Y=1|g(x))\geq T$ is equivalent to $g(x)\geq T$.
>
> **Q2** These two fields are highly interrelated in the sense that in both of these fields, a competent decision-maker is selected for a particular input feature. However, the difference between these fields is that in L2D systems, due to the some reasons such as transparency and responsibility, the decision is made either by human or the classifier, while in DCS systems the solutions of the classifier can be fused with each other. Such extensions to L2D problems are also studied in the literature, e.g., Charusaie et al. 2024, and Steyvers et al. 2022. We make sure to include these references as well as DCS references in our manuscript.
>
> Charusaie, Mohammad-Amin, Amirmehdi Jafari Fesharaki, and Samira Samadi. "Defer-and-Fusion: Optimal Predictors that Incorporate Human Decisions."
>
> Steyvers, Mark, et al. "Bayesian modeling of human–AI complementarity." Proceedings of the National Academy of Sciences 119.11 (2022): e2111547119.
>
>
> ## Weaknesses
>
>
> **The general framework provided by the author can address various constraint problems, but it does not seem to be reflected in the experimental section (especially in cases involving multiple constraints), only considering simple single-constraint situations and basic comparisons.** We first should note that, as we mention in the main rebuttal section, the ACSIncome experiment controlls two constraints, namely, the disparity of true negative and false negative predictions for two demographic groups. In the following, we further introduce a new set of multi-constraint experiments in the Hate Speech experiment.
>
>
> We believe that the comments of the reviewer is remained unfinished, or otherwise we are not sure how the references on **Expert Intervention Budget** and **Long-Tail Classification** are related to their comments. We have indeed cited these references in the current version of the manuscript.
>
> ## New Experiment on Hatespeech Dataset
>
> We further experimented our algorithm on Hatespeech dataset (Davidson et al. 2017) that is a content-moderation task to flag hateful and offensive tweets. The results are plotted in the submitted PDF. We used a pre-trained model (Blodgett et al. 2016) to detect whether a tweet is written in an African-American (AA) or not (NAA). We then used our algorithm to control the (i) parity of predicting a tweet hateful, or (ii) offensive, and (iii) false negative probability difference for group AA and NAA. We can observe in Figure (d) that the disparity in predicting a tweet "Offensive" for our algorithm (violet bars) is reduced to the tolerance $0.1$ by deferral to the human for group AA (see Figure (b)), and otherwise randomization of the results for group NAA (Figure (c)). Similarly, the disparity in predicting "Hate Speech" is diminishing in our algorithm (brown bars of Fig. 2,3, and 4). Furthermore, we can reduce the disparity of false negative prediction (finding a tweet Hate Speech or Offensive, given it is neither) in Figure (e). Here we note that the high variance in Fig. (e) is due to small size of non-offensive tweets. We furtherran experiments on **multi-constraint** setting in which both demographic disparity and human intervention budget is controlled. The accuracy and the true constraints are plotted in Fig. (h), (i), and (j). Similarly, Type-II error and human budget is controlled together and plotted in Fig. (k), (l), and (m).
>
> Davidson, Thomas, et al. "Automated hate speech detection and the problem of offensive language." Proceedings of the international AAAI conference on web and social media. Vol. 11. No. 1. 2017.
>
>
> Blodgett, Su Lin, Lisa Green, and Brendan O'Connor. "Demographic dialectal variation in social media: A case study of African-American English." arXiv preprint arXiv:1608.08868 (2016).

---

> > ### Comment · Reviewer_4QNG · 2024-08-09
> > **Response**
> >
> > Thank you for your elaborated response to my review. Please allow me a response to your response.
> >
> > add Q1/A1: I know that addressing the probability calibration issue caused by overfitting is challenging, so the author's plug-in method relies on excellent surrogate loss functions. However, we need to point out that in previous surrogate loss function, only comparisons between probabilities are required (e.g., comparisons may be accurate, but values may not be accurate, such as Mozannar et al. 2020), rather than direct use. In contrast, the author's method requires direct utilization of estimated probabilities that may be inaccurate for multiple operations, which may amplify errors. I know that calibrating probabilities is not the focus of this study, but I believe this is a potential risk of the plug-in method that warrants discussion.
> >
> > add Weaknesses: I apologize for not accurately expressing my confusion earlier. My main concern is that the authors claim their framework can solve various constraint problems, but in the experiment section, they don't compare their work with previous works that have already solved some of these constraint problems (such as expert intervention budget and long-tail classification). I think such comparative experiments are very important.
> >
> > In my opinion, the lack of comparative experiments is the main drawback of the work.

---

> ### Author Response · Authors · 2024-08-09
>
> We first thank the reviewer for their clarifications and their response. Here, we mention a few notes.
>
> Q1: We are glad that the reviewer agrees with us on our choice of surrogate functions. We further agree with the reviewer that linearly combining the scores might propagate the error, in a controlled manner, and in the value of objective and not the constraints. This is precisely what we have discussed after Theorem 5.1 regarding the constraint generalization.
>  There, we have discussed that for a given error in $\ell_{\infty}$ norm of the scores, the constraints can deter at most for that error, plus the  sample complexity term. Furthermore, in the error upper-bound (10) that is provided in **Theorem 5.3**, we observe that given the linear combination coefficient being bounded by $K$, and given the sensitivity factor being bounded away from $0$, **the objective cannot be far from the true objective, as long as score errors $\delta_0$ and $\delta_1$ are small enough**.  We note that, when the constraint is well within the feasibility set, the value $K$ is more tightly upper-bounded. Furthermore,  as we discussed it in the response to Q6 of Reviewer 7rh1, and as we have shown as an example, in Figure (f) and (g) of the submitted PDF, the constraints that lie within the feasibility set improve the sensitivity factor, thereby reducing the risk of error propagation.
>
>  ## Weaknesses
>
> We appreciate the reviewer's clarification of their comments. It is important to emphasize that our work does not aim to compete with other constrained L2D optimal methods but rather to unify these methods within a single framework. Our method should indeed provide a reasonable solution for simple constraints, and improve the empirical methods by obtaining the true optimal solution for cases in which **the true optimal is not known in the literature**, such as **L2D with fairness criteria (demographic parity, equalized odds, and equality of opportunity), or a combination of a set of constraints**. These claims are supported by the experiments within the manuscript and the rebuttal. However, we do not expect that our method outperforms methods that already achieve or approximate the true optimality, such as those dealing with expert intervention budgets or long-tail classification.
>
> We will further attempt to run the additional experiments suggested by the reviewer during the remaining discussion period. However, given the time constraints and the incompleteness of the reviewer's initial comments, it might not be feasible to complete them within this period.

---

> > ### Author Response · Authors · 2024-08-13
> > **New Comparative Experiments + Theoretical Equivalence**
> >
> > Dear Reviewer,
> >
> > We further thank you for the suggestion of comparing our unifying methods with other methods in specific cases. In the time we had we could compare our method to the two of the mentioned work, Liu et al 2024, and Narasimhan et al. 2022. for the sanity check. We have indeed simulated the post-hoc formula of (11) in Narasimhan et al. 2022 and the modified equation (9) of Liu et al. with One-vs-All Losses. Since these works are not tailored to handle hard thresholds, we have fitted the parameters for each specific tolerance via validation data in Narasimhan et al. and via re-training in Liu et al. and searching within 1000 values of deferral cost between -1 and 1. The comparison shows a competetive accuracy in our method compared to these baselines. We note here that the high variance of the accuracy in COMPAS dataset is already reported in Mozannar et al. 2023, Figure 3 (b).
> >
> > ### Hatespeech
> > | Tolerance | d-GNP Accuracy | d-GNP Constraint | Liu et al. 2024 Accuracy | Liu et al. 2024 Constraint | Narasimhan et al. 2022 Accuracy | Narasimhan et al. 2022 Constraint |
> > |----|---|---|---|---|---|----|
> > | 0.05      | **0.890 ± 0.004** | 0.049 ± 0.005 | 0.887 ± 0.004 | 0.043 ± 0.005 | 0.886 ± 0.005 | 0.047 ± 0.006 |
> > | 0.10      | **0.902 ± 0.005** | 0.099 ± 0.007 | 0.899 ± 0.005 | 0.092 ± 0.005 | 0.898 ± 0.005 | 0.098 ± 0.007 |
> > | 0.15      | **0.910 ± 0.004** | 0.147 ± 0.011 | 0.908 ± 0.005 | 0.137 ± 0.009 | 0.908 ± 0.005 | 0.148 ± 0.009 |
> > | 0.20      | **0.916 ± 0.004** | 0.200 ± 0.011 | 0.914 ± 0.005 | 0.186 ± 0.010 | 0.914 ± 0.004 | 0.195 ± 0.012 |
> > | 0.25      | **0.920 ± 0.004** | 0.244 ± 0.009 | 0.918 ± 0.004 | 0.233 ± 0.013 | 0.917 ± 0.005 | 0.240 ± 0.017 |
> > | 0.30      | **0.921 ± 0.005** | 0.287 ± 0.016 | 0.920 ± 0.005 | 0.276 ± 0.025 | 0.919 ± 0.005 | 0.283 ± 0.015 |
> >
> > ### COMPAS
> > | Tolerance | d-GNP Accuracy | d-GNP Constraint | Liu et al. 2024 Accuracy | Liu et al. 2024 Constraint | Narasimhan et al. 2022 Accuracy | Narasimhan et al. 2022 Constraint |
> > |---|---|---|---|---|---|---|
> > | 0.015  | 0.646 ± 0.022  | 0.014 ± 0.010 | **0.649 ± 0.031** | 0.003 ± 0.004 | **0.649 ± 0.031**  | 0.016 ± 0.018  |
> > | 0.075  | 0.647 ± 0.026  | 0.042 ± 0.033 | 0.648 ± 0.031 | 0.030 ± 0.022 | **0.650 ± 0.031**  | 0.053 ± 0.032  |
> > | 0.135  | 0.649 ± 0.026  | 0.072 ± 0.050 | **0.651 ± 0.026**  | 0.065 ± 0.040 | 0.649 ± 0.031  | 0.082 ± 0.048   |
> > | 0.195  | **0.656 ± 0.023** | 0.111 ± 0.074  | 0.654 ± 0.027 | 0.079 ± 0.054  | 0.652 ± 0.033 | 0.132 ± 0.069 |
> > | 0.255  | **0.656 ± 0.029** | 0.122 ± 0.085  | 0.653 ± 0.027  | 0.108 ± 0.085  | 0.655 ± 0.033 | 0.165 ± 0.095  |
> >
> > We further discuss the theoretical equivalence between our method and other methods that tackle human intervention budget and long-tail classification by which we conclude that the closeness of the accuracies to these baselines are not surprising.
> >
> >   For human intervention budget, since using Table 1 we know that the embedding functions are as
> >   $$\psi_0(x)=[\Pr(Y=0|X=x), \ldots, \Pr(Y=L|X=x), \Pr(Y=M|X=x)],$$
> >   and
> >   $$\psi_1(x)=[0, \ldots, 0, 1],$$  therefore, the optimal predictor using Theorem 4.1 and the discussion after (4) is equal to
> >   $$h^*(x)=\arg\max_{i\in [1:L]} \psi_0(x)-k\psi_1(x) $$,
> >   and $r^*(x) = 1$ if $L+1=\arg\max_{i\in [1:L+1]} \psi_0(x)-k\psi_1(x)$. Therefore, using the definitions of embedding functions, we have
> >   $$h^*(x)= \arg\max_{1:L} [\Pr(Y=1|X=x), \ldots, \Pr(Y=L|X=x)],$$ and $r(x)=1$ if $\Pr(Y=M|X=x)-k>\max \{\Pr(Y=1|X=x), \ldots, \Pr(Y=L|X=x)\}$. This is equivalent to what is proposed in (11) in Narasimhan et al. 2022.
> >
> >   For the long-tail classification problem using Table 1, we have
> >   $$\psi_0(x)= \Big[\frac{\Pr(Y=1|X=x)}{\alpha_{i_1}\Pr(Y\in G_{\alpha_{i_1}})}, \ldots, \frac{\Pr(Y=L|X=x)}{\alpha_{i_L}\Pr(Y\in G_{\alpha_{i_L}})}, 0\Big]$$
> >   and
> >   $$ \psi_t(x) = \frac{\Pr(Y\in G_t|X=x)}{\Pr(Y\in G_t)}[1, \ldots, 1, 0] - \frac{\alpha_t}{K}$$
> >   as embedding functions, where $i_j$ is the index of the group $G_{i_j}$ to which the label $j$ belongs. Now, since the first $L$ components of all $\psi_t$s are equal to each other, we have
> >   $$h^*(x)=\arg\max_{i\in [1:L]} \psi_0(x) - \sum_{t=1}^K k_t \psi_t(x) = \arg\max_{i\in [1:L]} \psi_0(x) = \arg\max \Big[\frac{\Pr(Y=1|X=x)}{\alpha_{i_1}\Pr(Y\in G_{\alpha_{i_1}})}, \ldots, \frac{\Pr(Y=L|X=x)}{\alpha_{i_L}\Pr(Y\in G_{\alpha_{i_L}})}\Big].$$
> >  Furthermore, the tightness condition $\mathbb{E}\big[\langle \psi_t(x), f^*(x)\rangle\big]=0$ concludes that $\alpha_{i_j}\Pr(Y\in G_{\alpha_{i_j}}) = \alpha^*_{j}$ where $\alpha^*_{j}$ defined in (8) of Narasimhan et al. 2024. These two conclude in the optimal classifier (8) of Narasimhan et al. 2024.
> >  Similarly, we can show that the optimal deferral strategy using $d$-GNP is
> >  $r(x)=1$ if $\sum_{t}\frac{k_t \Pr(Y\in G_t|X=x)}{\Pr(Y\in G_t)}>\max\Big[\frac{\Pr(Y=1|X=x)}{\alpha_{1}^*}, \ldots, \frac{\Pr(Y=L|X=x)}{\alpha_{L}^*}\Big]$ which is a similar rule to that of (8) in Narasimhan et al. by setting $\mu^*\_t=1-k_t \alpha_{i_t}$.

---

> > > ### Comment · Reviewer_4QNG · 2024-08-14
> > > **Concerns remain about the experimental component**
> > >
> > > Thank you to the authors for providing some of the experimental results. However, I’m afraid I still feel that this paper is not very complete. Despite the theoretical section being thorough, there are too many omissions in the experimental part that need to be addressed.
> > >
> > > Specifically:
> > >
> > > 1. There is a lack of separate experiments addressing the various constraints involved in this work.
> > > 2. There are no cross-experiments with different constraints (they do not need to be exhaustive).
> > > 3. There is a lack of comparative experiments with methods that have already addressed the constraints.
> > >
> > >
> > > Additionally, I have concerns regarding the experimental results related to the expert intervention budget that were added at the end. Although the method proposed by Narasimhan et al. (2022) cannot be directly compared (as the cost needs to be adjusted), according to the viewpoint in [1], their optimal solutions should be the same. Furthermore, you have employed the same post-processing method; therefore, the experimental results should be fundamentally similar. However, from the aforementioned experiments, it appears that the method presented in this paper performs better.
> > >
> > >
> > > [1] W. Jitkrittum, G. Neha, A. K. Menon, H. Narasimhan, A. Rawat, and S. Kumar. When Does Confidence-Based Cascade Deferral Suffice? In NeurIPS, 2023

---

> ### Author Response · Authors · 2024-08-14
> **A Resoponse to Your Concerns**
>
> We thank the reviewer for their response and comments our experimental results. As we have tried to show that via experimental results as well as theoretical results, our method provides the same theoretical post-processing method as of Narasimhan et al. 2022 for the case of human intervention budget. The differences of these methods are reflecting the way that the probabilities are estimated, i.e., the choice of loss function, and the encoding of deferral choice in the network. We have used a softmax loss function, while as mentioned, in our limited time, we implemented Section 4.1 of Narasimhan et al. that uses a form of one-vs-all loss. These two as we see have the difference of order of 0.001, while the standard deviation of accuracies is larger than this amount. As a result, we believe this is an evidence that the two methods lead to a similar result.
>
> Here, we want to reiterate the empirical results of our work:
>
> 1- We have obtained the L2D solution under demographic parity for COMPAS dataset and compared to Madras et al. 2018 and Mozannar et al. 2022.
>
> 2- We have obtained the L2D solution under equalized odds for ACS dataset.
>
> 3- In the rebuttal period, we have implemented our work to Hatespeech dataset and under demographic parity and equality of opportunity constraint.
>
> 4- To address the reviewer's concern, we have obtained the optimal solution under multiple constraints of demographic parity+human intervention budget and Type-II error+human intervention budget.
>
> 5- To address the reviewer's concern, we have compared our method with Narasimhan et al. 2022 and Liu et al. 2024 for Hatespeech dataset as well as COMPAS dataset.

---

### Official Review · Reviewer_Zrm7 · 2024-07-08

**Soundness:** 3
**Presentation:** 1
**Contribution:** 2
**Rating:** 5
**Confidence:** 3

**Summary:**

The paper studies multi-objective learn-to-defer problems, where the objectives include minimizing deferral loss and satisfying several constraints. It demonstrates that these problems are generally NP-hard and can be reduced to functional linear programming. Additionally, it shows that the problem can be further reduced to a d-dimensional generalized Neyman-Pearson problem and characterizes its solution when there is only one constraint. The paper also presents a unifying post-processing algorithm with generalization bounds and provides numerical validations.

**Strengths:**

The paper offers several theoretical insights into multi-objective learn-to-defer problems, enhancing the understanding of these problems. It also introduces a unifying post-processing algorithm with provable performance guarantees.

**Weaknesses:**

1.Inconsistent Notations and Typos: The paper is difficult to follow due to inconsistent notations and potential typos. For instance, $m$ can mean the expert decision (line 111) or the number of constraints (Equation 3). $r$ can mean the rejection function (line 110) or a distribution vector (Theorem 4.1).  In line 7 of Algorithm 1, should it be $\hat{C}(k)$ instead of $\hat{C}(t)$? And line 208 only defines  $\psi_{i}$ for $i=1,...,m+1$, then in line 209, what is $\psi_{0}$?

2.Limitations on Multiple Constraints: The paper acknowledges that the provided analyses for the algorithm (and possibly the algorithm itself) only apply to single constraint settings. More discussion on the challenges of extending these analyses (and potentially the algorithm) to multiple constraints settings would be helpful.

**Questions:**

1.Why in Theorem 4.2, $f^*_{k,p}(x)=\tau(\psi_1(x)-k\psi_0(x))$, while in Algorithm 1, its estimation version is $\hat{f}_{k,p}(x)=\tau(\hat{\psi}_0(x)-k\hat{\psi}_1(x))$ (where the positions are swapped)?


2.Can Algorithm 1 handle multiple constraints settings? Figure 1 depicts a diagram of multiple constraints settings, but it seems Algorithm 1 is designed for single constraint settings.

**Limitations:**

See weaknesses.

---

> ### Author Rebuttal · Authors · 2024-08-07
>
> We thank the reviewer for reading the paper and their comments. We are respectively address the concerns of the reviewer.
>
> 1. **1.Why in Theorem 4.2, $f_{k, p}*(x)=\tau(\psi_1(x)-k\psi_0(x))$, while in Algorithm 1, its estimation version is $\hat{f}_{k,p}(x)=\tau(\hat{\psi}_0(x)-k\hat{\psi}_1(x))$ (where the positions are swapped)?** This is a typo that we addressed in the newer version of this manuscript. In this version $\psi_0$ corresponds to the objective, while $\psi_1, \ldots, \psi_{m+1}$ correspond to the constraints.
>
> 2. **Can Algorithm 1 handle multiple constraints settings? Figure 1 depicts a diagram of multiple constraints settings, but it seems Algorithm 1 is designed for single constraint settings.** This algorithm, as mentioned in the main rebuttal can handle many constraints, just by adding the line of finding the empirical solution of $k_1, \ldots, k_m$ for (3) and based on validation dataset.
>
> 3. **Inconsistent Notations and Typos**: We have extensively proofread and resolved the typos that are raised by the reviewer. We refer the reviewer to the main rebuttal section for further details.
>
> 4. **Limitations on Multiple Constraints: The paper acknowledges that the provided analyses for the algorithm (and possibly the algorithm itself) only apply to single constraint settings. More discussion on the challenges of extending these analyses (and potentially the algorithm) to multiple constraints settings would be helpful.** As we further explained in the main rebuttal section, the algorithm can handle many constraints. The main result of this paper also hold for many constraints. Further extensions of the sample complexities are obtained in the main rebuttal section.
>
> Here, we would be grateful if the respected reviewer would let us know why they think our paper is not fit to this conference, since we believe there is a disparity between the comments and the scores of the reviewer. The typos and notation issues can be minor issues that will not entail any major concerns in our work.

---

> > ### Comment · Reviewer_Zrm7 · 2024-08-08
> >
> > Thank you for your response. It addresses some of my concerns and I have raised the score. However, I must emphasize that the typos and notation issues significantly hinder the understanding of the work. In my view, a high-quality paper should have minimal typos and consistent notations to ensure clarity and readability.

---

> ### Author Response · Authors · 2024-08-09
>
> Dear Reviewer,
>
> Thank you for your response and for raising the score. We are glad that we could address your concerns. Regarding the notation issues, we assure you that we will carefully proofread the next version of the manuscript to eliminate any typos.

---

### Official Review · Reviewer_7rh1 · 2024-07-15

**Soundness:** 3
**Presentation:** 3
**Contribution:** 3
**Rating:** 6
**Confidence:** 3

**Summary:**

The paper introduces a unifying post-processing framework for multi-objective learn-to-defer problems, allowing the system to defer tasks to an expert under specified constraints. By generalizing the Neyman-Pearson lemma, the paper derives the Bayes optimal solution for this framework and develops an algorithm to estimate it. The proposed algorithm is evaluated using the COMPAS and ACSIncome datasets.

**Strengths:**

1. The paper introduces a general post-processing framework.
2. The paper is well-written and has excellent illustrations, for example, Figure 1.
3. The framework accounts for multiple objectives in L2D.
4. The method can potentially be extended beyond the L2D setting to apply to other constrained objectives.

**Weaknesses:**

1. The proposed algorithm requires the estimation of scores and $\hat \psi$ (Lines 4 and 5 of Algorithm 1). However, accurately estimating these values can be potentially challenging.
2. The experiments could potentially be expanded further.

**Questions:**

1. The multi-class Neyman-Pearson (NP) paradigm has been studied in [66]. Could the authors summarize the key differences between the framework proposed in this paper and the prior work?

2. What is the motivation for imposing specific constraints in learning to defer? Could this compromise the accuracy of the process?

3. How does the proposed algorithm empirically compare with the baselines on the ACSIncome datasets? Do you think the experiments could be expanded further?

4. It seems that the method could potentially be applicable to other constrained objectives beyond the L2D setting. Could the authors further elaborate on this?

5. Could the authors further comment on the challenges of extending the analysis and algorithm to multiple constraint cases?

6. How are the tolerances chosen during the experiments? Can the assumptions in Theorem 5.3 be verified in practice?

**Limitations:**

Yes.

---

> ### Author Rebuttal · Authors · 2024-08-07
>
> We thank the reviewer for the positive and constructive comments on this paper. In the following, we respond their questions in order
>
> **Q1** A: This work that we have further referenced in our manuscript, as we mentioned after (6), has three main differences with our work. *(i)* This work only considers the Type-$k$ error and is designed to find the solution of the optimal classifier given that specific constraints. In contrary, our work can contain all types of constraints that are in form of loss function (e.g., fairness criteria, human intervention budget, long-tail classification). *(ii)* This work is based on strong duality theorem that as we discussed in Appendix E is not considered in our work, due to its limitations (see the counterexample of Appendix E). *(iii)* In our work, we not only find the optimal solution to the constrained optimization problem (3), but also show that all solutions to this problem are of the form that we have formulated in (8).
>
> **Q2** A: The L2D systems can be used in many applications, including applications that are sensitive in nature. As an instance, we can design an algorithm that either classifies a disease based on the data of a patient, or defer the decision to the doctor. This system in total can be unfair, meaning that the error that is induced by this system not detecting the disease for different demographic groups can be different. As another example, as we further explained our experiment for Reviwer 4QNG, this can be the case when we want to detect Hatespeech from a tweet automatically or using our agents. This decision could entail different errors for the tweets that are written with an African-American dialect and the ones that are not. Our results helps controlling this algorithmic unfairness. Since we add a constraint to our optimization problem, the corresponding search space reduces, and therefore the accuracy can be compromised, which is the case in many algorithmically fair methods.
>
> **Q3** A:  Our experimental setting now have expanded and include content moderation experiments on Hate Speech Dataset. We refer the reviewer to the rebuttal section of Reviewer 4QNG for the details of these experiments. Note that there is no known basline to ensure a certain equalized odds constraint for an L2D system. Particularly in our setting in which the model is a random forest. However, in the next revision, we will compare our method with best fair classifier and an adapted version of Mozannar et al. 2020 for random forests in which we shift thresholds for different demographic group.
>
> **Q4** A very interesting application of $d$-GNP is its use in fair vanilla multi-class classification. This theorem can show that for controlling demographic parity or equality of opportunity, we need to first learn the scores, and then add or multiply a value to these scores based on the demographic identity.
>  In fact, or an $L$-class classifier, if we aim to set constraints on demographic parity  $\big|\Pr(\hat{Y}=0|A=0)-\Pr(\hat{Y}=0|A=1)\big|\leq \delta$ or equality of opportunity $\big|\Pr(\hat{Y}=0|Y=0, A=0)-\Pr(\hat{Y}=0|Y=0, A=1)\big|\leq \delta$ on Class $0$, then we can follow similar steps as in Appendix D to find the embedding functions as
>     $$
>     \psi_{\mathrm{DP}} = s(A)\big[1, 0, \ldots, 0\big],
>     $$
>     and
>     $$
>     \psi_{\mathrm{EO}} = t(A, 0)\big[\Pr(Y=0|x), 0, \ldots, 0\big].
>     $$
>     As a result, since the accuracy embedding function is $\psi_0(x)=\big[\Pr(Y=0|x), \ldots, \Pr(Y=L|x)\big]$, then, by neglecting the effect of randomness, the optimal classifier under such constraints are as
>     $$
>     h_{\mathrm{DP}}(x)=\arg\max\big\(\Pr(Y=0|x)-ks(A), \Pr(Y=1|x), \ldots, \Pr(Y=L|x)\big\),
>     $$
>     and
>     $$
>     h_{\mathrm{EO}}(x)=\arg\max\big\(\Pr(Y=0|x)\big(1-kt(A, 0)\big), \Pr(Y=1|x), \ldots, \Pr(Y=L|x)\big\).
>     $$
>     Equivalently, for demographic parity, the optimal classifier includes a shift on the score of Class $0$ as a function of demographic group, and for equality of opportunity, the optimal classifier includes a multiplication of the score of Class $0$ with a value that is a function of demographic group. It is easy to show that under condition of positivity of the multiplied value, these classifiers both reduce to thresholding rules in binary setting.
>
> **Q5** A: We refer the reviewer to the main rebuttal for the response to this question.
>
> **Q6** A: The tolerances are chosen in a manner that capture the dynamics of the accuracy within that range. If we permit for a very large tolerance, the accuracy saturates to its Bayes optimal value. This is what occurs at the right part of ACSIncome figure.
> The main assumption of Theorem 5.3 is the sensitivity assumption of Definition 5.2. To test that assumption, we can set $\gamma=1$, and reduce this assumption to a lower-bound on the derivative of the constraint w.r.t the coefficient $k$. We have plotted this in Fig. (f) and (g) of the submitted PDF and we observe that for the cases that $k$ induce a constraint within the plausible constraints, this derivative is lower-bounded. In fact, Fig. (f) shows the accuracy of the estimator for two coefficients of corresponding constraints of demographic disparities of detecting a tweet Offensive (DP-O) or Hate-Speech (DP-HS). We observe that as long as there is dynamics in the accuracy in terms of this coefficients, then the derivative of the constraints in terms of this coefficient is bounded below. This is particularly observable in right-most part of Fig. (g) for DP-O and the left-most part for DP-HS.
>
> **Q7** We should mention that estimating the value of the scores are equivalent to estimating the value of a set of posterior probabilities that are used in obtaining these scores. This is a task for which machine learning literature has developed corresponding tools such as deep neural networks, random forests, etc. Therefore, we don't believe the estimation of our scores is of particular hardness compared to this literature.

---

> > ### Comment · Reviewer_7rh1 · 2024-08-12
> >
> > Thank you for your detailed response. I will keep my score as is.

---

### Official Review · Reviewer_xkzY · 2024-07-28

**Soundness:** 3
**Presentation:** 3
**Contribution:** 3
**Rating:** 7
**Confidence:** 4

**Summary:**

This paper aims to provide a provably consistent unified post-processing framework of learning to defer with constraints. The problem of constrained L2D is first reduced to a linear programming problem. Then the linear programming problem is further tackled with a generalized version of Neymar-Pearson lemma, which lead to an efficient solver of this problem. Non-asymptotic analysis is conducted to further provide guarantee for the empirical version of the algorithm. Experimental results validate the efficiency of the proposed solver.

**Strengths:**

1. A unified post-processing framework for constrained L2D is proposed in this paper, which allows training a randomized classifier-rejector tuple with small number of validation data combined with a trained model’s confidence output.

2. The hardness of directly solving the original problems and in-processing methods are thoroughly analyzed in this paper, which provides enough rationale for the proposed method.

3. The introduction of the generalized Neyman-Pearson lemma is novel to the field of L2D, which can provide insights for the future works of this field.

**Weaknesses:**

1. The related works are moved to the appendices, which can be confusing to readers that are new to this field. I suggest the authors use a more compact type setting in the contribution part of Section 1 and reduce/integrate some of the contributions to make more spaces. The paragraph ‘Type of Constraints’ can also be reduced given the Table 1.

2. The notations in this paper need further proofread, e.g., the realization and random variables are used improperly in line 141, line 188; the quantity $a^{i}$ in the algorithm 1 seems to be unused in this algorithm.

**Questions:**

1. It is mentioned that solving (2) is NP-hard. While the proof is quite clear, I wonder if such hardness is important. In my opinion, a common practice that can avoid directly solving this problem is using a surrogate loss instead of 0-1 loss and integrating the expert coverage constraint into the loss like that in the Selective-net. Can you further make some discussions on this point?

2. If we have rather accurate class probability estimates and expert accuracy, why not use it as the solver of this problem?

**Limitations:**

Please see the weaknesses.

---

> ### Author Rebuttal · Authors · 2024-08-07
>
> We first thank the reviewer for their positive feedback to our submission. Here, we respond to their questions
>
> 1. **Q: It is mentioned that solving (2) is NP-hard. While the proof is quite clear, I wonder if such hardness is important. In my opinion, a common practice that can avoid directly solving this problem is using a surrogate loss instead of 0-1 loss and integrating the expert coverage constraint into the loss like that in the Selective-net. Can you further make some discussions on this point?** A: The point of the hardness theorem is to justify the use of randomness in multi-objective setting. We should note that the practical solutions to this problem  such as Selective-Net that use surrogates of the loss functions do not enjoy the guarantees for optimality. This is as opposed to the unconstrained case, in which there are continuous surrogate functions that are Fisher consistent, and therefore their minimization is equivalent to the $0-1$ loss minimization.
>
> 2. **Q: If we have rather accurate class probability estimates and expert accuracy, why not use it as the solver of this problem?** A: The class probabilities as well the expert accuracy can help us find the optimal solution when we allow for randomness and using $d$-GNP. If, however, we aim to find the optimal deterministic solution that is not the case. In Appendix E we introduce an example that the human has perfect information of the label, while the input feature has no information about the label. In this case, it would be more efficient to defer the decision to the human. However, if we bound the budget of the human to $b$ proportion of samples, the optimal deterministic solution would be to defer on inputs that are arised with sum probability of $b$ and not to defer in the other $1-b$ proportion. Therefore, on the inputs with the same conditional accuracy and the same conditional expert accuracy, we have a different decision. This is the reason to the hardness theorem, i.e., finding that set of inputs that sum up to $b$ can be a complex task.
>
> 3. **Q: The related works are moved to the appendices, which can be confusing to readers that are new to this field. I suggest the authors use a more compact type setting in the contribution part of Section 1 and reduce/integrate some of the contributions to make more spaces. The paragraph ‘Type of Constraints’ can also be reduced given the Table 1.** A: This was a decision due to the lack of space. We will make make more space for the related works in the next version of the manuscript, as suggested by the reviewer.
>
> 4. **Q: The notations in this paper need further proofread, e.g., the realization and random variables are used improperly in line 141, line 188; the quantity $a_i$ in the algorithm 1 seems to be unused in this algorithm.** As mentioned in our main rebuttal, we have made an extensive effort to proofread the paper. We have addressed the typos mentioned by the reviewer and appreciate their thorough reading of the manuscript.

---

> > ### Comment · Reviewer_xkzY · 2024-08-12
> >
> > Thank you for your response. Your discussions on the motivation of this work and the hardness analyses have solved my concerns. I’ll keep my decision of acceptance.

---

### Author Rebuttal · Authors · 2024-08-07

We first thank all reviewers for the time they have put in writing this set of constructive reviews. In particular, we are glad that the reviewers found our method "well-written", "thoroughly analyzed", "novel to the field of L2D", "enhancing the understanding of these problems", "can be extended beyond the L2D setting", and with results that "validate the efficiency of the proposed solver". In the following, we first reiterate the main results of our paper. Next, we address main issues raised by the reviewers.

## Main Results
* We find a generalization of Neyman-Pearson lemma (Thm. 4.1) using which we can solve a variety of constrained learning problems, including multi-objective L2D problems.
* We formulate the multi-objective L2D problem when the constraints are corresponded to expert intervention budget, OOD detection, long-tail classification, type-$k$ error, demographic parity, equality of opportunity, and equalized odds in terms of this generalization of N-P lemma.
* We find the parameters of this solution in closed-form (Thm. 4.2)
* We closely analyze the sample complexity of the prediction of this solution from empirical data (Thm. 5.1 and 5.3)
* We find the hardness of constrained L2D problems in lack of randomness (Thm. 3.1)
* We have an impossibility result of in-processing methods (Prop F.2)
* We have experiments on controlling demographic parity and equalized odds for ACS and COMPAS datasets. (in the rebuttal, we introduce a new set of experiments of Hatespeech dataset and for multi-constraint setting)

### Typos and Proofreading
We have put an extensive time in proofreading the manuscript and we have further considered all the typos that are raised by the esteemed reviewers. In the new version of manuscript, $\psi_0$ corresponds to the objective, and $\psi_1, \ldots, \psi_{m+1}$ corresponds to the constraints, $f$ is the vector prediction function, $L$ is the number of classes, and $n$ is the size of validation dataset. Moreover, the capital letters are reserved for random variables, and the small letters are reserved for the scalar values. Furthermore, Algorithm 1 is modified notation-wise, and to support multiple constraints.

### Multiple-Constraint (MC) Setting
1. We first note that the main result of this paper, **Theorem 4.1**, which formulated the **Bayes optimal** solution of the constrained optimization problem (3) is designed to handle **multiple constraints simultaneously**.

2. Theorem 4.2 is merely written to simplify the search for coefficient $k$ into finding the root of a monotone function. Although in MC setting, each constraint is still monotone in terms of each coefficient $k_i$ (constraints are marginally monotone), this does not reduce the complexity, since there might be more than one solution of $k_1, \ldots, k_m$ to achieve $\delta_1, \ldots, \delta_m$. Instead, we should use search methods to find the correct coefficients for our dataset.

3. In the **experiment** section, the ACS dataset is tested for equalized odds constraints, which keeps the differences between the true positive rate of two demographic groups as well as the true negative rate of the two groups controlled. **This experiment is a multi-constraint optimization** with the results reflected in the right-most subfigure in Figure 2 of the manuscript. Further experiments are introduced in Reviewer 4QNG rebuttal.

 4. **The algorithm can support the MC setting**, by just adding one line in which we search to find the values of $k_1, \ldots, k_m$ that are empirical solutions to  (3) for the validation dataset.

 5. The **sample complexity analysis for the constraints in Theorem 5.1** are **easily generalizable** to MC setting. If we have $m$ constraints with the empirical value of them being bounded by $\delta_1, \ldots, \delta_m$, then by $m$ times using Theorem 5.1 and using union bound on probabilities, we can show that the true constraint values are bounded by $\delta_1+d_n(\epsilon/m), \ldots, \delta_m+d_n(\epsilon/m)$, respectively with probability $1-\sum_{i=1}^m \epsilon/m = 1-\epsilon$.

 6. The MC extension of **sample complexity analysis for the objective of Theorem 5.3** although not readily but **is achievable by modification of the proof** and the assumption that **true scores are available**, and the optimization is done on the empirical dataset. In the proof of this theorem, in (77), we have offered a decomposition of the objective generalization Lagrangian generalization $D_{k}(f_1, f_2)$ and constraint generalization. We can define a similar Lagrangian generalization $D_{k_1, \ldots, k_m}(f_1, f_2)$. Then, following the same steps as in (80), and the equation after, and using Lipschitzness of $t_x(k)$, we can show that this value is bounded above by $2\sum_{i=1}^m |k_i^*-\hat k_i|$ . Furthermore, in case that each constraint achieves $\delta_i$, we can simply repeat the discussion above (78) and show that the constraints generalize. Therefore, it is remained to show that $k_i^*$ and $\hat k_i$ are closed to each other.  This entails that Definition 5.2 must hold for all $m$ constraints, i.e., the condition is $\big|\mathbb{E}\big[\langle f_{k}(x)-f_{k'}(x), \psi_{i}(x)\rangle\big]\big|\geq C\delta^{\gamma}$ where $||k-k'||\geq \delta$.  Next, due to the generalization of the constraints, we have $\mathbb{E}[\langle f_{\hat{k}}(x), \psi_i(x)\rangle]\in [\delta_i-d_n(\epsilon), \delta_i+d_n(\epsilon)]$ with probability at least $1-\epsilon$. Therefore, we can conclude that $d_n(\epsilon)\geq C\delta^{\gamma}$, or equivalently, $D_{\hat{k}_1, \ldots, \hat{k}_m}(f_1, f_2)$ is bounded above by $2m(d_n(\epsilon)/C)^{1/\gamma}$. This completes the proof.
 In multi-constraint setting and if we don't have the correct scores, we should use other proof techniques. The reason is that our proof is based on closeness of $\hat{k}$ and $k^*$ and in this case, we cannot assure their closeness with each other, although the objectives might still be close to each other.

---

### Decision · Program_Chairs · 2024-09-25

**Decision:**

Accept (poster)

**Comment:**

The submitted paper was reviewed by four reviewers, three of whom recommend its acceptance while one recommends its (borderline) rejection. A (partly intensive) exchange between authors and reviewers took place in the discussion phase. Overall, the reviewers appreciate the papers' proposed post-processing framework and the theoretical contributions. Concerns around notation, discussion of limitations, requirements for the proposed framework ([calibrated] estimation of the scores/probabilities), and parts of the experimental evaluation (multiple constraints, comparison with related works, ...) have been raised. The authors responded well to these concerns and were able to refute most of them - mainly a few concerns regarding the experimental evaluation remain. The paper is quite extensive and already covers a lot of interesting experimental results and even more came up during the discussion phase. Thus I am recommending the acceptance of the paper but strongly encourage the authors to incorporate the reviewers' comments carefully (in particular also discussing limitations in comparison to related works but also including the newly presented experimental results) when preparing the camera-ready version of their paper.